# Online (Multinomial) Logistic Bandit: Improved Regret and Constant Computation Cost

**Yu-Jie Zhang**[1]    **Masashi Sugiyama**[2,1]
[1] The University of Tokyo, Chiba, Japan
[2] RIKEN AIP, Tokyo, Japan

## Abstract

This paper investigates the logistic bandit problem, a variant of the generalized linear bandit model that utilizes a logistic model to depict the feedback from an action. While most existing research focuses on the binary logistic bandit problem, the multinomial case, which considers more than two possible feedback values, offers increased practical relevance and adaptability for use in complex decision-making problems such as reinforcement learning. In this paper, we provide an algorithm that enjoys both statistical and computational efficiency for the logistic bandit problem. In the binary case, our method improves the state-of-the-art binary logistic bandit method by reducing the per-round computation cost from $\mathcal{O}(\log T)$ to $\mathcal{O}(1)$ with respect to the time horizon $T$, while still preserving the minimax optimal guarantee up to logarithmic factors. In the multinomial case, with $K + 1$ potential feedback values, our algorithm achieves an $\widetilde{\mathcal{O}}(K\sqrt{T})$ regret bound with $\mathcal{O}(1)$ computational cost per round. The result not only improves the $\widetilde{\mathcal{O}}(K\sqrt{\kappa T})$ bound for the best-known tractable algorithm—where the large constant $\kappa$ increases exponentially with the diameter of the parameter domain—but also reduces the $\mathcal{O}(T)$ computational complexity demanded by the previous method.

## 1 Introduction

The stochastic linear bandit (SLB) [1, 2, 3] problem is a natural generalization of the classic stochastic multi-armed bandit problem [4] by incorporating side information into the decision-making process. In the SLB problem, a linear model is used to characterize the relationship between the reward $r_t \in \mathbb{R}$ and the learner's action $\mathbf{x}_t \in \mathcal{X} \subseteq \mathbb{R}^d$, whereas such an assumption is not always satisfied in real-world applications. Consequently, various models have been developed to account for the non-linear reward, including the generalized linear bandit (GLB) model [5] and kernelized bandit model [6]. The logistic bandit is a specific kind of GLB model by connecting the learner's $d$-dimensional action and the reward with a logistic model. Most existing work focuses on the binary case [7, 8, 9, 10]. The reward $r_t \in \{0, 1\}$ exhibits a binary value and the probability is modeled by $\Pr[r_t = 1 \mid \mathbf{x}_t] = \sigma(\mathbf{w}_*^\top \mathbf{x}_t)$, where $\sigma(z) = 1/(1 + \exp(-z))$ is a non-linear link function and $\mathbf{w}_* \in \mathcal{W} \subseteq \mathbb{R}^d$ is an unknown parameter. Compared to the SLB model, the logistic bandit model provides a more precise representation for a wide range of real-world application problems, where feedback exhibits discrete behavior. Moreover, from a theoretical perspective, it also serves as a basic setting for understanding the impact of non-linearity of the reward on the decision-making process. In this paper, we investigate a more general *multinomial logistic bandit* (MLogB) problem [11], in which the learner's action $\mathbf{x}_t$ results in feedback $y_t$ that could have $K + 1$ possible outcome values. The probability of each outcome is characterized with a logistic model (the formal definition is provided in Section 2.1). The MLogB model is of more practical interest compared to the binary case. For example, in the real-world application such as online advertising, there could be multiple possible feedback from customers, including "buy now", "add to cart", "view related item", and

37th Conference on Neural Information Processing Systems (NeurIPS 2023).

Table 1: Comparison in terms of the regret bound, computation cost and storage cost. For the regret bound, the logarithmic dependence on the time horizon $T$ is hidden by the $\widetilde{\mathcal{O}}(\cdot)$-notation. As for the computational cost (abbreviated as "Comput.") and storage cost, we only keep the dependence on the time step $t$. Notation "-" denotes that the algorithm is intractable for implementation.

| Setting | Algorithm | Regret | Comput. per Round | Storage Cost | Improved $\kappa$ | Constant Cost |
|---|---|---|---|---|---|---|
| binary | Logistic-UCB-1 [8] | $\widetilde{\mathcal{O}}(\sqrt{\kappa T})$ | $\mathcal{O}(t)$ | $\mathcal{O}(t)$ | ✗ | ✗ |
| | OFULog-r [9] | $\widetilde{\mathcal{O}}(\sqrt{T/\kappa_*})$ | $\mathcal{O}(t)$ | $\mathcal{O}(t)$ | ✓ | ✗ |
| | (ada)-OFU-ECOLog [10] | $\widetilde{\mathcal{O}}(\sqrt{T/\kappa_*})$ | $\mathcal{O}(\log t)$ | $\mathcal{O}(1)$ | ✓ | ✗ |
| | OL2M [7],GLOC [14] | $\widetilde{\mathcal{O}}(\kappa\sqrt{T})$ | $\mathcal{O}(1)$ | $\mathcal{O}(1)$ | ✗ | ✓ |
| | OFU-MLogB (Corollary 1) | $\widetilde{\mathcal{O}}(\sqrt{T/\kappa_*})$ | $\mathcal{O}(1)$ | $\mathcal{O}(1)$ | ✓ | ✓ |
| multinomial | MNL-UCB [11] | $\widetilde{\mathcal{O}}(K\sqrt{\kappa T})$ | $\mathcal{O}(t)$ | $\mathcal{O}(t)$ | ✗ | ✗ |
| | Improved MNL-UCB [11] | $\widetilde{\mathcal{O}}(K^{3/2}\sqrt{T})$ | — | $\mathcal{O}(t)$ | ✓ | – |
| | OFUL-MLogB (Theorem 4) | $\widetilde{\mathcal{O}}(K\sqrt{T})$ | $\mathcal{O}(1)$ | $\mathcal{O}(1)$ | ✓ | ✓ |

just leave without any click. Beyond addressing practical demands, studying the MLogB problem can shed light on other decision-making problems. For instance, in the theoretical reinforcement learning research [12, 13], the transition matrix is approximated using a linear model, which enables the application of the SLB method for balancing the exploration-exploitation trade-off. Since the transition matrix is inherently a probability matrix, the multinomial logistic model may be a more suitable structure for modeling transition probabilities between states.

As shown in Table 1, the logistic bandit problem has received much attention in the binary case. There are two main focuses arising from the non-linearity of the reward function: *statistical efficiency* and *resource overhead*. Regarding the statistical efficiency, a main focus is the algorithms' dependence on $\kappa = \max_{\mathbf{x}\in\mathcal{X}, \mathbf{w}\in\mathcal{W}} 1/\sigma'(\mathbf{w}^\top\mathbf{x})$, a crucial parameter capturing the non-linearity of the reward function. Since the binary logistic bandit is a special case of the generalized linear model, the result from [5] implies an $\widetilde{\mathcal{O}}(\kappa\sqrt{T})$ bound. However, the parameter $\kappa$ grows exponentially in terms of the diameter of the decision domain $\mathcal{W}$ and action space $\mathcal{X}$, making the linear dependence unfavorable. A pivotal advancement is made by [9], where the paper presents an algorithm that achieves the nearly minimax optimal bound $\widetilde{\mathcal{O}}(\sqrt{T/\kappa_*})$. Here, $\kappa_* = 1/\sigma'(\mathbf{w}_*^\top\mathbf{x}_*)$ is the non-linear parameter associated with the best action $\mathbf{x}_* = \arg\max_{\mathbf{x}\in\mathcal{X}}\sigma(\mathbf{w}_*^\top\mathbf{x})$, suggesting that non-linearity might be advantageous for statistical efficiency, rather than a hindrance. Alongside statistical efficiency considerations, the non-linearity of the feedback raises concerns about the algorithms' computation and storage efficiency. The methods [8, 9] with improved dependence on $\kappa$ usually require storing and optimizing over all historical data to estimate the unknown parameter, leading to an $\mathcal{O}(t)$ computation and storage cost at round $t \in [T]$. The pioneering work [7] provided the first efficient solution for binary logistic bandit with constant computation and storage costs and [14] further proposed an efficient algorithm for generalized linear bandit, but their regret bounds still exhibit a linear dependence on $\kappa$. Recently, a jointly efficient algorithm was proposed by [10], which achieves the optimal dependence on $\kappa$ with a $\mathcal{O}(\log t)$ computation cost and constant storage cost per round. However, it remains open whether the minimax optimal bound is achievable with constant computation cost independent of the time $t$.

Regrading the multinomial logistic bandit, the best-known feasible algorithm was proposed by [11]. This method achieves an $\widetilde{\mathcal{O}}(K\sqrt{\kappa T})$ regret bound, bearing an $\mathcal{O}(\sqrt{\kappa})$ dependence on the exponentially large constant. Moreover, it still demands $\mathcal{O}(t)$ computation and storage costs to optimize over all past data. The same study also introduced an $\widetilde{\mathcal{O}}(K^{3/2}\sqrt{T})$ bound with improved dependence on $\kappa$. Yet, this solution leans heavily towards theoretical insights and is intractable in implementation [11, Section 2.6]. Designing a practical or more efficient algorithm with improved dependence on $\kappa$ is still an unsolved challenge. More discussions on the related work and topics can be found in Section 4.

**Our Results.** In this paper, we provide an algorithm with both statistical and computational efficiency.

- For the multinomial logistic bandit, we propose OFUL-MLOGB, a jointly efficient method attaining an $\widetilde{\mathcal{O}}(K\sqrt{T})$ regret bound with $\mathcal{O}(1)$ in $T$ computation and storage cost per round. The result improves the previous work on the dependence of the large constant $\kappa$.
- For the binary case, our proposed OFUL-MLOGB can achieve the $\widetilde{\mathcal{O}}(\sqrt{T/\kappa_*})$ optimal bound up to logarithmic factors. Besides, our method reduces the computation cost of the state-of-the-art binary method [10] from $\mathcal{O}(\log t)$ to $\mathcal{O}(1)$ per round.

## 2 Multinomial Logistic Bandit with Improved Regret

This section provides preliminaries on the multinomial logistic bandit problem and optimistic algorithms, beginning with the problem formulation and notations. Then, we revisit the optimistic algorithms for the MLogB problem. Specifically, we investigate the previously best-known feasible algorithm, MNL-UCB algorithm [11], and propose an improved version with better dependence on the exponentially large constant $\kappa$ and the number of outcome values $K$.

### 2.1 Problem Formulation

The multinomial logistic bandit (MLogB) problem studies a $T$ round decision-making process between the learner and the environment. At the beginning of the iteration $t$, the learner will first select an action $\mathbf{x}_t \in \mathcal{X}$ from the feasible action set $\mathcal{X} \subseteq \mathbb{R}^d$ and then submit it to the environment. After that, a response $y_t \in \{0\} \cup [K]$ with $K + 1$ possible outcomes (like "buy now", "add to chart", or "do nothing") is returned based on the learner's choice, where $K \in \mathbb{N}$. Specifically, the MLogB problem assumes that each outcome $k \in [K]$ is associated with a ground-truth parameter $\mathbf{w}_*^{(k)} \in \mathbb{R}^d$ and the probability of the outcome $\Pr[y_t = k \mid \mathbf{x}_t]$ follows the logistic model,

$$\Pr[y_t = k \mid \mathbf{x}_t] = \frac{\exp\left((\mathbf{w}_*^{(k)})^\top \mathbf{x}_t\right)}{1 + \sum_{j=1}^K \exp\left((\mathbf{w}_*^{(j)})^\top \mathbf{x}_t\right)} \quad \text{and} \quad \Pr[y_t = 0 \mid \mathbf{x}_t] = 1 - \sum_{k=1}^K \Pr[y_t = k \mid \mathbf{x}_t].$$

For notational simplicity, we denote by $W_* = [\mathbf{w}_*^{(1)}, \ldots, \mathbf{w}_*^{(K)}]^\top \in \mathbb{R}^{K \times d}$ the matrix for the unknown parameter and define the softmax function $\boldsymbol{\sigma} : \mathbb{R}^K \mapsto [0, 1]^K$ by

$$[\boldsymbol{\sigma}(\mathbf{z})]_k = \frac{\exp([\mathbf{z}]_k)}{1 + \sum_{j=1}^K \exp([\mathbf{z}]_j)} \text{ for all } k \in [K] \quad \text{and} \quad [\boldsymbol{\sigma}(\mathbf{z})]_0 = \frac{1}{1 + \sum_{j=1}^K \exp([\mathbf{z}]_j)}, \quad (1)$$

where $[\cdot]_k$ denotes the $k$-th entry of the input vector. Then, the probability of the outcome can also be written in a concise way as $\Pr[y_t = k \mid \mathbf{x}_t] = [\boldsymbol{\sigma}(W_* \mathbf{x}_t)]_k$. Besides, each outcome is associated with a fixed and known reward. We denote by $\rho_k \in \mathbb{R}_+$ the reward for the outcome $k \in [K]$, and let $\rho_0 = 0$ for the outcome $y_t = 0$. Therefore, the expected reward of the learner's action $\mathbf{x}_t$ is defined as $r(\mathbf{x}_t) = \sum_{k=0}^K \Pr[y_t = k \mid \mathbf{x}_t] \cdot \rho_k = \boldsymbol{\rho}^\top \boldsymbol{\sigma}(W_* \mathbf{x}_t)$. Let $\mathbf{x}_* = \arg\max_{\mathbf{x} \in \mathcal{X}} \boldsymbol{\rho}^\top \boldsymbol{\sigma}(W_* \mathbf{x})$. The goal of the learner is to maximize the cumulative reward, which is equivalent to minimizing the regret

$$\text{Reg}_T = \sum_{t=1}^T \boldsymbol{\rho}^\top \left( \boldsymbol{\sigma}(W_* \mathbf{x}_*) - \boldsymbol{\sigma}(W_* \mathbf{x}_t) \right), \quad (2)$$

When $K = 1$ and $\rho_1 = 1$, MLogB recovers the binary logistic bandit by $r(\mathbf{x}) = \sigma(\mathbf{w}_*^\top \mathbf{x}) = 1/(1 + \exp(-\mathbf{w}_*^\top \mathbf{x}))$, where $\mathbf{w}_* \in \mathbb{R}^d$ is a unknown parameter.

**Exponentially Large Constant $\kappa$.** In the logistic bandit problem, the non-linearity of the reward function is captured by the gradient of the link function $\nabla \boldsymbol{\sigma} : \mathbf{z} \in \mathbb{R}^d \mapsto \text{diag}(\boldsymbol{\sigma}(\mathbf{z})) - \boldsymbol{\sigma}(\mathbf{z}) \boldsymbol{\sigma}(\mathbf{z})^\top$. The analysis typically that requires that the gradient term is bounded from below and thus one would define the constant $\kappa \triangleq 1/\min_{W \in \mathcal{W}, \mathbf{x} \in \mathcal{X}} \lambda_{\min}(\nabla \boldsymbol{\sigma}(W\mathbf{x}))$ such that $\frac{1}{\kappa} I_d \preccurlyeq \nabla \boldsymbol{\sigma}(W\mathbf{x})$ for any $W \in \mathcal{W}$ and $\mathbf{x} \in \mathcal{X}$, where $\lambda_{\min} : \mathbb{R}^{K \times K} \to \mathbb{R}^K$ is the minimum eigenvalue of the input matrix. In the binary case ($K = 1$), one can show that $\kappa = \max_{\mathbf{w} \in \mathcal{W}, \mathbf{x} \in \mathcal{X}} \{1 + \exp(\mathbf{w}^\top \mathbf{x}) + \exp(-\mathbf{w}^\top \mathbf{x})\} = \mathcal{O}(e^{SX})$, where $S$ and $X$ are the diameters of the parameter space $\mathcal{W}$ and action space $\mathcal{X}$. In the multinomial case, the paper [11, Section 3] also shows that $\kappa$ is an exponentially large constant with respect to $S$ and $X$. Thus, an algorithm with improved dependence on $\kappa$ is demanded.

### 2.2 Assumptions and Notations

Same as the previous work for multinomial logistic bandit [11], we use the following assumptions.

**Assumption 1.** The norm of the action is bounded by 1, i.e., $\|\mathbf{x}\|_2 \le 1$ for any $\mathbf{x} \in \mathcal{X}$.

**Assumption 2.** The reward vector $\boldsymbol{\rho} \in \mathbb{R}_+^K$ and its norm is bounded by $R$, i.e., $\|\boldsymbol{\rho}\|_2 \le R$.

**Assumption 3.** The norm of the parameter $W_* \in \mathbb{R}^{K \times d}$ is bounded by $S$, i.e., $\|W_*\|_F \le S$, where $\|\cdot\|_F$ denotes the Frobenius norm of a matrix.

**Assumption 4.** Let $\nabla\sigma(\mathbf{z}) : \mathbf{z} \in \mathbb{R}^d \mapsto \mathrm{diag}(\boldsymbol{\sigma}(\mathbf{z})) - \boldsymbol{\sigma}(\mathbf{z})\boldsymbol{\sigma}(\mathbf{z})^\top$. For all $\mathbf{x} \in \mathcal{X}$ and $W \in \mathcal{W}$, we have $\lambda_{\min}(\nabla\sigma(W\mathbf{x})) \geq 1/\kappa$ and $\lambda_{\max}(\nabla\sigma(W\mathbf{x})) \leq L$, where $\lambda_{\max} : \mathbb{R}^{K \times K} \to \mathbb{R}^K$ and $\lambda_{\min} : \mathbb{R}^{K \times K} \to \mathbb{R}^K$ take the maximum and minimum eigenvalues of the input, respectively.

**Other Notations.** The following notations are used in the paper. Given a $K$-by-$d$ matrix $W$, we denote by $\overrightarrow{W}$ its $Kd$-dimensional vectorization. For any positive semi-definite $H \in \mathbb{R}^{Kd \times Kd}$, we define the norm $\|\overrightarrow{W}\|_H = \sqrt{\langle \overrightarrow{W}, H\overrightarrow{W}\rangle}$. The notation $\otimes$ is used for the standard Kronecker product. When the input is a vector $\overrightarrow{W} \in \mathbb{R}^{Kd}$, we treat it as a $Kd \times 1$ matrix. Moreover, for any symmetric matrix $A, B \in \mathbb{R}^{Kd \times Kd}$, we denote by $A \succeq B$ that $A - B$ is a semi-positive definite matrix. We use $\mathcal{F}_t = \sigma(\mathbf{x}_1, y_1, \dots, y_{t-1}, \mathbf{x}_t)$ to denote the filtration, which encodes the information collected so far before receiving $y_t$. Finally, $\mathcal{O}(\cdot)$ is used to highlight the dependence on $d$, $K$, $\kappa$, and $T$. With $\widetilde{\mathcal{O}}(\cdot)$-notation, we further hide the dependence on the dimension $d$ and logarithmic factors.

## 2.3 Optimistic Algorithm with Improved Bound

This part revisit the principle of optimism in the face of uncertainty (OFU) [15] and introduces an improved version of the MNL-UCB algorithm [11] with better dependence on $\kappa$ and $K$.

**Optimism in the Face of Uncertainty.** The OFU principle is a fundamental paradigm for addressing the exploration-exploitation dilemma in bandits. At each iteration $t$, the algorithm selects the arm by the rule $\mathbf{x}_t = \arg\max_{\mathbf{x}\in\mathcal{X}} \widetilde{r}_t(\mathbf{x})$, where $\widetilde{r}_t(\mathbf{x})$ is an optimistic estimate of the true reward $r(\mathbf{x})$ satisfying $\widetilde{r}_t(\mathbf{x}) \geq r(\mathbf{x})$ for all $\mathbf{x} \in \mathcal{X}$. Based on the OFU rule, one can show that $\mathrm{Reg}_T \leq \sum_{t=1}^T (\widetilde{r}(\mathbf{x}_t) - r(\mathbf{x}_t))$, indicating that a tighter optimistic estimate $\widetilde{r}_t$ will lead to a tighter regret bound. Therefore, designing a tight optimistic estimate $\widetilde{r}_t$ is essential for the OFU algorithm.

In the context of the logistic bandit, a common practice is to construct a confidence set $\mathcal{C}_t \subset \mathbb{R}^{K \times d}$, which is supposed to contain the true parameter $W_*$ with high probability. As such, the learner can obtain the optimistic reward for each arm $\mathbf{x} \in \mathcal{X}$ by $\widetilde{r}_t(\mathbf{x}) = \arg\max_{W \in \mathcal{C}_t} \boldsymbol{\rho}^\top \boldsymbol{\sigma}(W\mathbf{x})$. A tighter confidence set will lead to a tighter optimistic reward, and thus resulting in a better regret bound.

**Improved Concentration Set.** Given the reward is generated by the multinomial logistic model, we can employ the maximum likelihood estimation (MLE) method to learn the unknown parameter $W_*$. Specifically, after observing the action-feedback pairs $\{(\mathbf{x}_s, y_s)\}_{s=1}^{t-1}$, one can train the model by

$$W_t^{\mathtt{MLE}} = \arg\min_{W \in \mathbb{R}^{K \times d}} \mathcal{L}_t(W) \triangleq \sum_{s=1}^{t-1} \ell_s(W) + \frac{\lambda}{2}\|W\|_{\mathrm{F}}^2, \tag{3}$$

where $\ell_t(W) = \sum_{k=0}^K \mathbb{1}\{y_t = k\} \cdot \log\left(1/[\boldsymbol{\sigma}(W\mathbf{x}_t)]_k\right)$ is the multiclass logistic loss established over $(\mathbf{x}_t, y_t)$ and $\lambda > 0$ is the regularization parameter. Let $\overrightarrow{W} \in \mathbb{R}^{Kd}$ be the vectorized parameter. We also define the gradient $\mathbf{g}_t(\overrightarrow{W})$ and the Fisher information matrix $H_t(W)$ of the logistic loss by

$$\mathbf{g}_t(\overrightarrow{W}) \triangleq \nabla\mathcal{L}_t(\overrightarrow{W}) \quad \text{and} \quad H_t(W) \triangleq \lambda I + \sum_{s=1}^{t-1} \nabla\boldsymbol{\sigma}(W\mathbf{x}_s) \otimes \mathbf{x}_s\mathbf{x}_s^\top,$$

where $\overrightarrow{W} \in \mathbb{R}^{Kd}$ is the vectorized parameter and $\nabla\boldsymbol{\sigma} : \mathbf{z} \mapsto \mathrm{diag}(\boldsymbol{\sigma}(\mathbf{z})) - \boldsymbol{\sigma}(\mathbf{z})\boldsymbol{\sigma}(\mathbf{z})^\top$ is the first order derivative of the reward vector $\boldsymbol{\sigma}(\mathbf{z})$. Then, we are ready to present our confidence set for the maximum likelihood estimator, which exhibits $\mathcal{O}(\sqrt{K})$ improvement over that in [11].

**Theorem 1.** *Set the parameter* $\lambda = \mathcal{O}(dK\log(t/\delta))$ *with a certain* $\delta \in (0, 1]$. *For each iteration* $t \in [T]$, *we define the confidence set as*

$$\mathcal{C}_t(\delta) \triangleq \left\{ W \in \mathcal{W} \ \middle| \ \left\|\mathbf{g}_t(\overrightarrow{W}) - \mathbf{g}_t(\overrightarrow{W}_t^{\mathtt{MLE}})\right\|_{H_t^{-1}(W)} \leq \beta_t(\delta) \right\}, \tag{4}$$

*where* $\beta_t(\delta) = 4\sqrt{Kd(1+S)\log\left(2\left(1 + t/d\right)/\delta\right)} = \mathcal{O}(\sqrt{dK\log t})$ *is the radius of the set and* $\mathcal{W} = \{W \in \mathbb{R}^{K \times d} \mid \|W\|_{\mathrm{F}} \leq S\}$. *Then, we have* $\Pr\left[\forall t \geq 1, W_* \in \mathcal{C}_t(\delta)\right] \geq 1 - \delta$.

One advantage of the confidence set (4) is that its radius $\beta_t(\delta)$ is independent of the exponentially large constant $\kappa$ and thus is much tighter than the confident set constructed for the generalized

---

**Algorithm 1** MNL-UCB+

---

**Input:** regularization coefficient $\lambda$, probability $\delta$.

1: Initialize $H_1 = \lambda I_{Kd}$ and $\overrightarrow{W}_1^{\texttt{MLE}}$ as any point in $\mathcal{W}$
2: **for** $t = 1, \ldots, T$ **do**
3:    Construct $\widetilde{r}_t$ and select the arm by (5). Then, the learner receives $y_t$.
4:    Train the estimator $W_{t+1}^{\texttt{MLE}}$ by (3) and construct the confidence set $\mathcal{C}_{t+1}(\delta)$ as (4).
5: **end for**

---

linear bandit [5], whose radius exhibits a linear dependence on $\kappa$. The $\kappa$-independent set are first established by [8] for the binary logistic bandit problem and then adapted to multinomial setting by [11] with refined analysis on the self-normalized martingale tail-inequality for multinomial noise. Our confidence set is tighter than that in [11] by improving the radius from $\beta_t(\delta) = \mathcal{O}(K\sqrt{d \log t})$ to $\beta_t(\delta) = \mathcal{O}(\sqrt{dK \log t})$. The improvement is based on a slightly refined self-normalized tail-inequality for the multinomial case, whose formal description is provided in Appendix C.1.

**Construction of Optimistic Reward.** Based on the confidence set (4), we can construct the optimistic reward and select the arm for each iteration by

$$\widetilde{r}_t(\mathbf{x}) = \arg\max_{W \in \mathcal{C}_t(\delta)} \boldsymbol{\rho}^\top \boldsymbol{\sigma}(W\mathbf{x}) \quad \text{and} \quad \mathbf{x}_t = \arg\max_{\mathbf{x} \in \mathcal{W}} \widetilde{r}_t(\mathbf{x}). \tag{5}$$

We have following guarantee for the algorithm based on the MLE (3) and the selection rule specified in (4) and (5). We summarize the algorithmic procedure in Algorithm 1.

**Theorem 2.** *Under the same conditions as Theorem 1. Let $\delta \in (0, 1]$. Algorithm 1 ensures*

$$\text{Reg}_T \leq \mathcal{O}\left(\min\{d \log T \sqrt{\kappa KT}, dK \log T \sqrt{T} + \kappa d^2 K \log^2 T\}\right)$$

*with probability at least $1 - \delta$ when we set the parameter $\lambda = \mathcal{O}(dK \log(T/\delta))$.*

**Remark 1** (Improved dependence on $K$ and $\kappa$)**.** Our method achieves the $\widetilde{\mathcal{O}}(\sqrt{\kappa KT})$ and $\widetilde{\mathcal{O}}(K\sqrt{T})$ regret bounds for MLogB problem simultaneously. The first bound slightly improves the $\widetilde{\mathcal{O}}(K\sqrt{\kappa T})$ guarantee of the MNL-UCB algorithm [11] by an $\mathcal{O}(\sqrt{K})$ factor while the second one is independent of the exponentially large constant $\kappa$ in its leading term. An $\widetilde{\mathcal{O}}(K^{\frac{3}{2}}\sqrt{T})$ bound is also attained by [11]. However, their proposed method is intractable as its confidence set is established on all minimal elements of partially Loewner-ordered set $C_t(\delta) \cap \mathcal{W}$ [11, Appendix D]. The computation cost of identifying all minimal elements and projecting onto the proposed confidence set is prohibitive. Our solution is free from such demands by using a different rule to construct the optimistic reward.

## 3   Jointly Efficient Algorithm

In this section, we introduce OFUL-MLogB, an algorithm with jointly computational and statistical efficiency for the MLogB problem. We will first discuss the efficiency concern of the existing methods in the literature and then introduce our algorithm, followed by a technical highlight.

### 3.1   Efficiency Concerns

The algorithms for logistic bandit crucially rely on two components to ensure the statistical efficiency: the MLE (3) and the optimistic rule (5) for constructing $\widetilde{r}_t$. However, the implementation of both components could be inefficient by requiring $\mathcal{O}(t)$ computation cost per online iteration.

**Computation and Storage Cost of Maximum Likelihood Estimation.** For logistic bandit or even the generalized linear bandit problem [3, 8, 9, 11], MLE is a widely used tool to learn the unknown parameter. To solve the optimization problem, the gradient-based method, e.g. the projected gradient descent [16], are usually applied. However, as discussed in [10], the optimization of the MLE problem typically requires $\mathcal{O}(t \log(1/\epsilon))$ gradient step to achieve $\epsilon$-accuracy. Besides, the loss function $\mathcal{L}_t$ is established on all historical data $\{(\mathbf{x}_s, y_s)\}_{s=1}^{t-1}$, resulting in an $\mathcal{O}(t)$ gradient query complexity for each gradient step and $\mathcal{O}(t)$ storage cost, and thus is inefficient.

**Computation and Storage Cost of Optimistic Reward Construction.** The construction of the optimistic reward $\widetilde{r}_t(\mathbf{x})$ requires to solve the optimization problem (5). However, the objective

---

**Algorithm 2** OFUL-MLogB

---

**Input:** regularization coefficient $\lambda$, probability $\delta$, step size $\eta$.

1: Initialize $H_1 = \lambda I_{Kd}$ and $\overrightarrow{W}_1^{\texttt{OL}}$ as any point in $\mathcal{W}$
2: **for** $t = 1, \ldots, T$ **do**
3:     Select the arm by $\mathbf{x}_t = \arg\max_{\mathbf{x} \in \mathcal{X}} \widetilde{r}_t(\mathbf{x})$ and receive $y_t$.
4:     Update $\widetilde{H}_t = H_t + \eta \nabla \boldsymbol{\sigma}(W_t^{\texttt{OL}} \mathbf{x}_t) \otimes \mathbf{x}_t \mathbf{x}_t^{\top}$
5:     Update the estimator $\overrightarrow{W}_{t+1}^{\texttt{OL}}$ for the next iteration by (6)
6:     Update $H_{t+1} = H_t + \nabla \boldsymbol{\sigma}(W_{t+1}^{\texttt{OL}} \mathbf{x}_t) \otimes \mathbf{x}_t \mathbf{x}_t^{\top}$ and
7:     Construct the optimistic reward by $\widetilde{r}_{t+1}(\mathbf{x}) = \boldsymbol{\rho}^{\top} \boldsymbol{\sigma}(W_{t+1}^{\texttt{OL}} \mathbf{x}) + \epsilon_{t+1}^{\texttt{fst}}(\mathbf{x}) + \epsilon_{t+1}^{\texttt{snd}}(\mathbf{x})$ as (8).
8: **end for**

---

function $\boldsymbol{\rho}^{\top} \boldsymbol{\sigma}(W\mathbf{x})$ is non-concave and the decision domain $\mathcal{C}_t(\delta)$ is non-convex, making the maximization problem $\widetilde{r}_t(\mathbf{x}) = \arg\max_{W \in \mathcal{C}_t(\delta)} \boldsymbol{\rho}^{\top} \boldsymbol{\sigma}(W\mathbf{x})$ computationally challenging. In the binary case, the paper [9] proposed a convex relaxation of the confidence set $\mathcal{C}_t(\delta)$, whereas the relaxed confidence set is still established on all historical data, resulting in $\mathcal{O}(t)$ time complexity and $\mathcal{O}(t)$ storage cost at iteration $t$. The optimistic estimate construction in the previous work for multinomial logistci bandits [11] is computationally efficient without involving any optimization problem solving. However, it will lead to an inferior regret bound of $\widetilde{\mathcal{O}}(\sqrt{\kappa T})$.

In the binary case, the work [10] proposed a jointly efficient algorithm, which achieved a nearly minimax optimal bound with $\mathcal{O}(\log t)$ computation cost per iteration and $\mathcal{O}(1)$ storage cost. However, as we will discuss in Section 3.3, it is hard to apply their analysis to the multinomial case.

### 3.2 Efficient Algorithm

In this section, we proposed a novel algorithm which only requires $\mathcal{O}(1)$ computation cost per iteration and $\mathcal{O}(1)$ storage cost. The algorithm can achieve the best known results both for binary and multinomial logistic bandits. We have introduced new ingredients both on the algorithm design and regret analysis to achieve the jointly efficient algorithm.

**Efficient Online Estimation.** Instead of performing MLE, we run an online mirror descent algorithm to estimate parameter:

$$\overrightarrow{W}_{t+1}^{\texttt{OL}} = \arg\min_{\overrightarrow{W} \in \mathcal{W}} \langle \nabla \ell_t(\overrightarrow{W}_t^{\texttt{OL}}), \overrightarrow{W} \rangle + \frac{1}{2\eta} \|\overrightarrow{W} - \overrightarrow{W}_t^{\texttt{OL}}\|_{\widetilde{H}_t}^2, \ \forall t \geq 1 \qquad (6)$$

where $\eta > 0$ is the step size to be specified later and the first iteration model $\overrightarrow{W}_1^{\texttt{OL}}$ can be initialized as any point in the domain $\mathcal{W} = \{\overrightarrow{W} \in \mathbb{R}^{Kd} \mid \|\overrightarrow{W}\|_2 \leq S\}$. We set the matrix as $\widetilde{H}_t = H_t + \eta \nabla \boldsymbol{\sigma}(W_t^{\texttt{OL}} \mathbf{x}_t) \otimes \mathbf{x}_t \mathbf{x}_t^{\top}$, where $H_t = \lambda I + \sum_{s=1}^{t-1} \nabla \boldsymbol{\sigma}(W_{s+1}^{\texttt{OL}} \mathbf{x}_s) \otimes \mathbf{x}_s \mathbf{x}_s^{\top}$. Both $\widetilde{H}_t$ and $H_t$ can be updated incrementally.

We show the online estimator (6) enjoys computational, storage, and statistical efficiency. Since (6) exhibits a standard online mirror descent formulation [17], it can be solved with a single projected gradient step with the following equivalent formulation by

$$\overrightarrow{Z}_{t+1} = \overrightarrow{W}_t^{\texttt{OL}} - \eta \widetilde{H}_t^{-1} \nabla \ell_t(\overrightarrow{W}_t^{\texttt{OL}}) \quad \text{and} \quad \overrightarrow{W}_{t+1}^{\texttt{OL}} = \arg\min_{\overrightarrow{W} \in \mathcal{W}} \|\overrightarrow{W} - \overrightarrow{Z}_{t+1}\|_{\widetilde{H}_t}.$$

For the gradient descent step above, the most time-consuming operation is maintaining the inverse of the matrix $\widetilde{H}_t$. Since $\nabla \boldsymbol{\sigma}(W_t^{\texttt{OL}} \mathbf{x}_t) \otimes \mathbf{x}_t \mathbf{x}_t^{\top}$ is a rank-$K$ matrix, it can be calculated by the Sherman-Morrison-Woodbury formula with $\mathcal{O}(d^2 K^3)$ cost per round. As for the projection step, since $\widetilde{H}_t$ is positive semi-definite matrix, it can be solved in $\mathcal{O}(K^3 d^3)$ [18, Section 4].* As a consequence, our algorithm achieves a light update with $\mathcal{O}(1)$ cost per round. Regarding storage cost, our proposed estimator eliminates the need to store all historical data and updates in a one-pass fashion, requiring only $\mathcal{O}(1)$ storage cost throughout the learning process. Moreover, the estimator is also statistically efficient. We can construct the following $\kappa$-independent confidence set similar to that in Theorem 1.

---

*In the high-dimensional case, one can also employ Lemma 13 of [10] to perform the projection step, which ensures $1/\tau$-error with $\mathcal{O}(d^2 \log \tau)$ computation complexity per iteration.

**Theorem 3.** *Let $\delta \in (0,1]$ and $\alpha = 2(1+S) + \ln(K+1)$. Set the parameter $\eta = \alpha/2$ and $\lambda = \max\{28Kd\alpha, 7\sqrt{6}\alpha S\}$. For each iteration $t \in [T]$, we define the confidence set as*

$$\mathcal{C}_t^{\texttt{OL}}(\delta) \triangleq \left\{ W \in \mathcal{W} \;\middle|\; \|\overrightarrow{W}_t^{\texttt{OL}} - \overrightarrow{W}\|_{H_t} \leq \beta_t^{\texttt{OL}}(\delta) \right\}, \tag{7}$$

*where $\beta_t^{\texttt{OL}}(\delta) = \mathcal{O}\big(\log K \log t \sqrt{Kd}\big)$. Then, we have $\Pr\big[\forall t \geq 1, W_* \in \mathcal{C}_t(\delta)\big] \geq 1 - \delta$. Moreover, the computation cost of solving* (6) *is $\mathcal{O}(1)$ per round.*

**Remark 2** (Comparison with the online estimator [10])**.** Our algorithm design is inspired by the online estimator [10] developed for the binary case, while achieving even lighter cost by novel algorithm ingredients and analysis. As discussed in Section 3.3, the analysis for the binary setting is hard to be applied to the multinomial case. Specifically, the paper [10] has proposed an intermediary decision $\overline{W}_t$ in the analysis to prove the statistical efficiency of the proposed estimator. However, the favorable property of the intermediate decision only holds in the binary case. To this end, we have proposed a new intermediary decisions, which not only help to prove the statistically efficient of our estimator but also eliminate the requirement of the exploration step. Besides, we have also introduced a novel algorithm ingredient to further speed-up the algorithm. Instead of learning with original loss function as in [10], by a more refined exploitation of the negative term in the analysis, we show it is sufficient to learn with the first order approximate $\langle \nabla \ell_t(\overrightarrow{W}_t^{\texttt{OL}}), \overrightarrow{W} \rangle$ of $\ell_t(W)$ with the adjusted local norm $\|\cdot\|_{\widetilde{H}_t}$. Our new algorithm not only enjoys a computation efficiency improvement from $\mathcal{O}(\log t)$ in [10] to $\mathcal{O}(1)$, but also is free from any exploration step required by the previous work. We provide a technical highlight in Section 3.3.

**Efficient Optimistic Reward Construction.** Although the confidence set $\mathcal{C}_t^{\texttt{OL}}(\delta)$ is convex, the optimistic rule (5) by $\widetilde{r}_t(\mathbf{x}) = \arg\min_{W \in \mathcal{C}_t^{\texttt{OL}}(\delta)} \boldsymbol{\rho}^\top \boldsymbol{\sigma}(W\mathbf{x})$ still involves inefficient non-concave optimization problem solving. In this part, we propose a novel optimistic reward that can be solved in a constant time per round.

**Proposition 1.** *For any $\mathbf{x} \in \mathcal{X}$ and iteration $t \in [T]$, the optimistic reward is constructed by*

$$\widetilde{r}_t^{\texttt{OL}}(\mathbf{x}) = \boldsymbol{\rho}^\top \boldsymbol{\sigma}(W_t^{\texttt{OL}}\mathbf{x}) + \epsilon_t^{\texttt{fst}}(\mathbf{x}) + \epsilon_t^{\texttt{snd}}(\mathbf{x}). \tag{8}$$

*In above, $\epsilon_t^{\texttt{fst}}(\mathbf{x}) = \beta_t^{\texttt{OL}}(\delta) \cdot \|H_t^{-\frac{1}{2}}(I_K \otimes \mathbf{x})\nabla\boldsymbol{\sigma}(W_t^{\texttt{OL}}\mathbf{x})\boldsymbol{\rho}\|_2$ and $\epsilon_t^{\texttt{snd}}(\mathbf{x}) = 3R\big(\beta_t^{\texttt{OL}}\big)^2 \cdot \|(I_K \otimes \mathbf{x}^\top)H_t^{-1/2}\|_2^2$ are the bonus. Then, we have $\widetilde{r}_t^{\texttt{OL}}(\mathbf{x}) \geq \boldsymbol{\rho}^\top \boldsymbol{\sigma}(W_*\mathbf{x})$ for all $t \geq 1$ and $\mathbf{x} \in \mathcal{X}$ with probability at least $1 - \delta$.*

Proposition 1 constructs the optimistic reward by adding the "bonus" to the reward empirically estimated by $W_t^{\texttt{OL}}$. Different from the term used in [11], our bonus terms are independent of the exponentially large constant and thus can lead to an improved $\widetilde{\mathcal{O}}(K\sqrt{T})$ bound. The optimistic rule (8) does not involve any optimization problem solving and can be calculated in an $\mathcal{O}(1)$ cost.

**Overall Algorithm and Guarantees.** We overall procedures in Algorithm 2. For the general multinomial setting, it ensures an $\widetilde{\mathcal{O}}(K\sqrt{T})$ regret guarantee and an $\mathcal{O}(1)$ computation cost.

**Theorem 4.** *Under the same condition as Theorem 3, Algorithm 2 ensures*

$$\text{Reg}_T = \mathcal{O}\left(Kd\log K(\log T)^{\frac{3}{2}}\sqrt{T} + \kappa K^{\frac{3}{2}}d^2(\log K)^2(\log T)^3\right) = \widetilde{\mathcal{O}}(K\sqrt{T}).$$

*The computation cost of Algorithm 2 is bounded by $\mathcal{O}(1)$ for each round $t \in [T]$.*

**Remark 3** (On the $\widetilde{\mathcal{O}}(\sqrt{T/\kappa_*})$ bound)**.** In the binary setting, [9, 10] show that an $\widetilde{\mathcal{O}}(\sqrt{T/\kappa_*})$ minimax optimal is achievable with $\kappa_* = 1/\sigma'(\mathbf{w}_*^\top \mathbf{x}_*)$. However, due to the multinomial behavior of the feedback, it is unclear how to achieve such a rate in MLogB case (see the discussion in Appendix C.5). Besides, it also raises concerns about efficiency when applying the method developed for binary case to multinomial setting. In particular, the $\widetilde{\mathcal{O}}(\sqrt{T/\kappa_*})$ is achieved by the rule $\arg\max_{\mathbf{x}\in\mathcal{X}, \mathbf{w}\in\mathcal{C}_t(\delta)} \sigma(\mathbf{w}^\top\mathbf{x})$ in [9, 10]. The optimization can be efficiently solved in the binary case since one can simply eliminate the non-linearity of the reward function by the relationship $\sigma(z_1) > \sigma(z_2)$ for any $z_1 \in \mathbb{R} > z_2 \in \mathbb{R}$. Such a condition does not hold in MLogB problem.

When reduced to the binary case $K = 1$, our algorithm can also achieves the minimax regret bound.

**Corollary 1.** *When $K = 1$, the multinomial logistic bandit reduces to the binary logistic bandit problem. Then, under the same conditions as Theorem 4, Algorithm 2 with the optimistic rule $\widetilde{r}_t(\mathbf{x}) = \arg\max_{\mathbf{w}\in\mathcal{C}_t^{\texttt{OL}}(\delta)} \mathbf{w}^\top\mathbf{x}$ ensures $\text{Reg}_T \leq \widetilde{\mathcal{O}}(\sqrt{T/\kappa_*})$ with probability at least $1 - \delta$.*

### 3.3 Analysis

This section presents the proof sketch for Theorem 3, which plays an key role in the analysis for Algorithm 2. For notational simplicity, we will drop the superscript OL in this section.

**Leveraging Negative Terms for Efficient Update.** If we update the model with the original loss function by $\overrightarrow{W}_{t+1} = \arg\min_{W \in \mathcal{W}} \ell_t(W) + \frac{1}{2\eta}\|\overrightarrow{W} - \overrightarrow{W}_t\|_{H_t}^2$, the arguments in [10] shows that the estimation error between $\overrightarrow{W}_{t+1}$ and $\overrightarrow{W}_*$ can be bounded by their gap on the loss function

$$\|\overrightarrow{W}_{t+1} - \overrightarrow{W}_*\|_{H_{t+1}}^2 \lesssim \sum_{s=1}^{t} \ell_s(W_*) - \sum_{s=1}^{t} \ell_s(W_{s+1}). \tag{9}$$

However, the update rule with original loss will lead to an $\mathcal{O}(\log t)$ computation cost per iteration. To facilitate a more efficient algorithm, we introduce the update rule with the linearized loss $\langle \nabla\ell_t(\overrightarrow{W}_t), \overrightarrow{W}\rangle$ and the adjusted norm $\widetilde{H}_t$ as (6). As we have shown in the proof of Lemma 12 in Appendix C.1. Denote by $\widetilde{\ell}_t(W) = \langle \nabla\ell_t(\overrightarrow{W}_t), \overrightarrow{W}\rangle + \frac{1}{2}\|\overrightarrow{W} - \overrightarrow{W}_t\|_{\nabla^2\ell_t(\overrightarrow{W}_t)}^2$ the second-order surrogate of $\ell_t(W)$. The efficient update rule will introduce an additional term

$$\sum_{s=1}^{t} \langle \nabla\ell_s(\overrightarrow{W}_{s+1}) - \nabla\widetilde{\ell}_s(\overrightarrow{W}_{s+1}), \overrightarrow{W}_{s+1} - \overrightarrow{W}_*\rangle.$$

We handle the additional term by the self-concordant property of the logistic loss and exploiting a negative term ignored in the previous analysis. Since the logistic loss is a $\sqrt{6}$-self-concordant-like function [19, Lemma 4], then Theorem 3 of [19] indicates that the additional term can be bounded by

$$\sum_{s=1}^{t} \langle \nabla\ell_t(\overrightarrow{W}_{s+1}) - \nabla\widetilde{\ell}_s(\overrightarrow{W}_{s+1}), \overrightarrow{W}_{s+1} - \overrightarrow{W}_*\rangle \leq \sum_{s=1}^{t} \sqrt{6K}S\|\overrightarrow{W}_{s+1} - \overrightarrow{W}_s\|_{\nabla^2\ell_s(\xi_s)}^2,$$

where $\xi_s \in \mathbb{R}^{Kd}$ is on the line connecting $\overrightarrow{W}_{s+1}$ and $\overrightarrow{W}_s$. Besides, by a refined analysis of the online mirror descent (OMD) update (6), we identify an additional *negative term* $-\sum_{s=1}^{t}\|\overrightarrow{W}_{s+1} - \overrightarrow{W}_s\|_{H_s}^2$ on the right hand side of (9). By properly choosing the coefficient $\lambda$, one can cancel the additional term by the negative term and achieves (9) with the efficient update. We note that the negative term in the OMD analysis is also found crucial in the gradient-variation regret of non-stationary online learning [20, 21] as well as its applications to game theory [22] and the SEA model [23].

**Novel construction of the intermediary prediction.** Then, we can further bound the right hand side of (9) by inserting an intermediary loss $\ell_s(\widetilde{W}_s)$ as

$$\sum_{s=1}^{t} \ell_s(W_*) - \sum_{s=1}^{t} \ell_s(W_{s+1}) = \underbrace{\sum_{s=1}^{t} \ell_s(W_*) - \sum_{s=1}^{t} \ell_s(\widetilde{W}_s)}_{\texttt{term (A)}} + \underbrace{\sum_{s=1}^{t} \ell_s(\widetilde{W}_s) - \sum_{s=1}^{t} \ell_s(W_{s+1})}_{\texttt{term (B)}}.$$

In the binary case, inspired by the study [24] for the binary online logistic regression, [10] propose to construct $\widetilde{w}_s = \arg\min_{w \in \mathcal{W}_s} \ell_b(w^\top x_s, +1) + \ell_b(w^\top x_s, 0) + \frac{1}{2\eta}\|w - w_s\|_{H_s}^2$, where $\ell_b$ is the binary logistic loss and $\mathcal{W}_s$ is a shrank decision domain. Then, both term (A) and term (B) can be bounded without the dependence on $\kappa$. However, the analysis for term (B) crucially relies on the condition $|\sigma(\widetilde{w}_s^\top x_s) - y_s| \cdot |\sigma(\widetilde{w}_s^\top x_s) - 1 + y_s| = \sigma'(\widetilde{w}_s^\top x_s)$ to eliminate the dependence on $\kappa$, where $y_s \in \{0, 1\}$ is the one-dimensional feedback. It is hard to show such an relationship in the multinomial case since the feedback $y_s$ has multiple value. A similar challenge is also observed in the recent study on online multiclass logistic regression [25, Appendix F]. One might consider whether it is possible to construct $\widetilde{W}_t$ with the update rule developed in [25]. However, since the online update rule of [25] requires to perform over $\mathbb{R}^{K \times d}$, the learned parameter would become unbounded, which makes it is hard to provide an upper bound for term (A).

To this end, we design a new intermediary term by $\ell_m(\widetilde{z}_s, y_s)$, where $\ell_m$ is the multiclass logistic loss. The prediction is constructed by $\widetilde{z}_s = \sigma^+ (\mathbb{E}_{W \sim P_s}[\sigma(W x_s)])$ with the Gaussian distribution $P_s = \mathcal{N}(\overrightarrow{W}_s, \alpha H_s^{-1})$, where $\sigma^+$ is a pseudo inverse function of the sigmoid function $\sigma$. Such

an integral construction $\widetilde{\mathbf{z}}_s = \boldsymbol{\sigma}^+ (\mathbb{E}_{W \sim P_s}[\boldsymbol{\sigma}(W\mathbf{x}_s)])$ is previously used in the online logistic regression literature [26, 27] but we tailored the tool to our analysis with a different construction of the distribution $P_s$. Lemma 13 and Lemma 14 in Appendix C.1 shows that

$$\texttt{term (A)} \lesssim (\log K + \log t) \log t \quad \text{and} \quad \texttt{term (B)} \lesssim \sum_{s=1}^{t} \|\overrightarrow{W}_s - \overrightarrow{W}_{s+1}\|_{H_s}^2 + Kd \log K \log t,$$

Then, combining the upper bound of term (A) and term (B) and eliminating the additional term $\sum_{s=1}^{t} \|\overrightarrow{W}_s - \overrightarrow{W}_{s+1}\|_{H_s}^2$ with the negative term obtained by a refined analysis of online mirror descent, we complete the proof of Theorem 3.

## 4 More Discussions and Related Work

This section begins with a discussion on the tightness of the proposed bounds.

**On the Tightness of Our Bounds.** In this paper, we introduced OFUL-MLogB, a jointly efficient algorithm that simultaneously achieves regret bounds of $\widetilde{\mathcal{O}}(\sqrt{\kappa K T})$ and $\widetilde{\mathcal{O}}(K\sqrt{T})$.[†] The tightness of these bounds, with respect to $\kappa$ and $T$, was detailed in Remark 3. Regarding the number of feedback values $K$, [11] claimed the optimality of a linear dependence on $K$. However, our findings revealed a nuanced interplay between $\kappa$ and $K$. The $\widetilde{\mathcal{O}}(\sqrt{\kappa K T})$ bound does not conflict with the lower bound argument by [11], given that the non-linear constant $\kappa$ is also associated with $K$. Beyond $\kappa$, the norm of the unknown parameter $S$ and maximum norm $R$ of the reward vector $\boldsymbol{\rho}$ can also depend on $K$ based on the problem's specifics. It is an interesting direction to understand the interrelation of these constants by establishing a lower bound. Additionally, our method has a linear dependence on $d$. In the finite-arm case, an $\mathcal{O}(\sqrt{d})$ dependence might be attainable with a SupLin-type algorithm [28].

Below, we introduce more related works on logistic bandit and the related topics.

**Logistic Bandit.** While the logistic bandit is a specific instance of the generalized linear bandit model [5, 29, 30, 31, 32, 33, 34], the algorithms proposed for GLB tend to exhibit a linear dependence on the nonlinear term $\kappa$, which is exponentially large in the logistic bandit case. Therefore, addressing the non-linearity of the reward function warrants specialized consideration. Besides the UCB-type algorithms [8, 9] mentioned in Section 1, for the $N$-arm case, [35] proposed an experimental design-based algorithm providing an $\widetilde{\mathcal{O}}(\sqrt{d \log NT / \kappa_*})$ regret bound with better dependence on $d$. However, the previous methods were built upon the MLE estimator, whose optimization demands $\mathcal{O}(t)$ computation and storage complexity for the $t$-th iteration. To the best of our knowledge, the only known jointly computational and statistical efficient algorithm was proposed by [10], which achieves a nearly minimax optimal regret bound with computation cost of $\mathcal{O}(\log t)$ per round. In addition to the frequentist bounds, there are also researches on logistic bandit from the Bayesian view. [36] showed the Bayesian regret of the Thompson sampling method is independent of $\kappa$ (even in the lower order term) when the feasible domain is identical to the parameter domain, i.e., $\mathcal{X} = \mathcal{W}$. [37] further proved the $\kappa$-independent bound with weaker conditions.

**Multinomial Logit (MNL) Bandit.** Another relevant line of research is the multinomial logit contextual bandit problem [38, 39, 40, 41, 42], which generalizes the binary logistic bandit by allowing the learner to submit a subset of arms $S_t = \{\mathbf{x}_{t,i}\}_{i=1}^{K} \in \mathcal{X}$ to the users. The expected reward function is also modeled by the multinomial logit model: $\mathbb{E}[r_t | S_t] = \sum_{\mathbf{x}_{t,i} \in S_t} \rho_{t,i} \exp(\mathbf{w}_*^\top \mathbf{x}_{t,i}) / (1 + \sum_{\mathbf{x}_{t,i} \in S_t} \exp(\mathbf{w}_*^\top \mathbf{x}_{t,i}))$, where $\rho_{t,i}$ is the reward for arm $\mathbf{x}_{t,i}$ and $\mathbf{w}_* \in \mathbb{R}^d$ is an unknown parameter. There are also studies on the MNL bandit problem concerning the exponentially large constant $\kappa$. [42] proposed an optimistic algorithm with $\mathcal{O}(d\sqrt{T})$ regret bounds without $\kappa$ in its leading term, which improves the $\mathcal{O}(d\kappa\sqrt{T})$ bound with better dependence on $\kappa$. Considering uniform reward, i.e., $\rho_{t,i} = 1$ for all $t \in [T]$ and $i \in [N]$, [43] further showed an $\widetilde{\mathcal{O}}(d\sqrt{T/\kappa_*})$ bound. To the best of our knowledge, all the existing methods with improved $\kappa$ are established on the MLE estimator. It would be an interesting future direction to develop jointly efficient algorithm for the MNL bandit problem.

---

[†]Theorem 5 in Appendix C.3.2 shows that OFUL-MLogB also attains the $\widetilde{\mathcal{O}}(\sqrt{\kappa K T})$ regret bound.

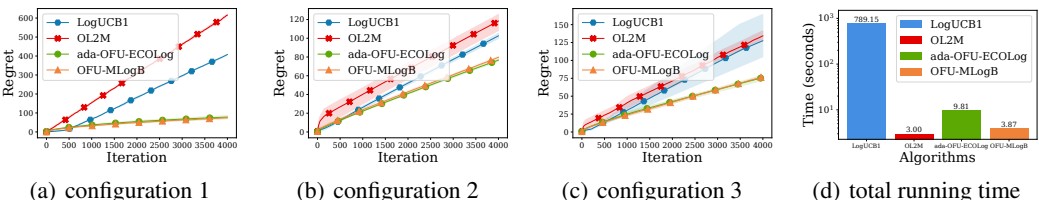

(a) configuration 1      (b) configuration 2      (c) configuration 3      (d) total running time

Figure 1: Performance and computation cost comparison for binary logistic bandit.

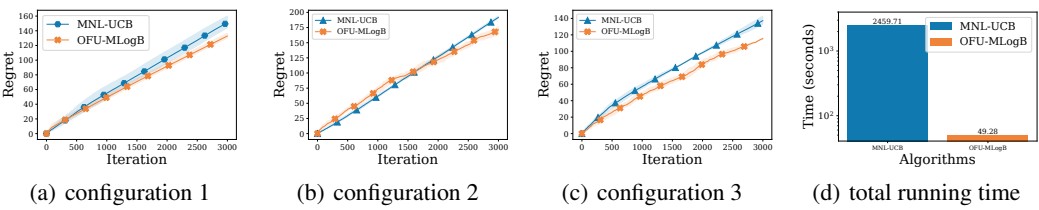

(a) configuration 1      (b) configuration 2      (c) configuration 3      (d) total running time

Figure 2: Performance and computation cost comparison for multinomial logistic bandit.

## 5 Experiments

This section validates the statistical and computation efficiency of the proposed method by experiments. We conduct a bandit learning for $T = 3000$ iterations. In each experiment, we run experiments on 6 random configurations, in which the arm set and the underlying parameter are randomly sampled. Specially, $|\mathcal{X}| = 20$ actions are randomly sampled from a 2-dimensional sphere of radius 1. In the binary case, the norm of the unknown parameter $\mathbf{w}_*$ is set as $S = 5$. In the multinomial case, we set $K = 4$ with $S = 1$. The reward vector is set as $\boldsymbol{\rho} = [0.25, 0.5, 0.75, 1]$. For each configuration, we report the averaged results over 10 trials. More details can be found in Appendix D.

**Experimental Results.** Figure 4 provides a comparison of performance and computation costs in the binary case. The algorithm `O2LM` [7], which has a constant computational cost per iteration, is used as a comparison baseline. Our algorithm demonstrates a time complexity akin to `O2LM`, affirming its computational efficiency. Compared to the state-of-the-art binary logistic bandit algorithm `ada-OFU-ECOLog` [10], our `OFU-MLogB` method has lighter computational overheads while preserving similar empirical performance. Figure 4 illustrates the comparison in the multinomial setting. Our algorithm is around 50 times faster than `MNL-UCB` [11] for running $T = 3000$ iterations and achieves better empirical performance. More experimental results on other configurations and the running time curve that increases along the iterations can be found in Appendix D.

## 6 Conclusion

This paper proposed a jointly efficient algorithm OFUL-MLogB for both binary and multinomial logistic bandit problems with constant computation cost per round and improved regret guarantees. For the multinomial setting, our method improves over the best-known feasible algorithm both on the dependence of $\kappa$ and the computation cost. When reduced to the binary case, OFUL-MLogB also contributes to improve the computation cost of previous method from $\mathcal{O}(\log t)$ to $\mathcal{O}(1)$ per round while still preserving the $\widetilde{\mathcal{O}}(\sqrt{T/\kappa_*})$ minimax optimal bound up to logarithmic factors. A promising future direction is to consider the multinomial logit model in reinforcement learning. Besides, it is still open on how to achieve the $\widetilde{\mathcal{O}}(\sqrt{T/\kappa_*})$ bound for the multinomial case.

## Acknowledgments

YJZ and MS were supported by the Institute for AI and Beyond, UTokyo. YJZ was also supported by Todai Fellowship. The authors would thank Peng Zhao for helpful discussions and Jing Wang for the assistance in experiments.

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

# A Properties of the Logistic Regression

This section collects several key properties of the logistic loss used throughout the paper.

## A.1 Multiclass Logistic Loss

We have the following property of the multiclass logistic loss.

**Property 1.** *Let $y \in \mathcal{Y}$ with $\mathcal{Y} = \{0\} \cup [K]$ and $\ell : \mathbb{R}^K \times \mathcal{Y} \mapsto \mathbb{R}$ be the multiclass logistic loss defined by*

$$\ell(\mathbf{z}, y) = \sum_{k=0}^{K} \mathbb{1}\{y = k\} \cdot \log\left(\frac{1}{[\boldsymbol{\sigma}(\mathbf{z})]_k}\right), \tag{10}$$

*where $\boldsymbol{\sigma} : \mathbb{R}^K \to [0, 1]^K$ is the vector valued link function defined by $[\boldsymbol{\sigma}(\mathbf{z})]_k = \exp(z_k)/(1 + \sum_{j=1}^{K} \exp(z_j))$ and $[\boldsymbol{\sigma}(\mathbf{z})]_0 = \exp(z_k)/(1 + \sum_{j=1}^{K} \exp(z_j))$. Let $\mathbf{y} = [\mathbb{1}\{y = 1\}, \ldots, \mathbb{1}\{y = K\}] \in \mathbb{R}^K$ and $\sigma_k(\mathbf{z}) = [\boldsymbol{\sigma}(\mathbf{z})]_k$ be the $k$-th entry of $\boldsymbol{\sigma}(\mathbf{z})$. Then the first, second and third derivations w.r.t. the first argument of the loss function can be written as*

$$\nabla_z \ell(\mathbf{z}, y) = \boldsymbol{\sigma}(\mathbf{z}) - \mathbf{y},$$

$$\nabla_z^2 \ell(\mathbf{z}, y) = \mathrm{diag}(\boldsymbol{\sigma}(\mathbf{z})) - \boldsymbol{\sigma}(\mathbf{z})\boldsymbol{\sigma}(\mathbf{z})^\top,$$

$$\nabla_z^3 \ell(\mathbf{z}, y)[\mathbf{u}] = \sum_{k=1}^{K} u_k \sigma_k(\mathbf{z}) \left( 2\boldsymbol{\sigma}(\mathbf{z})\boldsymbol{\sigma}(\mathbf{z})^\top - \mathrm{diag}(\boldsymbol{\sigma}(\mathbf{z})) - \mathbf{e}_k \boldsymbol{\sigma}(\mathbf{z})^\top - \boldsymbol{\sigma}(\mathbf{z})\mathbf{e}_k^\top + E_k \right),$$

*where $u_k$ is the $k$-th entry of the vector $\mathbf{u} \in \mathbb{R}^K$ and $E_k \in \mathbb{R}^{K \times K}$ is a "one-hot" matrix whose $(k, k)$-th entry equals to 1 while others equaling to 0. Besides, $\mathbf{e}_k$ is the one-hot vector whose $k$-th entry equals to 1. In the above, $\nabla_z^3 \ell(\mathbf{z}, y)[\mathbf{u}] = \lim_{t \to 0} t^{-1} \left( \nabla_z^2 \ell(\mathbf{z} + t\mathbf{u}, y) - \nabla_z^2 \ell(\mathbf{z}, y) \right)$.*

When we train a linear model with the logistic regression loss, we can calculate its gradient and Hessian as follows.

**Property 2.** *Denote by the logistic loss defined over the data point $(\mathbf{x}_t, y_t) \in \mathcal{X} \times \mathcal{Y}$ by*

$$\ell_t(W) = \ell(W\mathbf{x}, y) = \sum_{k=0}^{K} \mathbb{1}\{y_t = k\} \cdot \log\left(1/[\boldsymbol{\sigma}(W\mathbf{x}_t)]_k\right). \tag{11}$$

*Then, the gradient $\nabla \ell_t(\overrightarrow{W}) \in \mathbb{R}^{Kd}$ and the Hessian $\nabla^2 \ell_t(W) \in \mathbb{R}^{Kd \times Kd}$ of the loss function with respective to the vectorized model $\overrightarrow{W}$ is given by*

$$\nabla \ell_t(\overrightarrow{W}) = (\boldsymbol{\sigma}(W\mathbf{x}_t) - \mathbf{y}_t) \otimes \mathbf{x}_t \quad \text{and} \quad \nabla^2 \ell_t(W) = \mathrm{diag}(\boldsymbol{\sigma}(W\mathbf{x}_t)) - \boldsymbol{\sigma}(W\mathbf{x}_t)\boldsymbol{\sigma}(W\mathbf{x}_t)^\top \otimes \mathbf{x}_t \mathbf{x}_t^\top.$$

Besides, we can obtain the gradient of the vector-valued link function as follows.

**Property 3.** *Let the sigmoid function $\boldsymbol{\sigma} : \mathbb{R}^K \mapsto [0, 1]^K$ be defined as (1) and $\sigma_k(\mathbf{z})$ denote the $k$-th entry of $\boldsymbol{\sigma}(\mathbf{z})$. We have*

$$\nabla \sigma_k(\mathbf{z}) = \sigma_k(\mathbf{z}) \cdot (\mathbf{e}_k - \boldsymbol{\sigma}(\mathbf{z}));$$

$$\nabla^2 \sigma_k(\mathbf{z}) = \sigma_k(\mathbf{z}) \left( 2\boldsymbol{\sigma}(\mathbf{z})\boldsymbol{\sigma}(\mathbf{z})^\top - \mathrm{diag}(\boldsymbol{\sigma}(\mathbf{z})) - \mathbf{e}_k \boldsymbol{\sigma}(\mathbf{z})^\top - \boldsymbol{\sigma}(\mathbf{z})\mathbf{e}_k^\top + E_k \right),$$

*In above, $E_k \in \mathbb{R}^{K \times K}$ is an all zero matrix except that its $(k, k)$-th entry is 1. For the first order derivation, we can write it into a more concise formulation by the notation $\nabla \boldsymbol{\sigma}(\mathbf{z}) : \mathbb{R}^K \mapsto \mathbb{R}^{K \times K}$*

$$\nabla \boldsymbol{\sigma}(\mathbf{z}) \triangleq \frac{\partial \boldsymbol{\sigma}(\mathbf{z})}{\partial \mathbf{z}^\top} = \mathrm{diag}(\boldsymbol{\sigma}(\mathbf{z})) - \boldsymbol{\sigma}(\mathbf{z})\boldsymbol{\sigma}(\mathbf{z})^\top.$$

We have the following lemma for the logistic loss function.

**Lemma 1.** *Let $C > 0$, $\mathbf{a} \in [-C, C]^K$, $y \in \{0\} \cup [K]$ and $\mathbf{b} \in \mathbb{R}^K$. Then, we have*

$$\ell(\mathbf{a}, y) \geq \ell(\mathbf{b}, y) + \nabla \ell(\mathbf{b}, y)^\top (\mathbf{a} - \mathbf{b}) + \frac{1}{\log(K + 1) + 2(C + 1)} (\mathbf{a} - \mathbf{b})^\top \nabla^2 \ell(\mathbf{b}, y)(\mathbf{a} - \mathbf{b}).$$

*Proof.* Lemma 1 is essentially the Lemma 4 of [27] with a slightly different definition of the logistic loss. The $K+1$-class logistic loss used by [27] is defined as $\widetilde{\ell}(\widetilde{\mathbf{z}}, y) = \sum_{k=0}^{K} \mathbb{1}\{y = k\} \log\left(\sum_{k=0}^{K} \exp([\widetilde{\mathbf{z}}]_j)/\exp([\widetilde{\mathbf{z}}]_k)\right)$, where $\widetilde{\mathbf{z}} \in \mathbb{R}^{K+1}$ is a $K+1$-dimensional input. One can connect the two logistic losses by $\widetilde{\ell}([\mathbf{z};0], y) = \ell(\mathbf{z}, y)$. Then, we can prove Lemma 1 by directly applying Lemma 4 of [27] with the augmented vectors $\widetilde{\mathbf{a}} = [\mathbf{a};0]$ and $\widetilde{\mathbf{b}} = [\mathbf{b};0]$. $\qquad\square$

## A.2 Self-concordant-like Function

Then, we introduce the notion of self-concordant-like function, which generalizes the standard self-concordant notion. The self-concordant function is first introduced by [44] in the study of inner-point algorithm and the notion is further generalized by the following work [45, 19, 46] for analyzing the Newton-step methods. The self-concordant function enjoys several nice properties, which plays an important role in our local analysis. A comprehensive study for self-concordance function is provided by [46]. Here, we present the definition and necessary lemmas for our analysis.

**Definition 1** (Theorem 3 of [19]). *A convex function $\ell \in \mathcal{C}^3 : \mathbb{R}^d \to \mathbb{R}$ is $M$-self-concordant-like if and only if for any $\mathbf{x}, \mathbf{u}_1, \mathbf{u}_2, \mathbf{u}_3 \in \mathbb{R}^d$, we have*

$$\left|\langle D^3\ell(\mathbf{z})[\mathbf{u}_1]\mathbf{u}_2, \mathbf{u}_3\rangle\right| \leq M\|\mathbf{u}_1\|_2\|\mathbf{u}_2\|_{\mathbf{z}}\|\mathbf{u}_3\|_{\mathbf{z}},$$

*where $\|\mathbf{u}\|_{\mathbf{z}} := \sqrt{\mathbf{u}^\top \nabla^2 f(\mathbf{z})\mathbf{u}}$ is the local norm over the Hessian $\nabla^2\ell(\mathbf{z})$.*

The logistic loss is $\sqrt{6}$-self-concordant-like function.

**Lemma 2** (Lemma 4 of [19]). *The logistic loss defined as* (11) *is $\sqrt{6}\|\mathbf{x}_t\|_2$-self-concordant-like.*

The self-concordant function enjoys favorable properties, which help us to conduct local analysis for the logistic bandit problem. We list useful lemmas as follows.

**Lemma 3** (Bound of Hessian Map, Theorem 4(b) of [19]). *Let $\ell : \mathbb{R}^d \mapsto \mathbb{R}$ be an $M$-self-concordant-like function. Then, for any $\mathbf{z}_1, \mathbf{z}_2 \in \mathrm{dom}(\ell)$, we have*

$$e^{-M\|\mathbf{z}_1-\mathbf{z}_2\|_2}\nabla^2\ell(\mathbf{z}_1) \preccurlyeq \nabla^2\ell(\mathbf{z}_2) \preccurlyeq e^{M\|\mathbf{z}_1-\mathbf{z}_2\|_2}\nabla^2\ell(\mathbf{z}_1) \tag{12}$$

The following lemma is a consequence of the bound of the Hessian map.

**Lemma 4** (Corollary 2 of [46]). *Let $\ell : \mathbb{R}^d \mapsto \mathbb{R}$ be an $M$-self-concordant-like function. Then, for any $\mathbf{z}_1, \mathbf{z}_2 \in \mathrm{dom}(\ell)$, we have*

$$\frac{1 - e^{-M\|\mathbf{z}_1-\mathbf{z}_2\|_2}}{M\|\mathbf{z}_1-\mathbf{z}_2\|_2} \cdot \nabla^2\ell(\mathbf{z}_2) \preccurlyeq \int_0^1 \nabla^2\ell(\mathbf{z}_1 + \nu(\mathbf{z}_2 - \mathbf{z}_1))\mathrm{d}\nu \preccurlyeq \frac{e^{M\|\mathbf{z}_1-\mathbf{z}_2\|_2} - 1}{M\|\mathbf{z}_1-\mathbf{z}_2\|_2}\nabla^2\ell(\mathbf{z}_1).$$

We can also have the bound on the function value of the self-concordant functions.

**Lemma 5** (Proposition 10 of [46]). *Let $\ell : \mathbb{R}^d \mapsto \mathbb{R}$ be an $M$-self-concordant-like function. Then, for any $\mathbf{z}_1, \mathbf{z}_2 \in \mathrm{dom}(\ell)$, we have*

$$c(-M\|\mathbf{z}_1 - \mathbf{z}_2\|_2) \cdot \|\mathbf{z}_2 - \mathbf{z}_1\|_{\mathbf{z}_1}^2 \leq \ell(\mathbf{z}_2) - \ell(\mathbf{z}_1) - \nabla\ell(\mathbf{z}_1)^\top(\mathbf{z}_2 - \mathbf{z}_1) \leq c(M_f\|\mathbf{z}_1 - \mathbf{z}_2\|_2) \cdot \|\mathbf{z}_2 - \mathbf{z}_1\|_{\mathbf{z}_1}^2,$$

*where $c : \mathbb{R} \mapsto \mathbb{R}$ is the coefficient function defined as $c(x) = (e^x - x - 1)/x^2$.*

# B Omitted Proofs for Section 2.3

## B.1 Proof of Theorem 1

This section we present the proof of Theorem 1.

### B.1.1 Main Proof

Before introducing the main proof, we introduce a key lemma on an improved self-normalized martingale tail inequality for the multinomial noise, which helps to save an $\mathcal{O}(\sqrt{K})$ factor compared to the one obtained by [11]. The proof of Lemma 6 is provided in Appendix B.1.2

**Lemma 6.** *Let $\{\mathcal{F}_t\}_{t=1}^{\infty}$ be a filtration. Let $\{\mathbf{x}_t\}_{t=1}^{\infty}$ be a stochastic process in $\mathcal{B}_2(d) \triangleq \{\mathbf{x} \in \mathbb{R}^d \mid \|\mathbf{x}_t\|_2 \leq 1\}$ such that $\mathbf{x}_t$ is $\mathcal{F}_t$ measurable. Let $\{\varepsilon_t\}_{t=1}^{\infty}$ be a martingale difference sequence such that $\varepsilon_t \in \mathbb{R}^K$ is $\mathcal{F}_{t+1}$ measurable. Furthermore, assume that conditional on $\mathcal{F}_t$, we have $\|\varepsilon_t\|_1 \leq 2$ almost surely, and denoted by $\Sigma_t \triangleq \mathbb{E}[\varepsilon_t \varepsilon_t^\top \mid \mathcal{F}_t]$. Let $\lambda > 0$ and for any $t \geq 1$ define*

$$S_t = \sum_{s=1}^{t-1} \varepsilon_s \otimes \mathbf{x}_s \ \text{ and } \ H_t = \lambda I_{Kd} + \sum_{s=1}^{t-1} \Sigma_s \otimes \mathbf{x}_s \mathbf{x}_s^\top.$$

*Then for any $\delta \in (0, 1]$, we have*

$$\Pr\left[\exists t > 1, \|S_t\|_{H_t^{-1}} \geq \frac{\sqrt{\lambda}}{4} + \frac{4}{\sqrt{\lambda}} \log\left(\frac{|H_t|^{1/2}}{\delta \lambda^{Kd/2}}\right) + \frac{4}{\sqrt{\lambda}} Kd \log(2)\right] \leq \delta.$$

Lemma 6 is a counterpart of the variance-aware self-normalized tail inequality [8, Theorem 1] for the multinominal case. In comparison with the previously established result by [11], our inequality more effectively leverages the geometric structure of the noise in the multinomial logistic bandit problem, as indicated by the condition $\|\varepsilon_t\|_1 \leq 2$. The previous paper's condition, $\|\varepsilon_t\|_2 \leq \sqrt{K}$, is relatively loose, ultimately resulting in an additional $O(\sqrt{K})$ term in the concentration inequality. Then, we introduce the main proof of Theorem 1

*Proof of Theorem 1.* The optimality of the MLE estimator indicates that $\mathbf{g}_t(\overrightarrow{W}_t^{\mathrm{MLE}}) = \nabla \mathcal{L}_t(\overrightarrow{W}_t^{\mathrm{MLE}}) = 0$. In such a case, we have

$$\mathbf{g}_t(W_t^{\mathrm{MLE}}) - \mathbf{g}_t(W_*) = -\mathbf{g}_t(W_*) = \sum_{s=1}^{t-1}(\boldsymbol{\sigma}(W_* \mathbf{x}_s) - \mathbf{y}_s) \otimes \mathbf{x}_s - \lambda \overrightarrow{W}_* = \sum_{s=1}^{t-1} \varepsilon_s \otimes \mathbf{x}_s - \lambda \overrightarrow{W}_*,$$

where $\varepsilon_s = \boldsymbol{\sigma}(W_* \mathbf{x}_s) - \mathbf{y}_s$ is the noise with bounded $L_\infty$ norm as we have $\|\epsilon_s\|_\infty \leq \|\boldsymbol{\sigma}(W_* \mathbf{x}_s)\|_\infty + \|\mathbf{y}_s\|_\infty \leq 2$. Then, we can bound the gap between $\overrightarrow{W}_t^{\mathrm{MLE}}$ and $\overrightarrow{W}_*$ by

$$\|\mathbf{g}_t(W_t^{\mathrm{MLE}}) - \mathbf{g}_t(W_*)\|_{H_t^{-1}(W_*)} \leq \left\|\sum_{s=1}^{t-1} \varepsilon_s \otimes \mathbf{x}_s\right\|_{H_t^{-1}(W_*)} + \|\lambda \overrightarrow{W}_*\|_{H_t^{-1}(W_*)}$$

$$\leq \left\|\sum_{s=1}^{t-1} \varepsilon_s \otimes \mathbf{x}_s\right\|_{H_t^{-1}(W_*)} + \sqrt{\lambda} S$$

$$\leq \frac{\sqrt{\lambda}}{4} + \frac{4}{\sqrt{\lambda}} \log\left(\frac{|H_t|^{1/2}}{\delta \lambda^{Kd/2}}\right) + \frac{4}{\sqrt{\lambda}} Kd \log(2) + \sqrt{\lambda} S, \quad (13)$$

with probability at least $1 - \delta$ for all $t \geq 1$. In above, the matrix $H_t(W_*)$ is defined as $H_t(W_*) = \lambda I_{Kd} + \sum_{s=1}^{t-1} \Sigma(W_* \mathbf{x}_s) \mathbf{x}_s \mathbf{x}_s^\top$. The first inequality is due to the relationship $H_t^{-1}(W_*) \preccurlyeq 1/\lambda \cdot I_{Kd}$. The last inequality is by Lemma 6 since

$$\Sigma(W_* \mathbf{x}_s) = \mathrm{diag}(\boldsymbol{\sigma}(W_* \mathbf{x}_s)) - \boldsymbol{\sigma}(W_* \mathbf{x}_s)\boldsymbol{\sigma}(W_* \mathbf{x}_s)^\top = \mathbb{E}[\varepsilon_s \varepsilon_s^\top \mid \mathcal{F}_{s-1}].$$

Then, we can bound the determinate of $H_t(W_*)$ by

$$|H_t| \leq \left(\frac{\mathrm{Tr}(H_t)}{Kd}\right)^{Kd} \leq \left(\frac{\lambda Kd + \sum_{s=1}^{t-1} \sum_{k=1}^{K} \Sigma_k(W_* \mathbf{x}_s)\|\mathbf{x}_s\|_2}{Kd}\right)^{Kd} \leq (\lambda + t/d)^{Kd}. \quad (14)$$

Plugging (14) into (13), we have

$$\|\mathbf{g}_t(W_t^{\mathrm{MLE}}) - \mathbf{g}_t(W_*)\|_{H_t^{-1}(W_*)} \leq \left(\frac{1}{4} + S\right)\sqrt{\lambda} + \frac{2Kd}{\sqrt{\lambda}} \log\left(1 + \frac{t}{\lambda d}\right) + \frac{4Kd}{\sqrt{\lambda}} \log\left(\frac{2}{\delta}\right), \quad (15)$$

By setting

$$\lambda = \frac{4Kd \log(2(1 + t/d)/\delta)}{S + 1/4} = \mathcal{O}(dK \log(t/\delta)), \quad (16)$$

we have

$$\|\mathbf{g}_t(W_t^{\mathrm{MLE}}) - \mathbf{g}_t(W_*)\|_{H_t^{-1}(W_*)} \leq beta_t(\delta) \triangleq 4\sqrt{Kd(1 + S) \log(2(1 + t/d)/\delta)},$$

which completes the proof . $\qquad\square$

### B.1.2 Useful Lemmas

**Lemma 6.** *Let $\{\mathcal{F}_t\}_{t=1}^{\infty}$ be a filtration. Let $\{\mathbf{x}_t\}_{t=1}^{\infty}$ be a stochastic process in $\mathcal{B}_2(d) \triangleq \{\mathbf{x} \in \mathbb{R}^d \mid \|\mathbf{x}_t\|_2 \leq 1\}$ such that $\mathbf{x}_t$ is $\mathcal{F}_t$ measurable. Let $\{\varepsilon_t\}_{t=1}^{\infty}$ be a martingale difference sequence such that $\varepsilon_t \in \mathbb{R}^K$ is $\mathcal{F}_{t+1}$ measurable. Furthermore, assume that conditional on $\mathcal{F}_t$, we have $\|\varepsilon_t\|_1 \leq 2$ almost surely, and denoted by $\Sigma_t \triangleq \mathbb{E}[\varepsilon_t \varepsilon_t^{\top} \mid \mathcal{F}_t]$. Let $\lambda > 0$ and for any $t \geq 1$ define*

$$S_t = \sum_{s=1}^{t-1} \varepsilon_s \otimes \mathbf{x}_s \ \text{ and } \ H_t = \lambda I_{Kd} + \sum_{s=1}^{t-1} \Sigma_s \otimes \mathbf{x}_s \mathbf{x}_s^{\top}.$$

*Then for any $\delta \in (0, 1]$, we have*

$$\Pr\left[\exists t > 1, \|S_t\|_{H_t^{-1}} \geq \frac{\sqrt{\lambda}}{4} + \frac{4}{\sqrt{\lambda}} \log\left(\frac{|H_t|^{1/2}}{\delta \lambda^{Kd/2}}\right) + \frac{4}{\sqrt{\lambda}} Kd \log(2)\right] \leq \delta.$$

*Proof of Lemma 6.* The proof follows the pipeline in the analysis of the self-normalized tail-inequality in the previous studies [3, 8, 11]. Letting $\bar{H}_t = \sum_{s=1}^{t-1} \Sigma_s \otimes \mathbf{x}_s \mathbf{x}_s^{\top}$, we define the function

$$M_t(\boldsymbol{\xi}) \triangleq \exp(\boldsymbol{\xi}^{\top} S_t - \|\boldsymbol{\xi}\|_{\bar{H}_t}^2),$$

for any $t \geq 1$ and $\boldsymbol{\xi} \in \mathbb{R}^{Kd}$. For $t = 0$, let $M_0(\boldsymbol{\xi}) = 0$.

By Lemma 7 in Appendix B.1.2, we can show that $\{M_t(\boldsymbol{\xi})\}_{t=1}^{\infty}$ is a non-negative super-martingale for any $\boldsymbol{\xi} \in \frac{1}{2}\mathcal{B}_2(Kd)$. We note that our Lemma 7 is an imporved version of Lemma 7 in [11], which relaxs restriction on the feasible domain of $\boldsymbol{\xi}$ from $\boldsymbol{\xi} \in \frac{1}{\sqrt{K}}\mathcal{B}_2(Kd)$ to $\boldsymbol{\xi} \in \frac{1}{2}\mathcal{B}_2(Kd)$, resulting in saving an $\mathcal{O}(\sqrt{K})$ factor in the bound.

Then, let $h(\boldsymbol{\xi})$ be a probability density with support on $\frac{1}{2}\mathcal{B}_2(Kd)$ and define

$$\bar{M}_t \triangleq \int_{\boldsymbol{\xi}} M_t(\boldsymbol{\xi}) \mathrm{d}h(\boldsymbol{\xi}) = \int_{\boldsymbol{\xi}} \exp(\boldsymbol{\xi}^{\top} S_t - \|\boldsymbol{\xi}\|_{\bar{H}_t}^2) \mathrm{d}h(\boldsymbol{\xi})$$

for all $t \geq 1$. Lemma 20.3 of [15] shows that $\bar{M}_t$ is also a non-negative super-martingale and $\mathbb{E}[\bar{M}_0] = 1$. Then, the maximal inequality (Theorem 3.9 of [15]) shows that

$$\Pr\left[\sup_{t \in \mathbb{N}} \log(\bar{M}_t) \geq \log\left(\frac{1}{\delta}\right)\right] = \Pr\left[\sup_{t \in \mathbb{N}} \bar{M}_t \geq \frac{1}{\delta}\right] \leq \delta. \tag{17}$$

Next, we turn to exam the formulation of $\bar{M}_t$ and subsequently establish a connection with the term we aim to bound, $\|S_t\|_{H_t^{-1}}$. Let $h(\boldsymbol{\xi})$ be the density of an isotropic normal distribution with precision matrix $2\lambda I_{Kd}$ truncated on $\frac{1}{2}\mathcal{B}_2(Kd)$ and $N(h)$ be its normalization constant. Furthermore, let $g(\boldsymbol{\xi})$ be the density of the normal distribution with precision matrix $2H_t$ that is truncated on the ball $\frac{1}{4}\mathcal{B}_2(Kd)$. Following the arguments in the proof of [8, Theorem 1] (more precisely the arguments in deriving Eq.(11) in [8]), for any $t \geq 1$, one can show that

$$\bar{M}_t \geq \exp(\boldsymbol{\xi}^{\top} S_t - \|\boldsymbol{\xi}\|_{H_t}^2) \cdot \frac{N(g)}{N(h)}.$$

for any $\boldsymbol{\xi} \in \frac{1}{4}\mathcal{B}_2(Kd)$. Let $\boldsymbol{\xi}_0 \triangleq \frac{H_t^{-1} S_t}{\|S_t\|_{H_t^{-1}}} \cdot \frac{\sqrt{\lambda}}{4}$. One can check $\|\boldsymbol{\xi}_0\|_2 \leq \frac{1}{\sqrt{\lambda}} \cdot \frac{\sqrt{\lambda}}{4} \leq 1/4$. Then, we can further have

$$\log(\bar{M}_t) \geq \boldsymbol{\xi}_0^{\top} S_t - \|\boldsymbol{\xi}_0\|_{H_t}^2 + \log\left(\frac{N(g)}{N(h)}\right) = \frac{\sqrt{\lambda}}{4} \|S_t\|_{H_t^{-1}} - \frac{\lambda}{16} + \log\left(\frac{N(g)}{N(h)}\right). \tag{18}$$

Combining (17) and (18), for any $t \geq 1$, we have

$$\Pr\left[\|S_t\|_{H_t^{-1}} \leq \frac{\sqrt{\lambda}}{4} + \frac{4}{\sqrt{\lambda}} \log\left(\frac{N(h)}{\delta N(g)}\right)\right] \geq 1 - \delta.$$

We complete the proof with Lemma 6 of [8] such that

$$\log\left(\frac{N(h)}{N(g)}\right) \leq \log\left(\frac{|H_t|^{1/2}}{\lambda^{Kd/2}}\right) + Kd \log(2).$$

$\square$

**Lemma 7.** *For all $\boldsymbol{\xi} \in \frac{1}{2}\mathcal{B}_2(Kd)$, the sequence $\{M_t(\boldsymbol{\xi})\}_{t=1}^{\infty}$ is a non-negative super-martingale.*

*Proof of Lemma 7.* To show that $\{M_t(\boldsymbol{\xi})\}_{t=1}^{\infty}$ is a non-negative super-martingale, it is sufficient to prove $\mathbb{E}[M_{t+1}(\boldsymbol{\xi}) \mid \mathcal{F}_t] \leq M_t(\boldsymbol{\xi})$ for any $t > 1$ and $\boldsymbol{\xi} \in \frac{1}{2}\mathcal{B}_2(Kd)$. By the definition of $M_{t+1}(\boldsymbol{\xi})$, we have

$$\mathbb{E}[M_{t+1}(\boldsymbol{\xi}) \mid \mathcal{F}_t] = \mathbb{E}\left[\exp(\boldsymbol{\xi}^\top S_{t+1} - \|\boldsymbol{\xi}\|_{\bar{H}_{t+1}}^2) \mid \mathcal{F}_t\right]$$

$$= \mathbb{E}\left[\exp(\boldsymbol{\xi}^\top(\boldsymbol{\varepsilon}_t \otimes \mathbf{x}_t) - \boldsymbol{\xi}^\top(\Sigma_t \otimes \mathbf{x}_t\mathbf{x}_t^\top)\boldsymbol{\xi}) \mid \mathcal{F}_t\right] M_t(\boldsymbol{\xi})$$

$$= \mathbb{E}\left[\exp(\boldsymbol{\varepsilon}_t^\top(I_K \otimes \mathbf{x}_t^\top)\boldsymbol{\xi} - \boldsymbol{\xi}^\top(\Sigma_t \otimes \mathbf{x}_t\mathbf{x}_t^\top)\boldsymbol{\xi}) \mid \mathcal{F}_t\right] M_t(\boldsymbol{\xi}). \qquad (19)$$

Let $\boldsymbol{\xi} = [\boldsymbol{\xi}_1; \ldots; \boldsymbol{\xi}_K]$ and $\boldsymbol{\xi}_k \in \mathbb{R}^d$ is the vector containing $(k-1)d+1$-th to $kd$-th elements of $\boldsymbol{\xi}$. We can further check that

$$|\boldsymbol{\varepsilon}_t^\top(I_K \otimes \mathbf{x}_t^\top)\boldsymbol{\xi}| \leq \|\boldsymbol{\varepsilon}_t\|_1 \cdot \|(I_K \otimes \mathbf{x}_t^\top)\boldsymbol{\xi}\|_\infty = \|\boldsymbol{\varepsilon}_t\|_1 \cdot \max_{k \in K}|\boldsymbol{\xi}_k^\top \mathbf{x}_t| \leq 1,$$

where the first inequality is due to the Höler's inequality. The last inequality is by the condition $\|\boldsymbol{\varepsilon}_t\|_1 \leq 2$ and $\|\boldsymbol{\xi}_k\|_2 \leq \|\boldsymbol{\xi}\|_2 \leq 1/2$ for any $k \in [K]$. Then, we can further bound the expectation term in (19) by

$$\mathbb{E}\left[\exp(\boldsymbol{\varepsilon}_t^\top(I_K \otimes \mathbf{x}_t^\top)\boldsymbol{\xi} - \boldsymbol{\xi}^\top(\Sigma_t \otimes \mathbf{x}_t\mathbf{x}_t^\top)\boldsymbol{\xi}) \mid \mathcal{F}_t\right]$$

$$= \mathbb{E}\left[\exp(\boldsymbol{\varepsilon}_t^\top(I_K \otimes \mathbf{x}_t^\top)\boldsymbol{\xi}) \mid \mathcal{F}_t\right] \cdot \exp(-\boldsymbol{\xi}^\top(\Sigma_t \otimes \mathbf{x}_t\mathbf{x}_t^\top)\boldsymbol{\xi})$$

$$\leq \exp(\boldsymbol{\xi}^\top(I_K \otimes \mathbf{x}_t)\Sigma_t(I_K \otimes \mathbf{x}_t^\top)\boldsymbol{\xi}) \cdot \exp(-\boldsymbol{\xi}^\top(\Sigma_t \otimes \mathbf{x}_t\mathbf{x}_t^\top)\boldsymbol{\xi})$$

$$= \exp(\boldsymbol{\xi}^\top(\Sigma_t \otimes \mathbf{x}_t\mathbf{x}_t^\top)\boldsymbol{\xi}) \cdot \exp(-\boldsymbol{\xi}^\top(\Sigma_t \otimes \mathbf{x}_t\mathbf{x}_t^\top)\boldsymbol{\xi}) = 1,$$

where the first inequality is due to Lemma 8 and the last equality is due to mixed-product property of the Kronecker product. Therefore, we can show $\mathbb{E}[M_{t+1}(\boldsymbol{\xi}) \mid \mathcal{F}_t] \leq M_t(\boldsymbol{\xi})$ and complete the proof.

$\square$

**Lemma 8** (Lemma 6 of [11]). *Let $\boldsymbol{\varepsilon} \in \mathbb{R}^K$ be a zero-mean random vector with covariance matrix $\Sigma$. Then, for any vector $\mathbf{a} \in \mathbb{R}^K$ such that $|\boldsymbol{\varepsilon}^\top \mathbf{a}| \leq 1$, we have $\mathbb{E}[\exp(\boldsymbol{\varepsilon}^\top \mathbf{a})] \leq \exp(\mathbf{a}^\top \Sigma \mathbf{a})$.*

## B.2 Proof of Theorem 2

This section provides the proof for Theorem 2.

### B.2.1 Main Proof

*Proof of Theorem 2.* We prove the proposed algorithm can achieve $\mathcal{O}(d \log T \sqrt{\kappa KT})$ and $\mathcal{O}(dK \log T \sqrt{T} + \kappa d^2 K^2 \log^2 T)$ bounds simultaneously. Before presenting the proofs, we introduce two matrix that measures the local curvature of the loss function and will be used in the rest part of the proof:

$$A(\mathbf{x}, W_1, W_2) = \int_{v=0}^{1} \nabla \boldsymbol{\sigma}(vW_1\mathbf{x} + (1-v)W_2\mathbf{x})\mathrm{d}v; \qquad (20)$$

$$G_t(W_1, W_2) = \lambda I_{Kd} + \sum_{s=1}^{t-1} A(\mathbf{x}_s, W_1, W_2) \otimes \mathbf{x}_s\mathbf{x}_s^\top. \qquad (21)$$

**Analysis for the $\mathcal{O}(d \log T \sqrt{\kappa KT})$ Bound.** The regret can be bounded by

$$\mathrm{Reg}_T = \sum_{t=1}^{T} \boldsymbol{\rho}^\top \boldsymbol{\sigma}(W_*\mathbf{x}_*) - \sum_{t=1}^{T} \boldsymbol{\rho}^\top \boldsymbol{\sigma}(W_*\mathbf{x}_t)$$

$$\leq \sum_{t=1}^{T} \boldsymbol{\rho}^\top \boldsymbol{\sigma}(W_t^{\mathtt{OPT}}\mathbf{x}_t) - \sum_{t=1}^{T} \boldsymbol{\rho}^\top \boldsymbol{\sigma}(W_*\mathbf{x}_t)$$

$$= \sum_{t=1}^{T} \boldsymbol{\rho}^\top A(\mathbf{x}_t, W_*, W_t^{\mathtt{OPT}})(W_t^{\mathtt{OPT}} - W_*)\mathbf{x}_t$$

$$\leq \sum_{t=1}^{T} \|\boldsymbol{\rho}\|_{A(\mathbf{x}_t, W_*, W_t^{\mathtt{OPT}})} \cdot \|(W_t^{\mathtt{OPT}} - W_*)\mathbf{x}_t\|_{A(\mathbf{x}_t, W_*, W_t^{\mathtt{OPT}})}$$

$$\leq \sum_{t=1}^{T} \|\boldsymbol{\rho}\|_2 \cdot \|(W_t^{\mathtt{OPT}} - W_*)\mathbf{x}_t\|_{A(\mathbf{x}_t, W_*, W_t^{\mathtt{OPT}})}$$

$$\leq R \sum_{t=1}^{T} \|(W_t^{\mathtt{OPT}} - W_*)\mathbf{x}_t\|_{A(\mathbf{x}_t, W_*, W_t^{\mathtt{OPT}})}$$

In the above, the term $A(\mathbf{x}_t, W_*, W_t^{\mathtt{OPT}})$ is defined as (20) and the first inequality is a consequence of the optimistic rule (5) such that $(\mathbf{x}_t, W_t^{\mathtt{OPT}}) = \arg\max_{\mathbf{x} \in \mathcal{X}, W \in \mathcal{C}_t(\delta)} \boldsymbol{\rho}^\top \boldsymbol{\sigma}(W\mathbf{x})$ and the first equality is due to the mean value theorem for the vector valued function. The second inequality is due to the Cauchy-Schwarz inequality and we can obtain the the last second inequality by the fact $\nabla \boldsymbol{\sigma}(\mathbf{z}) \preccurlyeq I_K$ for any $\mathbf{z} \in \mathbb{R}^K$. The last inequality is due to Assumption 2.

Let $G_t^{-\frac{1}{2}}(W_1, W_2)$ be defined as (21). We can proceed to handle the gap between $W_t^{\mathtt{OPT}}$ and $W_*$ by

$$\sum_{t=1}^{T} \|(W_t^{\mathtt{OPT}} - W_*)\mathbf{x}_t\|_{A(\mathbf{x}_t, W_*, W_t^{\mathtt{OPT}})}$$

$$\leq \sum_{t=1}^{T} \|(W_t^{\mathtt{OPT}} - W_*)\mathbf{x}_t\|_2$$

$$= \sum_{t=1}^{T} \|(I_K \otimes \mathbf{x}_t^\top) \cdot (\overrightarrow{W}_t^{\mathtt{OPT}} - \overrightarrow{W}_*)\|_2$$

$$= \sum_{t=1}^{T} \|(I_K \otimes \mathbf{x}_t^\top) G_t^{-\frac{1}{2}}(W_*, W_t^{\mathtt{OPT}}) \cdot G_t^{\frac{1}{2}}(W_*, W_t^{\mathtt{OPT}})(\overrightarrow{W}_t^{\mathtt{OPT}} - \overrightarrow{W}_*)\|_2$$

$$\leq \sum_{t=1}^{T} \|(I_K \otimes \mathbf{x}_t^\top) G_t^{-\frac{1}{2}}(W_*, W_t^{\mathtt{OPT}})\|_2 \cdot \|G_t^{\frac{1}{2}}(W_*, W_t^{\mathtt{OPT}})(\overrightarrow{W}_t^{\mathtt{OPT}} - \overrightarrow{W}_*)\|_2$$

$$\leq \sum_{t=1}^{T} \|(I_K \otimes \mathbf{x}_t^\top) \bar{V}_t^{-\frac{1}{2}}\|_2 \cdot \|G_t^{\frac{1}{2}}(W_*, W_t^{\mathtt{OPT}})(\overrightarrow{W}_t^{\mathtt{OPT}} - \overrightarrow{W}_*)\|_2$$

$$\leq \underbrace{\sqrt{\sum_{t=1}^{T} \|(I_K \otimes \mathbf{x}_t^\top) \bar{V}_t^{-\frac{1}{2}}\|_2^2}}_{\texttt{term (a)}} \cdot \underbrace{\sqrt{\sum_{t=1}^{T} \|G_t^{\frac{1}{2}}(W_*, W_t^{\mathtt{OPT}})(\overrightarrow{W}_t^{\mathtt{OPT}} - \overrightarrow{W}_*)\|_2^2}}_{\texttt{term (b)}}. \tag{22}$$

where the first inequality is due to the fact $A(\mathbf{x}_t, W_*, W_t^{\mathtt{OPT}}) \preccurlyeq I_K$ and the second inequality is due to the Cauchy-Schwarz inequality. Since both $W_*, W_t^{\mathtt{OPT}} \in \mathcal{W}$, the third inequality is by the condition $G_t(W_*, W_t^{\mathtt{OPT}}) \succcurlyeq \bar{V}_t \triangleq \lambda I_{Kd} + \frac{1}{\kappa} \sum_{s=1}^{t-1} I_K \otimes \mathbf{x}_s \mathbf{x}_s^\top$. We can show the last inequality by using Cauchy-Schwarz inequality again.

Then, we proceed to bound term (a) and term (b) respectively.

$$\texttt{term (a)} = \sqrt{\sum_{t=1}^{T} \|(I_K \otimes \mathbf{x}_t^\top) \bar{V}_t^{-\frac{1}{2}}\|_2^2}$$

$$= \sqrt{\sum_{t=1}^{T} \lambda_{\max}((I_K \otimes \mathbf{x}_t^\top) \bar{V}_t^{-1} (I_K \otimes \mathbf{x}_t))}$$

$$= \sqrt{\sum_{t=1}^{T} \lambda_{\max}\left((I_K \otimes \mathbf{x}_t^\top)\left(I_K \otimes \left(\lambda I_d + \frac{1}{\kappa} \sum_{s=1}^{t-1} \mathbf{x}_s \mathbf{x}_s^\top\right)\right)^{-1} (I_K \otimes \mathbf{x}_t)\right)}$$

$$= \sqrt{\sum_{t=1}^{T} \lambda_{\max}\left(I_K \otimes \mathbf{x}_t^\top \left(\lambda I_d + \frac{1}{\kappa}\sum_{s=1}^{t-1}\mathbf{x}_s\mathbf{x}_s^\top\right)^{-1}\mathbf{x}_t\right)}$$

$$= \sqrt{\sum_{t=1}^{T} \mathbf{x}_t^\top \left(\lambda I_d + \frac{1}{\kappa}\sum_{s=1}^{t-1}\mathbf{x}_s\mathbf{x}_s^\top\right)^{-1}\mathbf{x}_t}$$

$$\leq \sqrt{\kappa d \log\left(1 + \frac{T}{\kappa\lambda d}\right)} \qquad (23)$$

In above, we denote by $\lambda_{\max}$ the maximum eigenvalue of the input matrix. The second inequality is due to the property of the maximum eigenvalue such that $\lambda_{\max}(ABC) = \lambda_{\max}(BCA)$ for matrix $A, B$ and $C$. The last second equality is due to the mixed product property of the the Kronecker product and $(A \otimes B)^{-1} = A^{-1} \otimes B^{-1}$ for matrix $A$ and $B$. In the last inequality, we can bound the term with the standard arguments of the elliptical potential lemma [3, Lemma 11].

It remains to bound term (b). Since Theorem 1 shows that $W_* \in \mathcal{C}_t(\delta)$ with probability at least $1 - \delta$ for all $t > 1$ and the optimistic rule (5) ensures $W_t^{\texttt{OPT}} \in \mathcal{C}_t(\delta)$, Lemma 9 in Appendix B.2.2 indicates that the distance between $W_*$ and $W_t^{\texttt{OPT}}$ can be bounded by

$$\|G_t^{\frac{1}{2}}(W_*, W_t^{\texttt{OPT}})(\overrightarrow{W}_t^{\texttt{OPT}} - \overrightarrow{W}_*)\|_2 \leq 2\sqrt{1 + 2S}\beta_t(\delta).$$

Then, plugging the above displayed equation into term (b), we can bound this term by

$$\texttt{term (b)} \leq 2\sqrt{1 + 2S}\sqrt{\sum_{t=1}^{T}(\beta_t(\delta))^2} \leq 2\sqrt{(1 + 2S)T}\beta_T(\delta) = \mathcal{O}(\sqrt{dTK\log(T/\delta)}),$$

where the last inequality holds because $\beta_t(\delta)$ is a non-decreasing function with respect to $t$. A combination of the upper bound for term (a) and term (b) leads to

$$\text{Reg}_T \leq 8(1 + 2S)Rd\sqrt{\kappa KT \cdot \log\left(\frac{2(1 + T/d)}{\delta}\right) \cdot \log\left(1 + \frac{T}{\kappa d}\right)} = \mathcal{O}(d\log T\sqrt{\kappa KT}).$$

**Analysis for the** $\mathcal{O}(Kd\log T\sqrt{T} + \kappa Kd^2\log^2 T)$ **Bound.** Inspired by the analysis for binary logistic bandit problem [9, Appendix C], we employ a different decomposition compared to the one used in the first part of the analysis

$$\text{Reg}_T = \sum_{t=1}^{T}\sum_{k=1}^{K}\rho_k(\sigma_k(W_*\mathbf{x}_*) - \sigma_k(W_*\mathbf{x}_t))$$

$$\leq \sum_{t=1}^{T}\sum_{k=1}^{K}\rho_k(\sigma_k(W_t^{\texttt{OPT}}\mathbf{x}_t) - \sigma_k(W_*\mathbf{x}_t))$$

$$= \sum_{t=1}^{T}\sum_{k=1}^{K}\rho_k\nabla\sigma_k(W_*\mathbf{x}_t)^\top(W_t^{\texttt{OPT}} - W_*)\mathbf{x}_t + \sum_{k=1}^{K}\rho_k\|(W_t^{\texttt{OPT}} - W_*)\mathbf{x}_t\|_{\Xi_{k,t}}^2$$

$$\leq \sum_{t=1}^{T}\left|\sum_{k=1}^{K}\rho_k\nabla\sigma_k(W_*\mathbf{x}_t)^\top(W_t^{\texttt{OPT}} - W_*)\mathbf{x}_t\right| + \sum_{t=1}^{T}\left|\sum_{k=1}^{K}\rho_k\|(W_t^{\texttt{OPT}} - W_*)\mathbf{x}_t\|_{\Xi_{k,t}}^2\right|,$$
$$\underbrace{\hphantom{\sum_{t=1}^{T}\left|\sum_{k=1}^{K}\rho_k\nabla\sigma_k(W_*\mathbf{x}_t)^\top(W_t^{\texttt{OPT}} - W_*)\mathbf{x}_t\right|}}_{\texttt{term (c)}} \quad \underbrace{\hphantom{\sum_{t=1}^{T}\left|\sum_{k=1}^{K}\rho_k\|(W_t^{\texttt{OPT}} - W_*)\mathbf{x}_t\|_{\Xi_{k,t}}^2\right|}}_{\texttt{term (d)}}$$

where $\sigma_k : \mathbf{x} \mapsto [\boldsymbol{\sigma}(\mathbf{x})]_k$ is the $k$-th output of the vector-valued function $\boldsymbol{\sigma}(\mathbf{x})$ and $\Xi_{k,t} = \int_{\nu=0}^{1}(1 - \nu)\nabla^2\sigma_k((W_* + \nu(W_t^{\texttt{OPT}} - W_*))\mathbf{x}_t)d\nu$. In the above, the first inequality is due to the optimistic rule (5) and the last equality is due to the integral formulation of the Taylor series. Then, we proceed to handle term (c) and term (d) respectively.

As for term (c), we have

$$\texttt{term (c)} = \sum_{t=1}^{T}\left|\boldsymbol{\rho}\nabla\boldsymbol{\sigma}(W_*\mathbf{x}_t)^\top(\overrightarrow{W}_t^{\texttt{OPT}} - \overrightarrow{W}_*)\mathbf{x}_t\right|$$

$$= \sum_{t=1}^{T} \left| \boldsymbol{\rho} \nabla \boldsymbol{\sigma}(W_* \mathbf{x}_t)^\top (I_K \otimes \mathbf{x}_t^\top)(\overrightarrow{W}_t^{\mathtt{OPT}} - \overrightarrow{W}_*) \right|$$

$$= \sum_{t=1}^{T} \left| \boldsymbol{\rho} \nabla \boldsymbol{\sigma}(W_* \mathbf{x}_t)^\top (I_K \otimes \mathbf{x}_t^\top) G_t^{-\frac{1}{2}}(W_*, W_t^{\mathtt{OPT}}) \cdot G_t^{\frac{1}{2}}(W_*, W_t^{\mathtt{OPT}})(\overrightarrow{W}_t^{\mathtt{OPT}} - \overrightarrow{W}_*) \right|$$

$$\le \sum_{t=1}^{T} \| \overrightarrow{W}_t^{\mathtt{OPT}} - \overrightarrow{W}_* \|_{G_t(W_*, W_t^{\mathtt{OPT}})} \cdot \| G_t^{-\frac{1}{2}}(W_*, W_t^{\mathtt{OPT}})(I_K \otimes \mathbf{x}_t) \nabla \boldsymbol{\sigma}(W_* \mathbf{x}_t) \boldsymbol{\rho} \|_2$$

$$\le 2\sqrt{1 + 2S}\beta_T(\delta) \sum_{t=1}^{T} \| G_t^{-\frac{1}{2}}(W_*, W_t^{\mathtt{OPT}})(I_K \otimes \mathbf{x}_t) \nabla \boldsymbol{\sigma}(W_* \mathbf{x}_t) \boldsymbol{\rho} \|_2, \tag{24}$$

where the first inequality holds by Cauchy-Schwarz inequality and the last one is due to Lemma 9.
Let $\lambda_{\max}(\cdot)$ denote the maximum eigenvalue of the matrix. We can further bound (24) by

$$\sum_{t=1}^{T} \| G_t^{-\frac{1}{2}}(W_*, W_t^{\mathtt{OPT}})(I_K \otimes \mathbf{x}_t) \nabla \boldsymbol{\sigma}(W_* \mathbf{x}_t) \boldsymbol{\rho} \|_2$$

$$\le \sum_{t=1}^{T} \| \nabla \boldsymbol{\sigma}^{\frac{1}{2}}(W_* \mathbf{x}_t) \boldsymbol{\rho} \|_2 \cdot \| G_t^{-\frac{1}{2}}(W_*, W_t^{\mathtt{OPT}})(I_K \otimes \mathbf{x}_t) \nabla \boldsymbol{\sigma}^{\frac{1}{2}}(W_* \mathbf{x}_t) \|_2$$

$$\le R \sum_{t=1}^{T} \sqrt{\lambda_{\max}\left( (\nabla \boldsymbol{\sigma}^{\frac{1}{2}}(W_* \mathbf{x}_t) \otimes \mathbf{x}_t^\top) G_t^{-1}(W_*, W_t^{\mathtt{OPT}})(\nabla \boldsymbol{\sigma}^{\frac{1}{2}}(W_* \mathbf{x}_t) \otimes \mathbf{x}_t) \right)}$$

$$= R \sum_{t=1}^{T} \sqrt{\lambda_{\max}\left( (\nabla \boldsymbol{\sigma}(W_* \mathbf{x}_t) \otimes \mathbf{x}_t \mathbf{x}_t^\top) G_t^{-1}(W_*, W_t^{\mathtt{OPT}}) \right)}$$

$$\le R\sqrt{T} \sqrt{\sum_{t=1}^{T} \lambda_{\max}\left( (\nabla \boldsymbol{\sigma}(W_* \mathbf{x}_t) \otimes \mathbf{x}_t \mathbf{x}_t^\top) G_t^{-1}(W_*, W_t^{\mathtt{OPT}}) \right)}$$

$$\le R\sqrt{T} \sqrt{(1 + 2S)Kd \ln\left( 1 + \frac{T}{2\lambda} \right)} \tag{25}$$

where the first inequality is due to the fact that $\|A\mathbf{b}\|_2 \le \|A\|_2 \cdot \|\mathbf{b}\|$ for a matrix $A$ and vector $\mathbf{b}$. The second inequality is due to the definition of the induced norm $\|A\|_2 = \sqrt{\lambda_{\max}(A^T A)}$ and the mixed-product property of the Kronecker production. The last equality is due to the cycle property of the maximum eigenvalue such that $\lambda_{\max}(ABC) = \lambda_{\max}(CAB)$ for matrices $A, B$ and $C$ and we use the mixed-product property again. The last inequality is due to Lemma 10 in Appendix B.2.2.

Combining (24) and (25), we arrive

$$\mathtt{term\,(c)} \le 8R(1 + 2S)^{\frac{3}{2}} dK \sqrt{T \cdot \log\left( \frac{2(1 + t/d)}{\delta} \right) \cdot \log\left( 1 + \frac{T}{2d} \right)} = \mathcal{O}(dK \log T \sqrt{T}). \tag{26}$$

Then, we turn to handle term (d). Lemma 11 indicates that

$$\Xi_{k,t} = \int_{\nu=0}^{1} (1 - \nu) \nabla^2 \sigma_k((W_* + \nu(W_t^{\mathtt{OPT}} - W_*)) \mathbf{x}_t) \mathrm{d}\nu \preccurlyeq 3I_K \int_{\nu=0}^{1} (1 - \nu) \mathrm{d}\nu \preccurlyeq 3I_K. \tag{27}$$

As a consequence, we can bound term (d) by

$$\mathtt{term\,(d)} = \sum_{t=1}^{T} \left| \sum_{k=1}^{K} \rho_k \|(W_t^{\mathtt{OPT}} - W_*) \mathbf{x}_t\|_{\Xi_{k,t}}^2 \right|$$

$$\le 3 \sum_{t=1}^{T} \left| \sum_{k=1}^{K} \rho_k \|(W_t^{\mathtt{OPT}} - W_*) \mathbf{x}_t\|_2^2 \right|$$

$$\leq 3R \sum_{t=1}^{T} \|(W_t^{\texttt{OPT}} - W_*)\mathbf{x}_t\|_2^2$$

$$\leq 3R\beta_T^2(\delta) \sum_{t=1}^{T} \|(I_K \otimes \mathbf{x}_t^\top)\bar{V}_t^{-\frac{1}{2}}\|_2^2$$

$$\leq 3R\beta_T^2(\delta)\kappa d \log\left(1 + \frac{T}{\kappa\lambda d}\right)$$

where the first inequality is due to Assumption 2 and the second inequality is due to (27). The last second inequality can be obtained by a similar argument in obtaining (22). The last inequality can be obtained by the same argument as (23).

Combining (26) and (26), we obtain

$$\text{Reg}_T \leq 8R(1 + 2S)^{\frac{3}{2}} dK \sqrt{T \cdot \log\left(\frac{2(1 + T/d)}{\delta}\right) \cdot \log\left(1 + \frac{T}{2d}\right)}$$

$$+ 48R(1 + S)\kappa K d^2 \log\left(\frac{2(1 + T/d)}{\delta}\right) \cdot \log\left(1 + \frac{T}{2d}\right)$$

$$= \mathcal{O}(dK \log T \sqrt{T} + \kappa d^2 K \log^2 T).$$

We have completed the proof. $\qquad\square$

### B.2.2 Useful Lemma

This section provides the lemmas used in the main proof.

**Lemma 9.** *If $W_* \in \mathcal{C}_t(\delta)$, then for all $W \in \mathcal{C}_t(\delta)$:*

$$\|G_t^{\frac{1}{2}}(W_*, W)(\overrightarrow{W} - \overrightarrow{W}_*)\|_2 \leq 2\sqrt{1 + 2S}\beta_t(\delta),$$

*where $\beta_t(\delta) = 4\sqrt{Kd(1 + S)\log\left(2(1 + t/d)/\delta\right)} = \mathcal{O}(\sqrt{dK \log t})$ is defined in Theorem 1.*

*Proof of Lemma 9.* Lemma 3 in [11] shows that

$$\mathbf{g}_t(\overrightarrow{W}_*) - \mathbf{g}_t(\overrightarrow{W}) = G_t(W_*, W)(\overrightarrow{W}_* - \overrightarrow{W}),$$

for any $\overrightarrow{W} \in \mathcal{C}_t(\delta)$. Then, we have

$$\|G_t^{\frac{1}{2}}(W_*, W)(\overrightarrow{W} - \overrightarrow{W}_*)\|_2$$

$$= \|\mathbf{g}_t(\overrightarrow{W}_*) - \mathbf{g}_t(\overrightarrow{W})\|_{G_t^{-1}(W_*, W)}$$

$$\leq \|\mathbf{g}_t(\overrightarrow{W}_*) - \mathbf{g}_t(\overrightarrow{W}_t^{\texttt{MLE}})\|_{G_t^{-1}(W_*, W)} + \|\mathbf{g}_t(\overrightarrow{W}) - \mathbf{g}_t(\overrightarrow{W}_t^{\texttt{MLE}})\|_{G_t^{-1}(W_*, W)}$$

$$\leq \sqrt{1 + 2S}\left(\|\mathbf{g}_t(\overrightarrow{W}_*) - \mathbf{g}_t(\overrightarrow{W}_t^{\texttt{MLE}})\|_{H_t^{-1}(W_*)} + \|\mathbf{g}_t(\overrightarrow{W}) - \mathbf{g}_t(\overrightarrow{W}_t^{\texttt{MLE}})\|_{H_t^{-1}(W)}\right)$$

$$\leq 2\sqrt{1 + 2S}\beta_t(\delta),$$

where the last second inequality is due to generalized self-concordant properties of the logistic loss as shown by Lemma 13 of [11]. The last inequality is due to Theorem 1 and the condition that $W$ is contained in the confidence set $\mathcal{C}_t(\delta)$. $\qquad\square$

**Lemma 10.** *When the regularization parameter $\lambda > 2$, we have*

$$\sum_{t=1}^{T} \lambda_{\max}\left((\nabla\boldsymbol{\sigma}(W_*\mathbf{x}_t) \otimes \mathbf{x}_t\mathbf{x}_t^\top)G_t^{-1}(W_*, W_t^{\texttt{OPT}})\right) \leq (1 + 2S)Kd\ln\left(1 + \frac{T}{2\lambda}\right).$$

*Proof of Lemma 10.* Denoting by $\text{Tr}(A)$ the trace of matrix $A$, We have

$$\sum_{t=1}^{T} \lambda_{\max}\left((\nabla\boldsymbol{\sigma}(W_*\mathbf{x}_t) \otimes \mathbf{x}_t\mathbf{x}_t^\top)G_t^{-1}(W_*, W_t^{\texttt{OPT}})\right)$$

$$\leq \sum_{t=1}^{T} \mathrm{Tr}\Big( (\nabla \boldsymbol{\sigma}(W_* \mathbf{x}_t) \otimes \mathbf{x}_t \mathbf{x}_t^\top) G_t^{-1}(W_*, W_t^{\mathtt{OPT}}) \Big)$$

$$\leq (1 + 2S) \sum_{t=1}^{T} \mathrm{Tr}\Big( (\nabla \boldsymbol{\sigma}(W_* \mathbf{x}_t) \otimes \mathbf{x}_t \mathbf{x}_t^\top) H_t^{-1}(W_*) \Big),$$

where the last inequality is due to Lemma 4 of [11] such that $(1 + 2S)G_t(W_*, W_t^{\mathtt{OPT}}) \succcurlyeq H_t(W_*)$.

Let $M_t(W_*) = \frac{\lambda}{2} I_{Kd} + \sum_{s=1}^{t} \nabla \boldsymbol{\sigma}(W_* \mathbf{x}_s) \otimes \mathbf{x}_s \mathbf{x}_s^\top$. Then, we can further bound the last line of the above displayed equations by

$$(1 + 2S) \sum_{t=1}^{T} \mathrm{Tr}\Big( (\nabla \boldsymbol{\sigma}(W_* \mathbf{x}_t) \otimes \mathbf{x}_t \mathbf{x}_t^\top) H_t^{-1}(W_*) \Big)$$

$$= (1 + 2S) \sum_{t=1}^{T} \mathrm{Tr}\big( M_t(W_*) - M_{t-1}(W_*)) H_t^{-1}(W_*) \big)$$

$$\leq (1 + 2S) \sum_{t=1}^{T} \mathrm{Tr}\big( M_t(W_*) - M_{t-1}(W_*)) M_t^{-1}(W_*) \big)$$

$$\leq (1 + 2S) \sum_{t=1}^{T} \log \frac{|M_t(W_*)|}{|M_{t-1}(W_*)|}$$

$$\leq (1 + 2S) K d \ln \left( 1 + \frac{T}{2\lambda} \right)$$

The first inequality is by the fact $H_t(W_*) \succcurlyeq M_t(W_*)$ under the condition $\lambda \geq 2$. The last second inequality is due to Lemma 4.5 of [16]. $\qquad\square$

**Lemma 11.** *Let* $\sigma_k : \mathbf{z} \mapsto [\boldsymbol{\sigma}(\mathbf{x})]_k \in \mathbb{R}$ *by the k-th output of the vector-valued function* $\boldsymbol{\sigma}(\mathbf{z})$. *Then, for any* $\mathbf{z} \in \mathbb{R}^K$, *we have* $\nabla^2 \sigma_k(\mathbf{z}) \leq 3 I_K$ *for any* $k \in [K]$.

*Proof of Lemma 11.* For any $\mathbf{z} \in \mathbb{R}^K$, we have

$$3 I_K - \nabla^2 \sigma_k(\mathbf{z})$$
$$= 3 I_K + \sigma_k(\mathbf{z}) \cdot \big( \mathrm{diag}(\boldsymbol{\sigma}(\mathbf{z})) + \mathbf{e}_k \boldsymbol{\sigma}(\mathbf{z})^\top + \boldsymbol{\sigma}(\mathbf{z}) \mathbf{e}_k^\top \big) - 2\sigma_k \cdot (\mathbf{z}) \boldsymbol{\sigma}(\mathbf{z}) \boldsymbol{\sigma}(\mathbf{z})^\top - \sigma_k(\mathbf{z}) E_k$$
$$\succcurlyeq 3 I_K - 2\sigma_k(\mathbf{z}) \boldsymbol{\sigma}(\mathbf{z}) \boldsymbol{\sigma}(\mathbf{z})^\top + \sigma_k(\mathbf{z})(\mathbf{e}_k \boldsymbol{\sigma}(\mathbf{z})^\top + \boldsymbol{\sigma}(\mathbf{z}) \mathbf{e}_k^\top - E_k) \tag{28}$$
$$= 3 I_K - \sigma_k(\mathbf{z}) \boldsymbol{\sigma}(\mathbf{z}) \boldsymbol{\sigma}(\mathbf{z})^\top - \sigma_k(\mathbf{z})(\boldsymbol{\sigma}(\mathbf{z}) - \mathbf{e}_k)(\boldsymbol{\sigma}(\mathbf{z}) - \mathbf{e}_k)^\top \succcurlyeq 0 \tag{29}$$

where we denote by $E_k$ as the matrix where the entry at the $(k, k)$-th position is 1, and all other entries are 0. In the above, the first inequality is due to the definition of $\sigma_k$. The second inequality holds since $\mathrm{diag}(\boldsymbol{\sigma}(\mathbf{z}))$ is semi-positive defined matrix. The last inequality is a consequence of the fact that the maximum eigenvalue of $\sigma_k(\mathbf{z}) \boldsymbol{\sigma}(\mathbf{z}) \boldsymbol{\sigma}(\mathbf{z})^\top$ is bounded by $\sigma_k(\mathbf{z})(\boldsymbol{\sigma}(\mathbf{z}) - \mathbf{e}_k)(\boldsymbol{\sigma}(\mathbf{z}) - \mathbf{e}_k)^\top$ is less than 1. The maximum eigenvalue of $\sigma_k(\mathbf{z})(\boldsymbol{\sigma}(\mathbf{z}) - \mathbf{e}_k)(\boldsymbol{\sigma}(\mathbf{z}) - \mathbf{e}_k)^\top$ is bounded by 2. $\qquad\square$

## C    Omitted Proofs for Section 3

This section presents the omitted details for Section 3. To simplify the notation, we exclude the superscript OL in all the proofs within this section.

### C.1    Proof of Theorem 3

This section provides the proof of Theorem 3. We will first provide the main proof of Theorem 3 and the prove the technical lemma used in the main proof.

### C.1.1 Main Proof

*Proof of Theorem 3.* The proof shares the similarity with the online-to-confidence-set conversion technique [47, 14], where the estimation error of the online estimator $W_t^{\text{OL}}$ is bounded by the regret. However, we have introduced a novel algorithm and analysis ingredient to achieve jointly statistical and computation efficiency.

The following lemma provides the estimation error analysis. Drawing inspiration from the modern analysis of OMD [23], we explicitly extract the negative term $-\sum_{s=1}^{t}\|\overrightarrow{W}_{s+1} - \overrightarrow{W}_s\|_{H_s}^2$ in the upper bound, which is pivotal for our subsequent theoretical analysis and the algorithm design.

**Lemma 12.** *Under Assumptions 1 and 3, we consider the estimator update rule*

$$\overrightarrow{W}_{t+1} = \underset{W \in \mathcal{W}}{\arg\min}\left\{\widetilde{\ell}_t(W) + \frac{1}{2\eta}\|\overrightarrow{W} - \overrightarrow{W}_t\|_{H_t}^2\right\},$$

*where $\widetilde{\ell}_t(W) = \langle \nabla\ell_t(\overrightarrow{W}_t), \overrightarrow{W}\rangle + \frac{1}{2}\|\overrightarrow{W} - \overrightarrow{W}_t\|_{\nabla^2\ell_t(\overrightarrow{W}_t)}^2$ and $H_t = \lambda I_{Kd} + \sum_{s=1}^{t-1}\nabla^2\ell_s(W_{s+1}) \otimes \mathbf{x}_s\mathbf{x}_s^\top$. Then, letting $\alpha = \ln(K+1) + 2(S+1)$ and $\lambda > 0$, we have*

$$\|\overrightarrow{W}_{t+1} - \overrightarrow{W}_*\|_{H_{t+1}}^2 \leq \alpha\left(\sum_{s=1}^{t}\ell_s(W_*) - \sum_{s=1}^{t}\ell_s(W_{s+1})\right) + 4\lambda S^2$$

$$+ \sum_{s=1}^{t}\sqrt{6}\alpha S\|\overrightarrow{W}_{s+1} - \overrightarrow{W}_s\|_{I_K \otimes \mathbf{x}_s\mathbf{x}_s^\top}^2 - \sum_{s=1}^{t}\|\overrightarrow{W}_{s+1} - \overrightarrow{W}_s\|_{H_s}^2,$$

*when the step size is set as $\eta = \alpha/2$.*

Then, we focus on the first term of the right hand side. Inspired by the previous studies on binary logistic bandit [10], we decompose the regret into two part by inserting an intermediate decision.

$$\sum_{s=1}^{t}\ell_s(W_*) - \sum_{s=1}^{t}\ell_s(W_{s+1})$$

$$= \underbrace{\sum_{s=1}^{t}\ell_s(W_*) - \sum_{s=1}^{t}\ell(\widetilde{\mathbf{z}}_s, y_s)}_{\text{term (A)}} + \underbrace{\sum_{s=1}^{t}\ell(\widetilde{\mathbf{z}}_s, y_s) - \sum_{s=1}^{t}\ell_s(W_{s+1})]}_{\text{term (B)}},$$

where the $\widetilde{\mathbf{z}}_s$ is an aggregating forecaster for logistic loss defined by $\widetilde{\mathbf{z}}_s = \boldsymbol{\sigma}^+\left(\mathbb{E}_{W \sim P_s}[\boldsymbol{\sigma}(W\mathbf{x}_s)]\right)$ and $P_s = \mathcal{N}(\overrightarrow{W}_s, (1 + cH_s^{-1})$ is the Gaussian distribution with mean $\overrightarrow{W}_s$ and covariance matrix $cH_s^{-1}$, where $c > 0$ is a constant to be specified. In above, $\boldsymbol{\sigma}^+ : \Delta \mapsto \mathbb{R}^K$ is a pseudo-inverse function of $\boldsymbol{\sigma}(\cdot)$ whose $k$-th output is $[\boldsymbol{\sigma}^+(\mathbf{p})]_k = \log\left(p_k/(1 - \|\mathbf{p}\|_1)\right)$ for any $\mathbf{p} \in \{\mathbf{q} \in [0,1]^K | \|\mathbf{q}\|_1 < 1\}$. We bound the above two terms respectively. First, we show that the term (A) is bounded by $\mathcal{O}\left(\log^2 t\right)$ with high probability.

**Lemma 13.** *Let $\delta \in (0, 1]$. Under Assumptions 1 and 3, we have*

$$\Pr\left[\forall t \geq 1, \sum_{s=1}^{t}\ell_s(W_*) - \sum_{s=1}^{t}\ell(\widetilde{\mathbf{z}}_s, y_s) \leq \gamma_t^{\text{A}}(\delta)\right] \geq 1 - \delta,$$

*where the confidence radius is $\gamma_t^{\text{A}} = (\log(1+K) + 2\log(1+(1+K)t))\left(\frac{5}{4} + 4\log\left(\frac{\sqrt{1+2t}}{\delta}\right)\right) + 2 = \mathcal{O}\left((\log K + \log t)\log t\right)$.*

Besides, the term (b) can be bounded by the following lemma,

**Lemma 14.** *Under Assumptions 1 and 3, for any $c > 0$, we have*

$$\sum_{s=1}^{t}\ell(\widetilde{\mathbf{z}}_s, y_s) - \sum_{s=1}^{t}\ell_s(W_{s+1}) \leq \frac{1}{c}\sum_{s=1}^{t}\|\overrightarrow{W}_s - \overrightarrow{W}_{s+1}\|_{H_s}^2 + \gamma_t^{\text{B}}(\delta),$$

by setting $\lambda \geq \max\{2, 24Kdc\}$. In the above, $\gamma_t^{\text{B}}(\delta) = \sqrt{6}cKd\ln\left(1 + \frac{(t+1)L}{2\lambda}\right)$ and $L := \max_{W \in \mathbb{R}^{Kd}, t \in [T]} \nabla^2 \ell_t(W)$ is the smooth parameter of the logistic loss.

Combining Lemma 12, Lemma 13 and Lemma 14, we arrive at

$$\|\overrightarrow{W}_{t+1} - \overrightarrow{W}_*\|_{H_{t+1}}$$

$$\leq \sqrt{4\lambda S^2 + \alpha\gamma_t^{\text{A}} + \alpha\gamma_t^{\text{B}} + (\frac{\alpha}{c} - 1)\sum_{s=1}^{t}\|\overrightarrow{W}_{s+1} - \overrightarrow{W}_s\|_{H_s}^2 + \sqrt{6}\alpha S\sum_{s=1}^{t}\|\overrightarrow{W}_{s+1} - \overrightarrow{W}_s\|_{I_K \otimes \mathbf{x}_s \mathbf{x}_s^\top}^2}$$

$$\leq \sqrt{4\lambda S^2 + \alpha\gamma_t^{\text{A}} + \alpha\gamma_t^{\text{B}}}$$

$$= \sqrt{\underbrace{4\lambda S^2}_{=\mathcal{O}(Kd\log K)} + \underbrace{\alpha(3\log K + 2\log t)\left(\frac{5}{4} + 4\log\left(\frac{\sqrt{1+2t}}{\delta}\right)\right) + 2}_{=\mathcal{O}((\log K + \log t)\log t)} + \underbrace{7\alpha^2 Kd\ln\left(1 + (t+1)L\right)/\sqrt{6}}_{=\mathcal{O}(Kd\log^2 K \log t)}}$$

$$\triangleq \beta_t^{\text{OL}}(\delta).$$

For the last second inequality, we eliminate the additional term by the parameter setting $\eta = \alpha/2$, $c = 7\alpha/6$ and $\lambda \geq \{7\sqrt{6}\alpha S, 28Kd\alpha\}$ and due to the fact that $I_K \otimes \mathbf{x}_s \mathbf{x}_s^\top \preccurlyeq I_{Kd}$ under the condition $\|\mathbf{x}_t\|_2 \leq 1$ for any $t \in [T]$. In the above, we have $\lambda = \mathcal{O}(Kd\log K)$ and $\alpha = \mathcal{O}(\log K)$. Then, we can show $\|\overrightarrow{W}_{t+1} - \overrightarrow{W}_*\|_{H_{t+1}} \leq \beta_t^{\text{OL}}(\delta) = \mathcal{O}(\log K \log t\sqrt{Kd})$, which completes the proof. $\square$

### C.1.2  Proof of Lemma 12

*Proof of Lemma 12.* Let $\widetilde{\ell}_s(W) = \ell_s(W_s) + \langle \overrightarrow{W} - \overrightarrow{W}_s, \nabla\ell_s(\overrightarrow{W}_s)\rangle + \frac{1}{2}\|\overrightarrow{W} - \overrightarrow{W}_s\|_{\nabla^2\ell_s(\overrightarrow{W}_s)}^2$ be a second order approximation of the original function $\ell_s(W)$ at the point $\overrightarrow{W}_s$. The update rule (6) can be equally written as

$$\overrightarrow{W}_{s+1} = \arg\min_{W \in \mathcal{W}} \widetilde{\ell}_s(W) + \frac{1}{2\eta}\|\overrightarrow{W} - \overrightarrow{W}_s\|_{H_s}^2, \tag{30}$$

which is an implicit online mirror descent update with the loss function $\widetilde{\ell}_s$. Then, according to Lemma 16, the model $W_{t+1}$ ensures,

$$\langle \nabla\widetilde{\ell}_s(\overrightarrow{W}_{s+1}), \overrightarrow{W}_{s+1} - \overrightarrow{W}_*\rangle \leq \frac{1}{2\eta}\left(\|\overrightarrow{W}_s - \overrightarrow{W}_*\|_{H_s}^2 - \|\overrightarrow{W}_{s+1} - \overrightarrow{W}_*\|_{H_s}^2 - \|\overrightarrow{W}_{s+1} - \overrightarrow{W}_s\|_{H_s}^2\right).$$
$$\tag{31}$$

One the other hand, let $\alpha = \log(1 + K) + 2(1 + S)$. Since $\|W_*\mathbf{x}_t\|_\infty = \max_{k \in [K]}\{\|\mathbf{w}_*^{(k)}\|_2 \cdot \|\mathbf{x}_t\|_2\} \leq S$ and $\ell_s(W) = \ell(W\mathbf{x}_s, y_s)$, Lemma 1 shows

$$\ell_s(W_{s+1}) - \ell_s(W_*) \leq \langle \nabla\ell_s(W_{s+1}), \overrightarrow{W}_{s+1} - \overrightarrow{W}_*\rangle - \frac{1}{\alpha}\|\overrightarrow{W}_{s+1} - \overrightarrow{W}_*\|_{\nabla^2\ell_s(W_{s+1})}^2. \tag{32}$$

By setting $\eta = \alpha/2$, the combination of (31) and (32) shows

$$\ell_s(W_{s+1}) - \ell_s(W_*) \leq \langle \nabla\ell_s(W_{s+1}) - \nabla\widetilde{\ell}_s(W_{s+1}), \overrightarrow{W}_{s+1} - \overrightarrow{W}_*\rangle \tag{33}$$
$$+ \frac{1}{\alpha}\left(\|\overrightarrow{W}_s - \overrightarrow{W}_*\|_{H_s}^2 - \|\overrightarrow{W}_{s+1} - \overrightarrow{W}_*\|_{H_{s+1}}^2 - \|\overrightarrow{W}_{s+1} - \overrightarrow{W}_s\|_{H_s}^2\right).$$

In above, we can further bound inner product term by

$$\langle \nabla\ell_s(W_{s+1}) - \nabla\widetilde{\ell}_s(W_{s+1}), \overrightarrow{W}_{s+1} - \overrightarrow{W}_*\rangle$$
$$= \langle \nabla\ell_s(W_{s+1}) - \nabla\ell_s(W_s) - \nabla^2\ell_s(\overrightarrow{W}_s)(\overrightarrow{W}_{s+1} - \overrightarrow{W}_s), \overrightarrow{W}_{s+1} - \overrightarrow{W}_*\rangle$$
$$= \left\langle D^3\ell_s(\xi_{s+1})[\overrightarrow{W}_{s+1} - \overrightarrow{W}_s](\overrightarrow{W}_{s+1} - \overrightarrow{W}_s), \overrightarrow{W}_{s+1} - \overrightarrow{W}_*\right\rangle$$
$$= \left\langle D^3\ell_s(\xi_{s+1})[\overrightarrow{W}_{s+1} - \overrightarrow{W}_*](\overrightarrow{W}_{s+1} - \overrightarrow{W}_s), \overrightarrow{W}_{s+1} - \overrightarrow{W}_s\right\rangle$$

$$\leq \sqrt{6}\|\overrightarrow{W}_{s+1} - \overrightarrow{W}_*\|_2 \cdot \|\overrightarrow{W}_{s+1} - \overrightarrow{W}_s\|_{\nabla^2 \ell_s(\xi_{s+1})}^2$$

$$\leq \sqrt{6}S\|\overrightarrow{W}_{s+1} - \overrightarrow{W}_s\|_{I_K \otimes \mathbf{x}_s \mathbf{x}_s^\top}^2 \tag{34}$$

In above $\xi_{s+1}$ is a certain point on the line connecting $\overrightarrow{W}_t$ and $\overrightarrow{W}_{t+1}$. The notation is defined as $D^3 \ell_s(\overrightarrow{W})[\overrightarrow{U}] = \lim_{\alpha \to 0} \alpha^{-1} \left( \nabla^2 \ell_s(\overrightarrow{W} + \alpha \overrightarrow{U}) - \nabla^2 \ell_s(\overrightarrow{W}) \right)$. The first inequality is due to the definition of $\ell_s$ and the second inequality is by the Taylor series for the vector-valued function. The third inequality is because the third order derivative of the logistic loss is a symmetric tensor as shown in Property 1. The last second inequality is by Definition 1 since the multiclass logistic loss is a $\sqrt{6}$ self-concordant-like function. The last inequality is by the boundedness of the decision domain $\mathcal{W}$ and bounded action such that $\|\mathbf{x}\|_2 \leq 1$ for any $\mathbf{x} \in \mathcal{X}$ under Assumption 1.

Combining (33) and (34), we have

$$\ell_s(W_{s+1}) - \ell_s(W_*)$$
$$\leq \frac{1}{\alpha} \left( \|\overrightarrow{W}_s - \overrightarrow{W}_*\|_{H_s}^2 - \|\overrightarrow{W}_{s+1} - \overrightarrow{W}_*\|_{H_{s+1}}^2 - \|\overrightarrow{W}_{s+1} - \overrightarrow{W}_s\|_{H_s}^2 \right) + \sqrt{6}S\|\overrightarrow{W}_{s+1} - \overrightarrow{W}_s\|_{I_K \otimes \mathbf{x}_s \mathbf{x}_s^\top}^2,$$

Taking the summation of the above inequality over $t$ rounds and rearranging the term, we have

$$\|\overrightarrow{W}_{t+1} - \overrightarrow{W}_*\|_{H_{t+1}}^2 \leq \alpha \left( \sum_{s=1}^t \ell_s(W_*) - \sum_{s=1}^t \ell_s(W_{s+1}) \right) + \|\overrightarrow{W}_1 - \overrightarrow{W}_*\|_{H_1}^2$$

$$+ \sum_{s=1}^t \left( \sqrt{6}\alpha S\|\overrightarrow{W}_{s+1} - \overrightarrow{W}_s\|_{I_K \otimes \mathbf{x}_s \mathbf{x}_s^\top}^2 - \|\overrightarrow{W}_{s+1} - \overrightarrow{W}_s\|_{H_s}^2 \right)$$

$$\leq \alpha \left( \sum_{s=1}^t \ell_s(W_*) - \sum_{s=1}^t \ell_s(W_{s+1}) \right) + 4\lambda S^2$$

$$+ \sum_{s=1}^t \left( \sqrt{6}\alpha S\|\overrightarrow{W}_{s+1} - \overrightarrow{W}_s\|_{I_K \otimes \mathbf{x}_s \mathbf{x}_s^\top}^2 - \|\overrightarrow{W}_{s+1} - \overrightarrow{W}_s\|_{H_s}^2 \right),$$

which completes the proof.

$\square$

### C.1.3 Proof of Lemma 13

*Proof of Lemma 13.* Since the norm of the intermediate decision $\widetilde{\mathbf{z}}_s = \boldsymbol{\sigma}^+(\mathbb{E}_{W \sim P_s}[\boldsymbol{\sigma}(W\mathbf{x}_s)])$ is generally unbounded, as suggested by [26], we use the smoothed version $\widetilde{\mathbf{z}}_s^\mu = \boldsymbol{\sigma}^+ (\text{smooth}_\mu(\mathbb{E}_{W \sim P_s}[\boldsymbol{\sigma}(W\mathbf{x}_s)]))$ as an intermediate term in the analysis. In above, the smooth function $\text{smooth}_\mu : [0,1]^K \mapsto [0,1]^K$ with parameter $\mu \in [0, 1/2]$ is defined by $\text{smooth}_\mu(\mathbf{p}) = (1-\mu)\mathbf{p} + \mu\mathbf{1}/(K+1)$, where $\mathbf{1} \in \mathbb{R}^K$ is an all one vector.

Given the construction of the pseudo inverse function $\boldsymbol{\sigma}^+$ such that $\boldsymbol{\sigma}(\boldsymbol{\sigma}^+(\mathbf{p})) = \mathbf{p}$ for any $\mathbf{p} \in \{\mathbf{q} \in [0,1]^K \mid \|\mathbf{q}\|_1 < 1\}$, one can check that $\widetilde{\mathbf{z}}_s^\mu = \boldsymbol{\sigma}^+ (\text{smooth}_\mu(\boldsymbol{\sigma}(\widetilde{\mathbf{z}}_s)))$. Then, Lemma 17 shows that

$$\sum_{s=1}^t \ell(\widetilde{\mathbf{z}}_s^\mu, y_s) - \sum_{s=1}^t \ell(\widetilde{\mathbf{z}}_s, y_s) \leq 2\mu t. \tag{35}$$

Besides, Lemma 17 also shows that $\|\widetilde{\mathbf{z}}_s^\mu\|_\infty \leq \log(1 + (K+1)/\mu)$. Therefore, to prove the lemma, it is sufficient bound the gap between the loss of $W_*$ and $\widetilde{\mathbf{z}}_s^\mu$. By the definition of the loss function $\ell_s$, we have $\ell_s(W_*) = \ell(\mathbf{z}_s^*, \mathbf{y}_s)$, where $\mathbf{z}_s^* = W_* \mathbf{x}_s$. Then, we have

$$\sum_{s=1}^t \ell_s(W_*) - \sum_{s=1}^t \ell(\widetilde{\mathbf{z}}_s^\mu, y_s) = \sum_{s=1}^t \ell(\mathbf{z}_s^*, y_s) - \sum_{s=1}^t \ell(\widetilde{\mathbf{z}}_s^\mu, y_s)$$

$$\leq \sum_{s=1}^t \langle \nabla_z \ell(\mathbf{z}_s^*, y_s), \mathbf{z}_s^* - \widetilde{\mathbf{z}}_s^\mu \rangle - \sum_{s=1}^t \frac{1}{S_\mu} \|\mathbf{z}_s^* - \widetilde{\mathbf{z}}_s^\mu\|_{\nabla_z^2 \ell(\mathbf{z}_s^*, y_s)}^2$$

$$= \sum_{s=1}^{t} \langle \boldsymbol{\sigma}(\mathbf{z}_s^*) - \mathbf{y}_s, \mathbf{z}_s^* - \widetilde{\mathbf{z}}_s^{\mu} \rangle - \sum_{s=1}^{t} \frac{1}{S_{\mu}} \|\mathbf{z}_s^* - \widetilde{\mathbf{z}}_s^{\mu}\|_{\nabla \boldsymbol{\sigma}(\mathbf{z}_s^*)}^2 \qquad (36)$$

where $S_{\mu} = \log(K+1) + 2\log(1 + (K+1)/\mu)$. In above, the first inequality is due to Lemma 1. The second equality is by a direct calculation of the first order and Hessian of the logistic loss as shown in Property 1.

To bound the first term of the right hand side, we require the following lemma, which adapts lemma 6 to a one-dimension case and whose proof is provided C.1.5.

**Lemma 15.** *Let $\{\mathcal{F}_t\}_{t=1}^{\infty}$ be a filtration. Let $\{\mathbf{z}_t\}_{t=1}^{\infty}$ be a stochastic process in $\mathcal{B}_2(K) = \{\mathbf{z} \in \mathbb{R}^K \mid \|\mathbf{z}\|_{\infty} \le 1\}$ such that $\mathbf{z}_t$ is $\mathcal{F}_t$ measurable. Let $\{\varepsilon_t\}_{t=1}^{\infty}$ be a martingale difference sequence such that $\varepsilon_t \in \mathbb{R}^K$ is $\mathcal{F}_{t+1}$ measurable. Furthermore, assume that conditional on $\mathcal{F}_t$, we have $\|\varepsilon_t\|_1 \le 2$ almost surely, and denote by $\Sigma_t := \mathbb{E}[\varepsilon_t \varepsilon_t^{\top} \mid \mathcal{F}_t]$. Let $\lambda > 0$ and for any $t \ge$ define*

$$S_t = \sum_{s=1}^{t-1} \langle \varepsilon_s, \mathbf{z}_s \rangle \quad \text{and} \quad H_t = \lambda + \sum_{s=1}^{t-1} \|\mathbf{z}_s\|_{\Sigma_s}^2.$$

*Then for any $\delta \in (0,1]$, we have*

$$\Pr\left[ \exists t > 1, S_t \ge \sqrt{H_t} \left( \frac{\sqrt{\lambda}}{4} + \frac{4}{\sqrt{\lambda}} \log\left( \sqrt{\frac{H_t}{\lambda}} \right) + \frac{4}{\sqrt{\lambda}} \log(2/\delta) \right) \right] \le \delta.$$

To check the condition of Lemma 15, we let $\mathbf{d}_s = (\mathbf{z}_s^* - \widetilde{\mathbf{z}}_s^{\mu})/(S_{\mu} + S)$. Since $\|\mathbf{z}_s^*\|_{\infty} \le \max_{k \in [K]} \|\mathbf{w}_*^{(k)}\|_2 \|\mathbf{x}_s\|_2 \le S$ and $\|\widetilde{\mathbf{z}}_s^{\mu}\|_{\infty} \le \log(1 + (K+1)/\mu)$, one can verify that $\|\mathbf{d}_s\|_{\infty} \le 1$. Besides, since $\mathbf{z}_s^*$ and $\widetilde{\mathbf{z}}_s^{\mu}$ is independent of $\mathbf{y}_s$, the variable $\mathbf{d}_s$ is $\mathcal{F}_s$ measurable. Furthermore, we can check that $\mathbb{E}[(\boldsymbol{\sigma}(\mathbf{z}_s^*) - \mathbf{y}_s)(\boldsymbol{\sigma}(\mathbf{z}_s^*) - \mathbf{y}_s)^{\top} \mid \mathcal{F}_s] = \nabla \boldsymbol{\sigma}(\mathbf{z}_s^*)$ and $\|\boldsymbol{\sigma}(\mathbf{z}_s^*) - \mathbf{y}_s\|_1 \le 2$. Thus, a direct application of Lemma 15 shows that, with probability at least $1 - \delta$, we have

$$\sum_{s=1}^{t} \langle \boldsymbol{\sigma}(\mathbf{z}_s^*) - \mathbf{y}_s, \mathbf{z}_s^* - \widetilde{\mathbf{z}}_s^{\mu} \rangle \qquad (37)$$

$$= (S_{\mu} + S) \sum_{s=1}^{t} \langle \boldsymbol{\sigma}(\mathbf{z}_s^*) - \mathbf{y}_s, \mathbf{d}_s \rangle$$

$$\le (S_{\mu} + S) \sqrt{\lambda + \sum_{s=1}^{t} \|\mathbf{d}_s\|_{\nabla \boldsymbol{\sigma}(\mathbf{z}_s^*)}^2} \cdot \sqrt{\frac{\sqrt{\lambda}}{4} + \frac{4}{\sqrt{\lambda}} \log\left( \frac{\sqrt{1 + \sum_{s=1}^{t} \|\mathbf{d}_s\|_{\nabla \boldsymbol{\sigma}(\mathbf{z}_s^*)}^2}}{\delta} \right)}$$

$$\le (S_{\mu} + S) \sqrt{\lambda + \sum_{s=1}^{t} \|\mathbf{d}_s\|_{\nabla \boldsymbol{\sigma}(\mathbf{z}_s^*)}^2} \cdot \sqrt{\frac{\sqrt{\lambda}}{4} + \frac{4}{\sqrt{\lambda}} \log\left( \frac{\sqrt{1 + 2t}}{\delta} \right)}, \qquad (38)$$

for any $t \ge 1$. In above, the second inequality is a consequence of the fact $\|\mathbf{d}_s\|_{\Sigma(\mathbf{z}_s^*)}^2 = \mathbf{d}_s^{\top} \nabla \boldsymbol{\sigma}(\mathbf{z}_s^*) \mathbf{d}_s \le 2$. Then, combining (36) and (38) and setting $\lambda = 1$, we arrive [‡]

$$\sum_{s=1}^{t} \ell_s(W_*) - \sum_{s=1}^{t} \ell(\widetilde{\mathbf{z}}_s^{\mu}, \mathbf{y}_s)$$

$$\le (S_{\mu} + S) \sqrt{1 + \sum_{s=1}^{t} \|\mathbf{d}_s\|_{\Sigma(\mathbf{z}_s^*)}^2} \cdot \sqrt{\frac{1}{4} + 4\log\left( \frac{\sqrt{1+2t}}{\delta} \right)} - (S_{\mu} + S) \sum_{s=1}^{t} \|\mathbf{d}_s\|_{\Sigma(\mathbf{z}_s^*)}^2$$

$$\le (S_{\mu} + S) \left( 1 + \sum_{s=1}^{t} \|\mathbf{d}_s\|_{\Sigma(\mathbf{z}_s^*)} \right) + (S_{\mu} + S) \left( \frac{1}{4} + 4\log\left( \frac{\sqrt{1+2t}}{\delta} \right) \right) - (S_{\mu} + S) \sum_{s=1}^{t} \|\mathbf{d}_s\|_{\Sigma(\mathbf{z}_s^*)}^2$$

$$\le \frac{5}{4}(S_{\mu} + S) + 4(S_{\mu} + S) \log\left( \frac{\sqrt{1+2t}}{\delta} \right), \qquad (39)$$

---

[‡]We note that the $\lambda$ here is irrelevant of the algorithm, we can set it as any value.

where the second inequality is due to AM-GM inequality. Finally, combining (35) and (39), we have

$$\sum_{s=1}^{t} \ell_s(W_*) - \sum_{s=1}^{t} \ell(\widetilde{\mathbf{z}}_s, \mathbf{y}_s) \le \frac{5}{4}(S_\mu + S) + 4(S_\mu + S) \log\left(\frac{\sqrt{1+2t}}{\delta}\right) 2 + \mu t$$

$$\le 3\log(1 + (1+K)t)\left(\frac{5}{4} + 4\log\left(\frac{\sqrt{1+2t}}{\delta}\right)\right) + 2,$$

where the last inequality is by the parameter setting $\mu = 1/t$ and, as a consequence, $S_\mu = \log(1 + K) + 2\log(1 + (K+1)t)$. We have completed the proof. □

### C.1.4 Proof of Lemma 14

*Proof of Lemma 14.* Our proof starts with the observation made by [26, Proposition 2] that $\widetilde{\mathbf{z}}_s$ is an aggregating forecaster [48, Chapter, 3.5] for the logistic function, which satisfies

$$\ell(\widetilde{\mathbf{z}}_s, \mathbf{y}_s) \le -\ln\left(\mathbb{E}_{W \sim P_s}\left[e^{-\ell_s(W)}\right]\right).$$

Then, a direct calculation with the definition of Gaussian distribution $P_s \sim \mathcal{N}(\overrightarrow{W}_s, cH_s^{-1})$ gives

$$\ell(\widetilde{\mathbf{z}}_s, \mathbf{y}_s) \le -\ln\left(\mathbb{E}_{W \sim P_s}\left[e^{-\ell_s(W)}\right]\right) = -\ln\left(\frac{1}{Z_s}\int_{\mathbb{R}^{Kd}} e^{-L_s(W)}\mathrm{d}\overrightarrow{W}\right), \tag{40}$$

where we define the loss function $L_s(W) = \ell_s(W) + (2c)^{-1}\|\overrightarrow{W} - \overrightarrow{W}_s\|_{H_s}^2$ and $Z_s = \sqrt{(2\pi)^{Kd}c|H_s^{-1}|}$ being the normalization factor.

Then, we consider the following quadratic approximation,

$$\widetilde{L}_s(W) = L_s(W_{s+1}) + \langle \nabla L_s(W_{s+1}), \overrightarrow{W} - \overrightarrow{W}_{s+1} \rangle + \frac{1}{2c}\|\overrightarrow{W} - \overrightarrow{W}_{s+1}\|_{H_s}^2. \tag{41}$$

According to Lemma 18, we have

$$L_s(W) \le \widetilde{L}_s(W) + e^{6\|\overrightarrow{W} - \overrightarrow{W}_{s+1}\|_2^2}\|\overrightarrow{W} - \overrightarrow{W}_{s+1}\|_{\nabla \ell_s(W_{s+1})}^2.$$

Then, we can low bound the term in the expectation by

$$\mathbb{E}_{W \sim P_s}\left[e^{-\ell_s(W)}\right]$$

$$= \frac{1}{Z_s}\int_{\mathbb{R}^{Kd}} e^{-L_s(W)}\mathrm{d}W$$

$$\ge \frac{1}{Z_s}\int_{\mathbb{R}^{Kd}} e^{-\widetilde{L}_s(W) - e^{6\|\overrightarrow{W} - \overrightarrow{W}_{s+1}\|_2^2}\|\overrightarrow{W} - \overrightarrow{W}_{s+1}\|_{\nabla \ell_s(W_{s+1})}^2}\mathrm{d}W$$

$$= \frac{e^{-L_s(W_{s+1})}}{Z_s}\int_{\mathbb{R}^{Kd}} \widetilde{f}_{s+1}(W) \cdot e^{-\langle \nabla L_s(W_{s+1}), \overrightarrow{W} - \overrightarrow{W}_{s+1} \rangle}\mathrm{d}W, \tag{42}$$

where we define the function $\widetilde{f}_s : \mathcal{W} \mapsto \mathbb{R}$ as

$$\widetilde{f}_{s+1}(W) = \exp\left(-\frac{1}{2c}\|\overrightarrow{W} - \overrightarrow{W}_{s+1}\|_{H_s}^2 - e^{6\|\overrightarrow{W} - \overrightarrow{W}_{s+1}\|_2^2}\|\overrightarrow{W} - \overrightarrow{W}_{s+1}\|_{\nabla^2 \ell_s(W_{s+1})}^2\right).$$

Denote by $\widetilde{Z}_{s+1} = \int_{W \in \mathbb{R}^{Kd}} \widetilde{f}_{s+1}(W)\mathrm{d}W \le +\infty$ the normalization factor and $\widetilde{P}_{s+1}$ the distribution whose density function is $\widetilde{f}_{s+1}(W)/\widetilde{Z}_{s+1}$, we can further rewrite the last line of the above displayed equation (42) as

$$\mathbb{E}_{W \sim P_s}\left[e^{-\ell_s(W)}\right] \ge \frac{e^{-L_s(W_{s+1})}\widetilde{Z}_{s+1}}{Z_s}\mathbb{E}_{W \sim \widetilde{P}_{s+1}}\left[e^{-\langle \nabla L_s(W_{s+1}), \overrightarrow{W} - \overrightarrow{W}_{s+1} \rangle}\right]$$

$$\geq \frac{e^{-L_s(W_{s+1})}\widetilde{Z}_{s+1}}{Z_s} \exp\left(-\mathbb{E}_{W\sim\widetilde{P}_{s+1}}[\langle\nabla L_s(W_{s+1}),\overrightarrow{W}-\overrightarrow{W}_{s+1}\rangle]\right)$$

$$= \frac{e^{-L_s(W_{s+1})}\widetilde{Z}_{s+1}}{Z_s}, \tag{43}$$

where the second inequality is due to the Jensen's inequality and the last equality comes from the fact that $\widetilde{f}_{s+1}(W)/\widetilde{Z}_{s+1}$ is symmetry around $\overrightarrow{W}_{s+1}$ and thus $\mathbb{E}_{W\sim\widetilde{P}_{s+1}}[\langle\nabla L_s(W_{s+1}),\overrightarrow{W}-\overrightarrow{W}_{s+1}\rangle]=0$.

Plugging (43) into (40), we arrive

$$\ell(\widetilde{\mathbf{z}}_s,\mathbf{y}_s) \leq L_s(W_{s+1}) + \ln Z_s - \ln\widetilde{Z}_{s+1}, \tag{44}$$

where we can further convert the last term $-\ln\widetilde{Z}_{s+1}$ to

$$-\ln\widetilde{Z}_{s+1} = -\ln\left(\int_{W\in\mathbb{R}^{Kd}} e^{-\frac{1}{2c}\|\overrightarrow{W}-\overrightarrow{W}_{s+1}\|_{H_s}^2 - e^{6\|\overrightarrow{W}-\overrightarrow{W}_{s+1}\|_2^2}\|\overrightarrow{W}-\overrightarrow{W}_{s+1}\|_{\nabla^2\ell_s(W_{s+1})}^2}\,\mathrm{d}W\right)$$

$$= -\ln\left(\widehat{Z}_{s+1}\cdot\mathbb{E}_{W\sim\widehat{P}_{s+1}}\left[e^{-e^{6\|\overrightarrow{W}-\overrightarrow{W}_{s+1}\|_2^2}\|\overrightarrow{W}-\overrightarrow{W}_{s+1}\|_{\nabla^2\ell_s(W_{s+1})}^2}\right]\right)$$

$$\leq -\ln\widehat{Z}_{s+1} + \mathbb{E}_{W\sim\widehat{P}_{s+1}}\left[e^{6\|\overrightarrow{W}-\overrightarrow{W}_{s+1}\|_2^2}\|\overrightarrow{W}-\overrightarrow{W}_{s+1}\|_{\nabla^2\ell_s(W_{s+1})}^2\right]$$

$$= -\ln Z_s + \underbrace{\mathbb{E}_{W\sim\widehat{P}_{s+1}}\left[e^{6\|\overrightarrow{W}-\overrightarrow{W}_{s+1}\|_2^2}\|\overrightarrow{W}-\overrightarrow{W}_{s+1}\|_{\nabla^2\ell_s(W_{s+1})}^2\right]}_{:=\Omega_s} \tag{45}$$

where $\widehat{P}_{s+1} = \mathcal{N}(\overrightarrow{W}_{s+1}, cH_s^{-1})$ and $\widehat{Z}_{s+1} = \int_{\mathbb{R}^{Kd}} e^{-\frac{1}{2c}\|\overrightarrow{W}-\overrightarrow{W}_{s+1}\|_{H_s}^2}\,\mathrm{d}W$. In above, the first equality is by definition of $\widetilde{Z}_{s+1}$ and the second one comes from the definition of $\widehat{P}_{s+1}$. The last second inequality is due to the Jensen's inequality such that $\ln(\mathbb{E}[a])\geq\mathbb{E}[\ln a]$ for the random variable $a\in\mathbb{R}$. The last equality is by the fact $\widehat{Z}_{s+1} = \int_{\mathbb{R}^{Kd}} e^{-\frac{1}{2c}\|\overrightarrow{W}-\overrightarrow{W}_{s+1}\|_{H_s}^2}\,\mathrm{d}W = Z_s$.

Combining (44) and (45) and taking the summation overt $t$ iterations, we have

$$\sum_{s=1}^{t}\ell(\widetilde{\mathbf{z}}_s,\mathbf{y}_s) = \sum_{s=1}^{t}L_s(W_{s+1}) + \sum_{s=1}^{t}\Omega_s., \tag{46}$$

Following a similar argument in [27], we can bound the last term as follows. Specifically, by the Cauchy-Schwarz inequality, we can decompose the term $\Omega_s$ as

$$\Omega_s \leq \underbrace{\sqrt{\mathbb{E}_{W\sim\widehat{P}_{s+1}}\left[e^{12\|\overrightarrow{W}-\overrightarrow{W}_{s+1}\|_2^2}\right]}}_{\texttt{term (a-1)}}\underbrace{\sqrt{\mathbb{E}_{W\sim\widehat{P}_{s+1}}\left[\|\overrightarrow{W}-\overrightarrow{W}_{s+1}\|_{\nabla^2\ell_s(W_{s+1})}^4\right]}}_{\texttt{term (a-2)}},$$

Then, remarking that $\widehat{P}_{s+1} = \mathcal{N}\left(W_{s+1}, cH_s^{-1}\right)$, the key observation here is that $\overrightarrow{W}-\overrightarrow{W}_{s+1}$ also follows the same distribution as

$$\sum_{i=1}^{Kd}\sqrt{c\lambda_i\left(H_s^{-1}\right)}X_i\mathbf{e}_i, \quad \text{where } X_i \overset{i.i.d.}{\sim} \mathcal{N}(0,1), \ \forall i\in[Kd], \tag{47}$$

where $\{\mathbf{e}_i\in\mathbb{R}^{Kd}\}_{i=1}^{Kd}$ is a set of certain orthogonal basis and $\lambda_i := \lambda_i(H_s^{-1})$ denotes the $i$-th largest eigenvalue of $H_s^{-1}$. Then, we can bound the first term of term (a-1) by

$$\texttt{term (a-1)} \leq \sqrt{\mathbb{E}_{X_i}\left[\Pi_{i=1}^{Kd}e^{12c\lambda_i X_i^2}\right]} \leq \sqrt{\Pi_{i=1}^{Kd}\mathbb{E}_{X_i}\left[e^{\frac{12c}{\lambda}X_i^2}\right]}$$

$$= \left(\mathbb{E}_{X\sim\chi^2}\left[e^{\frac{12c}{\lambda}X}\right]\right)^{\frac{Kd}{2}} \leq \mathbb{E}_{X\sim\chi^2}\left[e^{\frac{6Kdc}{\lambda}X}\right],$$

where $\chi$ is the chi-square distribution and the last second inequality is due to $H_s \succeq \lambda I_{Kd}$ and thus $\lambda_i \leq 1/\lambda$. The last inequality is due to the Jensen's inequality since $Kd > 1$. Then, by further requiring $\lambda \geq 24Kdc$, we have

$$\texttt{term (a-1)} \leq \mathbb{E}_{X\sim\chi^2}\left[e^{\frac{X}{4}}\right] \leq \sqrt{2}, \tag{48}$$

where the last inequality is because the moment-generating function for $\chi^2$-distribution is bounded by $\mathbb{E}_{X \sim \chi^2}\left[e^{tX}\right] \leq 1/\sqrt{1-2t}$ for all $t \leq 1/2$.

Then, we consider the second on the upper bound of term (b). A direct calculation gives,

$$\texttt{term (a-2)} = \mathbb{E}_{W \sim \mathcal{N}\left(\mathbf{0}, \alpha H_s^{-1}\right)}\left[\|(\nabla^2 \ell_s(W_{s+1}))^{\frac{1}{2}}\overrightarrow{W}\|^4\right] = \mathbb{E}_{W \sim \mathcal{N}\left(\mathbf{0}, \alpha \widetilde{H}_s^{-1}\right)}\left[\|\overrightarrow{W}\|_2^4\right],$$

where $\widetilde{H}_s = (\nabla^2 \ell_s(W_{s+1}))^{-\frac{1}{2}} H_s (\nabla^2 \ell_s(W_{s+1}))^{-\frac{1}{2}}$. Then, let $\widetilde{\lambda}_i := \lambda_i\left(c\widetilde{H}_s^{-1}\right)$ be the $i$-th largest eigenvalue of the matrix. A similar decomposition as (47) shows that

$$\texttt{term (a-2)} = \sqrt{\mathbb{E}_{X_i \sim \mathcal{N}(0,1)}\left[\left\|\sum_{i=1}^{Kd}\sqrt{\widetilde{\lambda}_i}X_i \mathbf{e}_i\right\|_2^4\right]} = \sqrt{\mathbb{E}_{X_i \sim \mathcal{N}(0,1)}\left[\left(\sum_{i=1}^{Kd}\widetilde{\lambda}_i X_i^2\right)^2\right]}$$

$$= \sqrt{\sum_{i=1}^{Kd}\sum_{j=1}^{Kd}\widetilde{\lambda}_i\widetilde{\lambda}_j \mathbb{E}_{X_i, X_j \sim \mathcal{N}(0,1)}[X_i^2 X_j^2]} \leq \sqrt{3\sum_{i=1}^{Kd}\sum_{j=1}^{Kd}\widetilde{\lambda}_i\widetilde{\lambda}_j} = \sqrt{3}c\mathrm{Tr}\left(\widetilde{H}_s^{-1}\right),$$

where the last inequality is due to $\mathbb{E}_{X_i, X_j \sim \mathcal{N}(0,1)}[X_i^2 X_j^2] \leq 3$ for all $i, j \in [Kd]$ and the last equality comes from the fact that $\sum_{i=1}^{Kd}\widetilde{\lambda}_i = \mathrm{Tr}\left(c\widetilde{H}_s^{-1}\right)$. The notation $\mathrm{Tr}(A)$ is used to denote the trace of the matrix $A$.

Then, we proceed to bound the trace $\mathrm{Tr}\left(\widetilde{H}_s^{-1}\right)$ with $\widetilde{H}_s = H_s \nabla^2 \ell_s(W_{s+1})$. Recall the definition of $H_s = \lambda I_{Kd} + \sum_{\tau=1}^{s-1}\nabla^2 \ell_\tau(W_{\tau+1})$. We define $M_{s+1} = \lambda I_{Kd}/2 + \sum_{\tau=1}^{s}\nabla^2 \ell_\tau(W_{\tau+1})$. Under the condition $\lambda \geq 2$ for any $s \in [T]$, we have $\nabla^2 \ell_s(W) \preccurlyeq I_{Kd} \leq \frac{\lambda}{2}I_{Kd}$ for any $s \in [T]$ and $W \in \mathcal{W}$. Then, we have $H_s \succcurlyeq M_{s+1}$. Then, we can bound the trace by

$$\mathrm{Tr}\left(\widetilde{H}_s^{-1}\right) = \mathrm{Tr}\left(H_s^{-1}\nabla^2\ell_s(W_{s+1})\right) \leq \mathrm{Tr}\left(M_{s+1}^{-1}\nabla^2\ell_s(W_{s+1})\right)$$

$$= \mathrm{Tr}\left(M_{s+1}^{-1}(M_{s+1} - M_s))\right) \leq \log\frac{|M_{s+1}|}{|M_s|}.$$

where the last inequality is due to Lemma 4.5 of [16]. As a consequence, we have $\texttt{term (a-2)} \leq \sqrt{3}c\ln(|M_{s+1}|/|M_s|)$, which leads to

$$\Omega_s \leq \texttt{term (a-1)} \cdot \texttt{term (a-2)} \leq \sqrt{6}c\ln\left(\frac{|M_{s+1}|}{|M_s|}\right) \tag{49}$$

Combining (46) and (49), we obtain that

$$\sum_{s=1}^{t}\ell(\widetilde{\mathbf{z}}_s, \mathbf{y}_s) - \sum_{s=1}^{t}\ell_s(W_{s+1}) \leq \frac{1}{c}\sum_{s=1}^{t}\|\overrightarrow{W}_s - \overrightarrow{W}_{s+1}\|_{H_s}^2 + \sqrt{6}c\sum_{s=1}^{t}\ln\left(\frac{|M_{s+1}|}{|M_s|}\right).$$

W can further bound the last term of the displayed equality by $\sum_{s=1}^{t}\ln\left(|M_{s+1}|/|M_s|\right) \leq \ln\left(|M_{t+1}|/|\lambda/2 \cdot I_{Kd}|\right) \leq Kd\ln\left(1 + \frac{(t+1)L}{2\lambda}\right)$, which completes the proof. $\square$

### C.1.5 Useful Lemmas

*Proof of Lemma 15.* The proof of Lemma 15 shares the same spirit as that of Lemma 6. Let $\bar{H}_t = \sum_{s=1}^{t-1}\|\mathbf{z}_s\|_{\Sigma_s}^2$. We can also defined the function

$$M_t(\xi) = \exp(\xi S_t - \xi^2 \bar{H}_t)$$

for any $t \geq 1$ and $\xi \in \mathbb{R}$. For $t = 0$, we let $M_0(\xi) = 0$. Following the similar arguments in the proof of Lemma 7, one can show that the sequence $\{M_t(\xi)\}_{t=1}^{\infty}$ is a non-negative super-martingale when $\xi \leq \frac{1}{2}$. Then, let $h(\xi)$ be the density of the normal distribution with precision $2\lambda$ truncated on the 1-dimensional ball $\frac{1}{2}\mathcal{B}(1)$. We can define

$$\bar{M}_t = \int_{\xi}\exp(\xi S_t - \xi^2 \bar{H}_t)\mathrm{d}h(\xi).$$

By the similar arguments in deriving (17), we have

$$\Pr\left[\sup_{t\in\mathbb{N}}\log(\bar{M}_t)\geq\log\left(\frac{1}{\delta}\right)\right]=\Pr\left[\sup_{t\in\mathbb{N}}\bar{M}_t\geq\frac{1}{\delta}\right]\leq\delta.\qquad(50)$$

Following the arguments in the proof of [8, Theorem 1], for any $t\geq 1$, we have

$$\bar{M}_t\geq\exp(\xi S_t-\xi^2 H_t)\cdot\frac{N(g)}{N(h)}$$

for any $|\xi|\leq 1/4$. In the above, $g(\xi)$ is the normal distribution with precision $2H_t$ truncated on the interval $[-1/4,1/4]$. Then, let $\xi_0=\frac{1}{4}\sqrt{\frac{\lambda}{H_t}}$, we have $\xi_0\leq 1/4$ since $H_t\geq\lambda$. We can obtain that

$$\log(\bar{M}_t)\geq\xi_0 S_t-\xi_0^2 H_t+\log\left(\frac{N(g)}{N(h)}\right)=\frac{\sqrt{\lambda}}{4}\cdot\frac{S_t}{\sqrt{H_t}}-\frac{\lambda}{16}+\log\left(\frac{N(g)}{N(h)}\right).\qquad(51)$$

A combination of (50) and (51) shows that

$$\Pr\left[\forall t\geq 1,\frac{\sqrt{\lambda}}{4}\cdot\frac{S_t}{\sqrt{H_t}}-\frac{\lambda}{16}+\log\left(\frac{N(g)}{N(h)}\right)\leq\log\left(\frac{1}{\delta}\right)\right]\geq 1-\delta,$$

which indicates

$$\Pr\left[\forall t\geq 1, S_t\leq\sqrt{H_t}\left(\frac{\sqrt{\lambda}}{4}+\frac{4}{\sqrt{\lambda}}\log\left(\frac{N(h)}{\delta N(g)}\right)\right)\right]\geq 1-\delta.$$

We complete the proof with Lemma 6 of [8], which shows that $\log\left(N(h)/N(g)\right)\leq\log\left((2\sqrt{H_t})/\lambda\right)$.
$\square$

**Lemma 16** (Proposition 4.1 of [49]). *Let the $\mathbf{w}_{t+1}$ be the solution of the update rule*

$$\mathbf{w}_{t+1}=\arg\min_{\mathbf{w}\in\mathcal{V}}\eta\ell_t(\mathbf{w})+\mathcal{D}_\psi(\mathbf{w},\mathbf{w}_t),$$

*where $\mathcal{V}\subseteq\mathcal{W}\subseteq\mathbb{R}^d$ is a non-empty convex set and $\mathcal{D}_\psi(\mathbf{w}_1,\mathbf{w}_2)=\psi(\mathbf{w}_1)-\psi(\mathbf{w}_2)-\langle\nabla\psi(\mathbf{w}_2),\mathbf{w}_1-\mathbf{w}_2\rangle$ is the Bregman Divergence w.r.t. a strictly convex and continuously differentiable function $\psi:\mathcal{W}\mapsto\mathbb{R}$. Further supposing $\psi(\mathbf{w})$ is 1-strongly convex w.r.t. a certain norm $\|\cdot\|$ in $\mathcal{W}$, then there exists a $\mathbf{g}_t'\in\partial\ell_t(\mathbf{w}_{t+1})$ such that*

$$\langle\eta_t\mathbf{g}_t',\mathbf{w}_{t+1}-\mathbf{u}\rangle\leq\langle\nabla\psi(\mathbf{w}_t)-\nabla\psi(\mathbf{w}_{t+1}),\mathbf{w}_{t+1}-\mathbf{u}\rangle$$

*for any $\mathbf{u}\in\mathcal{W}$.*

**Lemma 17.** *Let $\ell(\mathbf{z},y)=\sum_{k=0}^K\mathbb{1}\{y=k\}\cdot\log\left(\frac{1}{[\boldsymbol{\sigma}(\mathbf{z})]_k}\right)$ and $\mathbf{z}\in\mathbb{R}^K$ be a $K$-dimensional vector. Define $\mathbf{z}^\mu\triangleq\boldsymbol{\sigma}^+\left(\mathrm{smooth}_\mu(\boldsymbol{\sigma}(\mathbf{z}))\right)$, where $\mathrm{smooth}_\mu(\mathbf{p})=(1-\mu)\mathbf{p}+\mu\mathbf{1}/(K+1)$. Then, for $\mu\in[0,1/2]$, we have*

$$\ell(\mathbf{z}^\mu,y)-\ell(\mathbf{z},y)\leq 2\mu$$

*for any $y\in\{0\}\cup[K]$. We also have $\|\mathbf{z}^\mu\|_\infty\leq\log(K/\mu)$.*

*Proof of Lemma 17.* The proof of Lemma 17 is extracted from the proof of [26, Lemma 3] with a slight modification to the logistic loss used in this paper. According to the definition of $\mathbf{z}^\mu$ and the fact that $\boldsymbol{\sigma}(\boldsymbol{\sigma}^+(\mathbf{p}))=\mathbf{p}$ for any $\mathbf{p}\in\{\mathbf{q}\in[0,1]^K\|\|\mathbf{q}\|_1<1\}$, we have $\boldsymbol{\sigma}(\mathbf{z}^\mu)=(1-\mu)\boldsymbol{\sigma}(\mathbf{z})+\mu\mathbf{1}/(K+1)$. Then, we have

$$\ell(\mathbf{z}^\mu,y)-\ell(\mathbf{z},y)=\sum_{k=0}^K\mathbb{1}\{y=k\}\cdot\log\left(\frac{[\boldsymbol{\sigma}(\mathbf{z})]_k}{(1-\mu)[\boldsymbol{\sigma}(\mathbf{z})]_k+\mu\mathbf{1}/(K+1)}\right)\leq\log(1-\mu)\leq 2\mu,$$

where the first inequality is due to $\sum_{k=0}^K\mathbb{1}\{y=k\}=1$ and the last inequality is due to $\log(1/(1-a))\leq a$ for $a\in[0,1/2]$. Besides, let $\widetilde{\mathbf{p}}=\boldsymbol{\sigma}(\mathbf{z})$. According to the definition of $\mathbf{z}^\mu$, we have

$$\|\mathbf{z}^\mu\|_\infty=\max_{k\in[K]}\left\{\left|\log\left(\frac{(1-\mu)\widetilde{p}_k+\mu/(K+1)}{(1-K\mu/(K+1))-(1-\mu)\|\widetilde{\mathbf{p}}\|_1}\right)\right|\right\}.$$

Since the term inside the logarithmic function can be bounded by

$$\frac{\mu}{1 + K(1 - \mu)} \leq \log\left(\frac{(1 - \mu)\widetilde{p}_k + \mu/(K+1)}{(1 - K\mu/(K+1)) - (1 - \mu)\|\widetilde{\mathbf{p}}\|_1}\right) \leq 1 + \frac{K+1}{\mu}$$

and $\mu/(1 + K(1 - \mu)) \leq 1$, we have $\|\mathbf{z}^\mu\|_\infty \leq \log(1 + (K+1)/\mu)$.

$\square$

**Lemma 18.** *Let $L_s(W) = \ell_s(W) + \frac{1}{2c}\|\overrightarrow{W} - \overrightarrow{W}_s\|_{H_s}^2$. Then, for any $W, W_{s+1} \in \mathcal{W}$, the quadratic approximation $\widetilde{L}_s(W) = L_s(W_{s+1}) + \langle\nabla L_s(W_{s+1}), \overrightarrow{W} - \overrightarrow{W}_{s+1}\rangle + \frac{1}{2c}\|\overrightarrow{W} - \overrightarrow{W}_{s+1}\|_{H_s}^2$ defined as* (41) *satisfies*

$$L_s(W) \leq \widetilde{L}_s(W) + e^{6\|\overrightarrow{W} - \overrightarrow{W}_{s+1}\|_2^2}\|\overrightarrow{W} - \overrightarrow{W}_{s+1}\|_{\nabla\ell_s(W_{s+1})}^2.$$

*Proof of Lemma 18.* According to [19, Lemma 4], $\ell_s$ is a $\sqrt{6}$-self-concordant-like function. Then, by Lemma 5 and the fact $c(x) \leq e^{x^2}$ for any $x \geq 0$, then for any $W \in \mathcal{W}$, we have

$$\ell_s(W) \leq \ell_s(W_{s+1}) + \langle\nabla\ell_s(W_{s+1}), \overrightarrow{W} - \overrightarrow{W}_{s+1}\rangle + e^{6\|\overrightarrow{W} - \overrightarrow{W}_{s+1}\|_2^2}\|\overrightarrow{W} - \overrightarrow{W}_{s+1}\|_{\nabla^2\ell_s(W_{s+1})}^2. \tag{52}$$

Besides, since $g_s(W) = \frac{1}{2c}\|\overrightarrow{W} - \overrightarrow{W}_s\|_{H_s}^2$ is a quadratic function, we have

$$g_s(W) = g_s(W_{s+1}) + \langle\nabla g_s(W_{s+1}), \overrightarrow{W} - \overrightarrow{W}_{s+1}\rangle + \frac{1}{2c}\|\overrightarrow{W} - \overrightarrow{W}_{s+1}\|_{H_s}^2. \tag{53}$$

Then combing (52) and (53), we can obtain an upper bound of $L_s(W) = \ell_s(W) + g_s(W)$ by

$$\begin{aligned}
L_s(W) &\leq L_s(W_{s+1}) + \langle\nabla L_s(W_{s+1}), \overrightarrow{W} - \overrightarrow{W}_{s+1}\rangle \\
&\quad + \frac{1}{2c}\|\overrightarrow{W} - \overrightarrow{W}_{s+1}\|_{H_s}^2 + e^{6\|\overrightarrow{W} - \overrightarrow{W}_{s+1}\|_2^2}\|\overrightarrow{W} - \overrightarrow{W}_{s+1}\|_{\nabla^2\ell_s(W_{s+1})}^2 \\
&\leq \widetilde{L}_s(W) + e^{6\|\overrightarrow{W} - \overrightarrow{W}_{s+1}\|_2^2}\|\overrightarrow{W} - \overrightarrow{W}_{s+1}\|_{\nabla^2\ell_s(W_{s+1})}^2,
\end{aligned}$$

We have complete the proof.

$\square$

## C.2 Proof of Proposition 1

This section presents the proof of Proposition 1.

### C.2.1 Main Proof

*Proof of Proposition 1.* Since $\boldsymbol{\rho}^\top\boldsymbol{\sigma}(W_*\mathbf{x}) \leq \boldsymbol{\rho}^\top\boldsymbol{\sigma}(W_t\mathbf{x}) + |\boldsymbol{\rho}^\top\boldsymbol{\sigma}(W_*\mathbf{x}) - \boldsymbol{\rho}^\top\boldsymbol{\sigma}(W_t\mathbf{x})|$, to show $\widetilde{r}_t^{\text{OL}}(\mathbf{x}) = \boldsymbol{\rho}^\top\boldsymbol{\sigma}(W_t\mathbf{x}) + \epsilon_t^{\text{fst}}(\mathbf{x}) + \epsilon_t^{\text{snd}}(\mathbf{x})$ is an optimistic estimate, it is sufficient to prove that $|\boldsymbol{\rho}^\top\boldsymbol{\sigma}(W_*\mathbf{x}) - \boldsymbol{\rho}^\top\boldsymbol{\sigma}(W_t\mathbf{x})| \leq \epsilon_t^{\text{fst}}(\mathbf{x}) + \epsilon_t^{\text{snd}}(\mathbf{x})$. We have the following decomposition:

$$\begin{aligned}
&\left|\boldsymbol{\rho}^\top\boldsymbol{\sigma}(W_*\mathbf{x}) - \boldsymbol{\rho}^\top\boldsymbol{\sigma}(W_t\mathbf{x})\right| \\
&= \left|\sum_{k=1}^K \rho_k(\sigma_k(W_*\mathbf{x}) - \sigma_k(W_t\mathbf{x}))\right| \\
&= \left|\sum_{k=1}^K \rho_k\nabla\sigma_k(W_t\mathbf{x})^\top(W_* - W_t)\mathbf{x} + \sum_{k=1}^K \rho_k\|(W_* - W_t)\mathbf{x}\|_{\Xi_{k,t}}^2\right| \\
&\leq \underbrace{\left|\sum_{k=1}^K \rho_k\nabla\sigma_k(W_t\mathbf{x})^\top(W_* - W_t)\mathbf{x}\right|}_{\texttt{term (a)}} + \underbrace{\left|\sum_{k=1}^K \rho_k\|(W_* - W_t)\mathbf{x}\|_{\Xi_{k,t}}^2\right|}_{\texttt{term (b)}},
\end{aligned}$$

where $\sigma_k : \mathbf{x} \mapsto [\boldsymbol{\sigma}(\mathbf{x})]_k$ is the $k$-th output of the vector-valued function $\boldsymbol{\sigma}(\mathbf{x})$ and $\Xi_{k,t} = \int_{\nu=0}^{1}(1 - \nu)\nabla^2\sigma_k((W_t + \nu(W_* - W_t))\mathbf{x})\mathrm{d}\nu$. In the above, the last equality is due to the integral formulation of the Taylor series.

Then, we proceed to analyze term (a) and term (b), respectively.

$$
\begin{aligned}
\texttt{term (a)} &= |\boldsymbol{\rho}^\top \nabla\boldsymbol{\sigma}(W_t\mathbf{x})(W_* - W_t)\mathbf{x}| \\
&= |\boldsymbol{\rho}^\top \nabla\boldsymbol{\sigma}(W_t\mathbf{x})(I_K \otimes \mathbf{x}^\top)(\overrightarrow{W}_* - \overrightarrow{W}_t)| \\
&= |\boldsymbol{\rho}^\top \nabla\boldsymbol{\sigma}(W_t\mathbf{x})(I_K \otimes \mathbf{x}^\top)H_t^{-\frac{1}{2}}H_t^{\frac{1}{2}}(\overrightarrow{W}_* - \overrightarrow{W}_t)| \\
&\leq \|\overrightarrow{W}_* - \overrightarrow{W}_t\|_{H_t} \cdot \|H_t^{-\frac{1}{2}}(I_K \otimes \mathbf{x})\nabla\boldsymbol{\sigma}(W_t\mathbf{x})\boldsymbol{\rho}\|_2 \\
&\leq \beta_t^{\texttt{OL}}(\delta) \cdot \|H_t^{-\frac{1}{2}}(I_K \otimes \mathbf{x})\nabla\boldsymbol{\sigma}(W_t\mathbf{x})\boldsymbol{\rho}\|_2 \\
&= \epsilon_t^{\texttt{fst}}(\mathbf{x}),
\end{aligned}
$$

where the last inequality is due to Theorem 3.

Then, we upper bound term (b) by $\epsilon_t^{\texttt{snd}}$ with the following arguments. For notation simplicity, we denote by $\boldsymbol{\xi}_{t,\nu} = W_t\mathbf{x} + \nu(W_* - W_t)\mathbf{x}$. Then, Lemma 11 indicates that

$$
\Xi_{k,t} = \int_{\nu=0}^{1}(1 - \nu)\nabla^2\sigma_k(\boldsymbol{\xi}_{t,\nu})\mathrm{d}\nu \preccurlyeq 3I_K \int_{\nu=0}^{1}(1 - \nu)\mathrm{d}\nu \preccurlyeq 3I_K.
$$

As a consequence, we can bound term (b) by

$$
\begin{aligned}
\texttt{term (b)} &\leq 3\left|\sum_{k=1}^{K}\rho_k\|(W_* - W_t)\mathbf{x}\|_2^2\right| \\
&\leq 3R\|(W_* - W_t)\mathbf{x}\|_2^2 \\
&= 3R\|(I_d \otimes \mathbf{x}^\top)(\overrightarrow{W}_* - \overrightarrow{W}_t)\|_2^2 \\
&= \leq 3R\|\overrightarrow{W}_* - \overrightarrow{W}_t\|_{H_t}^2 \cdot \|(I_K \otimes \mathbf{x}^\top)H_t^{-\frac{1}{2}}\|_2^2 \\
&\leq 3R\left(\beta_t^{\texttt{OL}}\right)^2 \cdot \|(I_K \otimes \mathbf{x}^\top)H_t^{-\frac{1}{2}}\|_2^2 \\
&= \epsilon_t^{\texttt{snd}}(\mathbf{x}),
\end{aligned}
$$

where the first inequality is due to Assumption 2 and the last inequality is due to Theorem 3. Combining the upper bound for term (a) and term (b), we have

$$
|\boldsymbol{\rho}^\top\boldsymbol{\sigma}(W_*\mathbf{x}) - \boldsymbol{\rho}^\top\boldsymbol{\sigma}(W_t\mathbf{x})| \leq \epsilon_t^{\texttt{fst}}(\mathbf{x}) + \epsilon_t^{\texttt{snd}}(\mathbf{x}), \tag{54}
$$

for any $\mathbf{x} \in \mathcal{X}$. We complete the proof by $\boldsymbol{\rho}^\top\boldsymbol{\sigma}(W_*\mathbf{x}) \leq \boldsymbol{\rho}^\top\boldsymbol{\sigma}(W_t\mathbf{x}) + |\boldsymbol{\rho}^\top\boldsymbol{\sigma}(W_*\mathbf{x}) - \boldsymbol{\rho}^\top\boldsymbol{\sigma}(W_t\mathbf{x})|$.

$\square$

### C.3 Proof of Theorem 4

*Proof of Theorem 4.* We first prove the regret bound and then discuss the computation cost.

#### C.3.1 Main Proof

**Regret Analysis.** We can bound the regret by two times of the bonus term over $\mathbf{x}_t$.

$$
\begin{aligned}
\text{Reg}_T &= \sum_{t=1}^{T}\left(\boldsymbol{\rho}^\top\boldsymbol{\sigma}(W_*\mathbf{x}_*) - \boldsymbol{\rho}^\top\boldsymbol{\sigma}(W_*\mathbf{x}_t)\right) \\
&\leq \sum_{t=1}^{T}\left(\boldsymbol{\rho}^\top\boldsymbol{\sigma}(W_t\mathbf{x}_*) + \epsilon_t^{\texttt{fst}}(\mathbf{x}_*) + \epsilon_t^{\texttt{snd}}(\mathbf{x}_*) - \boldsymbol{\rho}^\top\boldsymbol{\sigma}(W_*\mathbf{x}_t)\right) \\
&\leq \sum_{t=1}^{T}\left(\boldsymbol{\rho}^\top\boldsymbol{\sigma}(W_t\mathbf{x}_t) + \epsilon_t^{\texttt{fst}}(\mathbf{x}_t) + \epsilon_t^{\texttt{snd}}(\mathbf{x}_t) - \boldsymbol{\rho}^\top\boldsymbol{\sigma}(W_*\mathbf{x}_t)\right)
\end{aligned}
$$

$$\leq 2\sum_{t=1}^{T}\epsilon_t^{\texttt{fst}}(\mathbf{x}_t) + 2\sum_{t=1}^{T}\epsilon_t^{\texttt{snd}}(\mathbf{x}_t), \tag{55}$$

where the first inequality is due to Proposition 1 and the second inequality is due to the rule of constructing the optimistic reward. The last inequality is due to (54).

Then, we turn to analyze the upper bound for the first term and second term respectively.

$$\sum_{t=1}^{T}\epsilon_t^{\texttt{fst}}(\mathbf{x}_t)$$

$$= \sum_{t=1}^{T}\beta_t^{\texttt{OL}}(\delta) \cdot \|H_t^{-\frac{1}{2}}(I_K \otimes \mathbf{x}_t)\nabla\boldsymbol{\sigma}(W_t\mathbf{x}_t)\boldsymbol{\rho}\|_2$$

$$\leq \beta_T^{\texttt{OL}}(\delta)\sum_{t=1}^{T}\|H_t^{-\frac{1}{2}}(I_K \otimes \mathbf{x}_t)\nabla\boldsymbol{\sigma}(W_t\mathbf{x}_t)\boldsymbol{\rho}\|_2$$

$$\leq \beta_T^{\texttt{OL}}(\delta)\underbrace{\sum_{t=1}^{T}\|H_t^{-\frac{1}{2}}(I_K \otimes \mathbf{x}_t)\nabla\boldsymbol{\sigma}(W_{t+1}\mathbf{x}_t)\boldsymbol{\rho}\|_2}_{\texttt{term (a)}} + \beta_T^{\texttt{OL}}(\delta)\underbrace{\sum_{t=1}^{T}\|H_t^{-\frac{1}{2}}(I_K \otimes \mathbf{x}_t)\left(\nabla\boldsymbol{\sigma}(W_t\mathbf{x}_t) - \nabla\boldsymbol{\sigma}(W_{t+1}\mathbf{x}_t)\right)\boldsymbol{\rho}\|_2}_{\texttt{term (b)}}.$$

$$\tag{56}$$

For the first term, we have

$$\texttt{term (a)} \leq \sum_{t=1}^{T}\|\boldsymbol{\rho}\|_{\nabla\boldsymbol{\sigma}(W_{t+1}\mathbf{x}_t)} \cdot \|H_t^{-\frac{1}{2}}(I_K \otimes \mathbf{x}_t)\nabla\boldsymbol{\sigma}^{\frac{1}{2}}(W_{t+1}\mathbf{x})\|_2$$

$$\leq R\sum_{t=1}^{T}\sqrt{\lambda_{\max}\left((\nabla\boldsymbol{\sigma}^{\frac{1}{2}}(W_{t+1}\mathbf{x}) \otimes \mathbf{x}_t^\top)H_t^{-1}(\nabla\boldsymbol{\sigma}^{\frac{1}{2}}(W_{t+1}\mathbf{x}) \otimes \mathbf{x}_t)\right)}$$

$$= R\sum_{t=1}^{T}\sqrt{\lambda_{\max}\left((\nabla\boldsymbol{\sigma}(W_{t+1}\mathbf{x}_t) \otimes \mathbf{x}_t\mathbf{x}_t^\top)H_t^{-1}\right)}$$

$$\leq R\sqrt{T}\sqrt{\sum_{t=1}^{T}\lambda_{\max}\left((\nabla\boldsymbol{\sigma}(W_{t+1}\mathbf{x}_t) \otimes \mathbf{x}_t\mathbf{x}_t^\top)H_t^{-1}\right)}$$

$$\leq R\sqrt{T}\sqrt{Kd\ln\left(1 + \frac{TL}{2\lambda}\right)}$$

where the first inequality is due to the fact that $\|A\mathbf{b}\|_2 \leq \|A\|_2 \cdot \|\mathbf{b}\|$ for a matrix $A$ and vector $\mathbf{b}$. The second inequality is due to the definition of the induced norm for the matrix is defined as $\|A\|_2 = \sqrt{\lambda_{\max}(A^T A)}$ and the mixed-product property of the Kronecker production. The first equality is due to the cycle property of the maximum eigenvalue such that $\lambda_{\max}(ABC) = \lambda_{\max}(CAB)$ for matrices $A, B$ and $C$. The last second inequality is due to the Cauchy-Schwarz inequality. Finally, we can obtain the last inequality by Lemma 19, which can be seen as a matrix version of the elliptical potential lemma.

Then, we can bound term (b) as follows.

$$\texttt{term (b)} \leq \sum_{t=1}^{T}\|H_t^{-\frac{1}{2}}(I_K \otimes \mathbf{x}_t)\|_2 \cdot \|(\nabla\boldsymbol{\sigma}(W_t\mathbf{x}_t) - \nabla\boldsymbol{\sigma}(W_{t+1}\mathbf{x}_t))\boldsymbol{\rho}\|_2$$

$$= \sum_{t=1}^{T}\|H_t^{-\frac{1}{2}}(I_K \otimes \mathbf{x}_t)\|_2 \cdot \left\|\sum_{k=1}^{K}\rho_k(\nabla\sigma_k(W_t\mathbf{x}_t) - \nabla\sigma_k(W_{t+1}\mathbf{x}_t))\right\|_2$$

$$= \sum_{t=1}^{T}\|H_t^{-\frac{1}{2}}(I_K \otimes \mathbf{x}_t)\|_2 \cdot \left\|\sum_{k=1}^{K}\rho_k\nabla^2\sigma_k(\boldsymbol{\xi}_{t,k})(W_t - W_{t+1})\mathbf{x}_t\right\|_2$$

$$= \sum_{t=1}^{T} \|H_t^{-\frac{1}{2}}(I_K \otimes \mathbf{x}_t)\|_2 \cdot \left\|\sum_{k=1}^{K} \rho_k \nabla^2 \sigma_k(\boldsymbol{\xi}_{t,k})(I_K \otimes \mathbf{x}_t)^\top (\overrightarrow{W}_t - \overrightarrow{W}_{t+1})\right\|_2 \tag{57}$$

In the above, the second equality is due to the mean-value theorem, where denote by $\boldsymbol{\xi}_{t,k} \in \mathbb{R}^K$ a certain point on the line connecting $W_t \mathbf{x}_t$ and $W_{t+1}\mathbf{x}_t$. We can further bound the second term of the right hand side of the above displayed inequality by

$$\left\|\sum_{k=1}^{K} \rho_k \nabla^2 \sigma_k(\boldsymbol{\xi}_{t,k})(I_K \otimes \mathbf{x}_t^\top)(\overrightarrow{W}_t - \overrightarrow{W}_{t+1})\right\|_2$$

$$= \left\|\sum_{k=1}^{K} \rho_k \nabla^2 \sigma_k(\boldsymbol{\xi}_{t,k})(I_K \otimes \mathbf{x}_t^\top)H_t^{-\frac{1}{2}} H_t^{\frac{1}{2}}(\overrightarrow{W}_t - \overrightarrow{W}_{t+1})\right\|_2$$

$$\leq \left\|\sum_{k=1}^{K} \rho_k \nabla^2 \sigma_k(\boldsymbol{\xi}_{t,k})(I_K \otimes \mathbf{x}_t^\top)H_t^{-\frac{1}{2}}\right\|_2 \cdot \|\overrightarrow{W}_t - \overrightarrow{W}_{t+1}\|_{H_t}$$

$$\leq \frac{\alpha\sqrt{K}}{\lambda} \left\|\sum_{k=1}^{K} \rho_k \nabla^2 \sigma_k(\boldsymbol{\xi}_{t,k})(I_K \otimes \mathbf{x}_t^\top)H_t^{-\frac{1}{2}}\right\|_2$$

$$\leq \frac{\alpha\sqrt{K}}{\lambda} \sum_{k=1}^{K} \rho_k \|\nabla^2 \sigma_k(\boldsymbol{\xi}_{t,k})\|_2 \cdot \left\|(I_K \otimes \mathbf{x}_t^\top)H_t^{-\frac{1}{2}}\right\|_2$$

$$\leq \frac{3\alpha K R}{\lambda} \left\|(I_K \otimes \mathbf{x}_t^\top)H_t^{-\frac{1}{2}}\right\|_2 \tag{58}$$

where the first inequality is by the fact $\|A\mathbf{b}\|_2 \leq \|A\|_2 \cdot \|\mathbf{b}\|_2$ for any matrix $A$ and vector $\mathbf{b}$. The second inequality is due to Lemma 20. The last inequality is due to Lemma 11 such that $\nabla \sigma_k(\mathbf{z}) \preccurlyeq 3I_k$ for any $\mathbf{z} \in \mathbb{R}^K$ and $\sum_{k=1}^{K} \rho_k \leq \sqrt{K}R$.

Then plugging (58) into (57), we can bound term (b) by

$$\texttt{term (b)} \leq \frac{3\alpha K R}{\lambda} \sum_{t=1}^{T} \|H_t^{-\frac{1}{2}}(I_K \otimes \mathbf{x}_t)\|_2^2 \leq \frac{3\kappa K d R \alpha}{\lambda} \ln\left(1 + \frac{T}{\lambda\kappa}\right),$$

where the last inequality is due to Lemma 21. Combining the upper bound for term (a) and term (b) and plugging them into (56), we have

$$\sum_{t=1}^{T} \epsilon_t^{\texttt{fst}}(\mathbf{x}_t) \leq \beta_T^{\texttt{OL}}(\delta)\left(R\sqrt{T}\sqrt{Kd\ln\left(1 + \frac{TL}{2\lambda}\right)}\right) + \frac{3\kappa K d R \alpha \beta_T^{\texttt{OL}}(\delta)}{\lambda} \ln\left(1 + \frac{T}{\lambda\kappa}\right)$$

$$= \mathcal{O}\left(Kd\log K(\log T)^{\frac{3}{2}}\sqrt{T} + \kappa K^{\frac{3}{2}}d^{\frac{3}{2}}(\log K)^2(\log T)^2\right). \tag{59}$$

As for the term $\sum_{t=1}^{T} \epsilon_t^{\texttt{snd}}(\mathbf{x}_t)$, we have

$$\sum_{t=1}^{T} \epsilon_t^{\texttt{snd}}(\mathbf{x}_t) = 3R\left(\beta_t^{\texttt{OL}}\right)^2 \cdot \|(I_K \otimes \mathbf{x}_t^\top)H_t^{-\frac{1}{2}}\|_2^2$$

$$\leq 3R\left(\beta_t^{\texttt{OL}}\right)^2 \kappa d \ln\left(1 + \frac{T}{\lambda\kappa}\right)$$

$$= \mathcal{O}\left(\kappa K d^2(\log K)^2(\log T)^3\right). \tag{60}$$

Combining (59) and (60) with (55), we have

$$\text{Reg}_T \leq 2\sum_{t=1}^{T} \epsilon_t^{\texttt{fst}}(\mathbf{x}_t) + 2\sum_{t=1}^{T} \epsilon_t^{\texttt{snd}}(\mathbf{x}_t)$$

$$\leq 2\beta_T^{\texttt{OL}}(\delta)R\sqrt{T}\sqrt{d\ln\left(1 + \frac{TL}{2\lambda}\right)} + \frac{6\kappa K d R \alpha \beta_T^{\texttt{OL}}(\delta)}{\lambda} \ln\left(1 + \frac{T}{\lambda\kappa}\right)$$

$$+ 6R \left( \beta_t^{\text{OL}} \right)^2 \kappa d \ln \left( 1 + \frac{T}{\lambda \kappa} \right)$$

$$= \mathcal{O} \left( Kd \log K (\log T)^{\frac{3}{2}} \sqrt{T} + \kappa K^{\frac{3}{2}} d^2 (\log K)^2 (\log T)^3 \right).$$

which completes the proof for the regret bound. $\qquad\square$

### C.3.2 Additional Guarantee for Algorithm 2

This part shows that the OFUL-MLogB (Algorithm 2) also achieves an $\widetilde{\mathcal{O}}(\sqrt{\kappa K T})$ regret bound.

**Theorem 5.** *Under the same condition as Theorem 3, Algorithm 2 ensures*

$$\text{Reg}_T \leq \mathcal{O} \left( d \log K (\log T)^{\frac{3}{2}} \sqrt{\kappa K T} + \kappa K^{\frac{3}{2}} d^2 (\log K)^2 (\log T)^3 \right).$$

*Proof of Theorem 5.* The proof of Theorem 5 is almost the same as that for Theorem 4. We can decompose the regret as

$$\text{Reg}_T \leq 2 \sum_{t=1}^{T} \epsilon_t^{\text{fst}}(\mathbf{x}_t) + \sum_{t=1}^{T} \epsilon_t^{\text{snd}}(\mathbf{x}_t)$$

following the same arguments (55). The only difference is that we can bound the $\sum_{t=1}^{T} \epsilon_t^{\text{fst}}(\mathbf{x}_t)$ in (55) with a different arguments based on the local matrix $\bar{V}_t \triangleq \lambda I_{Kd} + \frac{1}{\kappa} \sum_{s=1}^{t-1} I_K \otimes \mathbf{x}_s \mathbf{x}_s^\top$.

$$\sum_{t=1}^{T} \epsilon_t^{\text{fst}}(\mathbf{x}_t)$$

$$= \sum_{t=1}^{T} \beta_t^{\text{OL}}(\delta) \cdot \| H_t^{-\frac{1}{2}} (I_K \otimes \mathbf{x}_t) \nabla \boldsymbol{\sigma}(W_t \mathbf{x}_t) \boldsymbol{\rho} \|_2$$

$$\leq R \beta_T^{\text{OL}}(\delta) \sum_{t=1}^{T} \| \bar{V}_t^{-\frac{1}{2}} (I_K \otimes \mathbf{x}_t) \|_2$$

$$\leq R \beta_T^{\text{OL}}(\delta) \sqrt{\kappa d T \log \left( 1 + \frac{T}{\kappa \lambda d} \right)}$$

$$= \mathcal{O}(d \log K (\log T)^{\frac{2}{3}} \sqrt{\kappa K T}) \tag{61}$$

where the first inequality is by the fact that $H_t \succcurlyeq \bar{V}_t$, $\nabla \boldsymbol{\sigma}(W_t \mathbf{x}_t) \preccurlyeq I_K$, and $\|\boldsymbol{\rho}\|_2 \leq R$. The second inequality can be obtained $\epsilon_t^{\text{snd}}$ with the same arguments as (60):

$$\sum_{t=1}^{T} \epsilon_t^{\text{snd}}(\mathbf{x}) \leq \mathcal{O} \left( \kappa K d^2 (\log K)^2 (\log T)^3 \right).$$

We completed the proof by combining the above three displayed bounds. $\qquad\square$

### C.3.3 Useful Lemmas

**Lemma 19.** *Let $H_t = \lambda I_{Kd} + \sum_{s=1}^{t-1} \nabla \boldsymbol{\sigma}(W_{s+1} \mathbf{x}_s) \otimes \mathbf{x}_s \mathbf{x}_s^\top$. Then, when $\lambda > 2$, we have*

$$\sum_{t=1}^{T} \lambda_{\max} \left( (\nabla \boldsymbol{\sigma}(W_{t+1} \mathbf{x}_t) \otimes \mathbf{x}_t \mathbf{x}_t^\top) H_t^{-1} \right) \leq K d \ln \left( 1 + \frac{TL}{2\lambda} \right),$$

*where $L = \max_{\mathbf{x} \in \mathcal{X}, W \in \mathcal{W}} \lambda_{\max}(\nabla \boldsymbol{\sigma}(W \mathbf{x}))$*

*Proof of Lemma 19.* Let $M_t = \frac{\lambda}{2} I_{Kd} + \sum_{s=1}^{t} \nabla \boldsymbol{\sigma}(W_{s+1} \mathbf{x}_s) \otimes \mathbf{x}_s \mathbf{x}_s^\top$. Then, we have $H_t - M_t = \frac{\lambda}{2} I_{Kd} - \nabla \boldsymbol{\sigma}(W_{t+1} \mathbf{x}_t) \otimes \mathbf{x}_t \mathbf{x}_t^\top$. Since $\nabla \boldsymbol{\sigma}(W_{t+1} \mathbf{x}_t) \otimes \mathbf{x}_t \mathbf{x}_t^\top \preccurlyeq I_{Kd}$, we have $H_t \succcurlyeq M_t$ when $\lambda > 2$. Then, we have

$$\sum_{t=1}^{T} \lambda_{\max} \left( (\nabla \boldsymbol{\sigma}(W_{t+1} \mathbf{x}_t) \otimes \mathbf{x}_t \mathbf{x}_t^\top) H_t^{-1} \right)$$

$$\leq \sum_{t=1}^{T} \mathrm{Tr}\left(\left(\nabla\boldsymbol{\sigma}(W_{t+1}\mathbf{x}_t) \otimes \mathbf{x}_t\mathbf{x}_t^\top\right)H_t^{-1}\right)$$

$$\leq \sum_{t=1}^{T} \mathrm{Tr}\left(M_t - M_{t-1})M_t^{-1}\right)$$

$$\leq \sum_{t=1}^{T} \log\frac{|M_t|}{|M_{t-1}|}$$

$$\leq Kd\ln\left(1 + \frac{TL}{2\lambda}\right)$$

where we denote by $\mathrm{Tr}(A)$ the trace of matrix A. The second inequality is by the condition $H_t \succcurlyeq M_t$. The last second inequality is due to Lemma 4.5 of [16].

$\square$

**Lemma 20.** *Let* $\overrightarrow{W}_{t+1} = \arg\min_{W\in\mathcal{W}}\langle\nabla\ell_t(\overrightarrow{W}_t), \overrightarrow{W}\rangle + \frac{1}{\alpha}\|\overrightarrow{W} - \overrightarrow{W}_t\|_{\widetilde{H}_t}^2$. *Then, we have*

$$\|\overrightarrow{W}_{t+1} - \overrightarrow{W}_t\|_{H_t} \leq \frac{\alpha}{2\lambda}\|\nabla\ell_t(\overrightarrow{W}_t)\|_2 \leq \frac{\alpha\sqrt{K}}{\lambda}.$$

*Proof of Lemma 20.* Let $F_t(\overrightarrow{W}) = \langle\nabla\ell_t(\overrightarrow{W}_t), \overrightarrow{W}\rangle + \frac{1}{\alpha}\|\overrightarrow{W} - \overrightarrow{W}_t\|_{\widetilde{H}_t}^2$. Since $\overrightarrow{W}_{t+1} = \arg\min_{W\in\mathcal{W}} F_t(\overrightarrow{W})$, we have $F_t(\overrightarrow{W}_{t+1}) \leq F_t(\overrightarrow{W}_t)$, which implies

$$\frac{1}{\alpha}\|\overrightarrow{W}_{t+1} - \overrightarrow{W}_t\|_{\widetilde{H}_t}^2 \leq \langle\nabla\ell_t(\overrightarrow{W}_t), \overrightarrow{W}_t - \overrightarrow{W}_{t+1}\rangle.$$

Then, by the Hölder's inequality, we can further bound the inner product term by

$$\langle\nabla\ell_t(\overrightarrow{W}_t), \overrightarrow{W}_t - \overrightarrow{W}_{t+1}\rangle \leq \|\overrightarrow{W}_{t+1} - \overrightarrow{W}_t\|_{\widetilde{H}_t} \cdot \|\nabla\ell_t(\overrightarrow{W}_t)\|_{\widetilde{H}_t^{-1}},$$

which indicates that

$$\|\overrightarrow{W}_{t+1} - \overrightarrow{W}_t\|_{\widetilde{H}_t} \leq \alpha\|\nabla\ell_t(\overrightarrow{W}_t)\|_{\widetilde{H}_t^{-1}}.$$

Since $\widetilde{H}_t \succcurlyeq H_t$ and $\widetilde{H}_t^{-1} \preccurlyeq I_K d/\lambda$, we can further have,

$$\|\overrightarrow{W}_{t+1} - \overrightarrow{W}_t\|_{H_t} \leq \|\overrightarrow{W}_{t+1} - \overrightarrow{W}_t\|_{\widetilde{H}_t} \leq \alpha\|\nabla\ell_t(\overrightarrow{W}_t)\|_{\widetilde{H}_t^{-1}} \leq \frac{\alpha}{2\lambda}\|\nabla\ell_t(\overrightarrow{W}_t)\|_2 \leq \frac{\alpha\sqrt{K}}{\lambda},$$

where the last inequality is due to the definition of $\nabla\ell_t(\overrightarrow{W}_t) = (\boldsymbol{\sigma}_t(\overrightarrow{W}_t) - \mathbf{y}_t) \otimes \mathbf{x}_t$ such that $\|\nabla\ell_t(\overrightarrow{W}_t)\|_2 \leq 2\sqrt{K}$. $\square$

**Lemma 21.** *Let* $H_t = \lambda I_{Kd} + \sum_{s=1}^{t-1}\nabla\boldsymbol{\sigma}(W_{s+1}\mathbf{x}_s) \otimes \mathbf{x}_s\mathbf{x}_s^\top$. *Then, we have*

$$\sum_{t=1}^{T}\|H_t^{-\frac{1}{2}}(I_K \otimes \mathbf{x}_t)\|_2^2 \leq \kappa d\ln\left(1 + \frac{T}{\lambda\kappa}\right),$$

*where* $\lambda > 2$.

*Proof of Lemma 21.* The proof of Lemma 21 shares the same spirits with that of Lemma 19. Let $\bar{V}_t = \frac{\lambda}{2}I_{Kd} + \frac{1}{\kappa}\sum_{s=1}^{t} I_K \otimes \mathbf{x}_s\mathbf{x}_s^\top$. We have $H_t \succcurlyeq \bar{V}_t$ when $\lambda > 2$. We can prove the lemma by

$$\sum_{t=1}^{T}\|H_t^{-\frac{1}{2}}(I_K \otimes \mathbf{x}_t)\|_2^2$$

$$= \sum_{t=1}^{T}\lambda_{\max}\left((I_K \otimes \mathbf{x}_t^\top)H_t^{-1}(I_K \otimes \mathbf{x}_t)\right)$$

$$\leq \sum_{t=1}^{T} \lambda_{\max} \left( (I_K \otimes \mathbf{x}_t^\top) \bar{V}_t^{-1} (I_K \otimes \mathbf{x}_t) \right)$$

$$= \sum_{t=1}^{T} \lambda_{\max} \left( (I_K \otimes \mathbf{x}_t^\top) \left( I_K \otimes \left( \lambda I_d + \frac{1}{\kappa} \sum_{s=1}^{t} \mathbf{x}_s \mathbf{x}_s^\top \right)^{-1} \right) (I_K \otimes \mathbf{x}_t) \right)$$

$$= \sum_{t=1}^{T} \lambda_{\max} \left( I_K \otimes \left( \mathbf{x}_t^\top \left( \lambda I_d + \frac{1}{\kappa} \sum_{s=1}^{t} \mathbf{x}_s \mathbf{x}_s^\top \right)^{-1} \mathbf{x}_t \right) \right)$$

$$= \mathbf{x}_t^\top \left( \lambda I_d + \frac{1}{\kappa} \sum_{s=1}^{t} \mathbf{x}_s \mathbf{x}_s^\top \right)^{-1} \mathbf{x}_t \leq \kappa d \ln \left( 1 + \frac{T}{\lambda \kappa} \right),$$

where the first inequality is due to $H_t \succcurlyeq \bar{V}_t$. The second equality is due to the definition of $\bar{V}_t$ and the third equality is due to the mixed-product property of the Kronecker product. The last inequality can be obtain by the standard elliptical potential lemma [3, Lemma 11]. $\qquad\square$

### C.3.4 Computation Cost of Algorithm 2

---
**Algorithm 2** OFUL-MLogB

---
**Input:** regularization coefficient $\lambda$, probability $\delta$, step size $\eta$.
1: Initialize $H_1 = \lambda I_{Kd}$ and $\overrightarrow{W}_1$ as any point in $\mathcal{W}$
2: **for** $t = 1, \ldots, T$ **do**
3:     Select the arm by $\mathbf{x}_t = \arg\max_{\mathbf{x} \in \mathcal{X}} \widetilde{r}_t(\mathbf{x})$ and receive $y_t$.
4:     Update $\widetilde{H}_t = H_t + \eta \nabla \boldsymbol{\sigma}(W_t \mathbf{x}_t) \otimes \mathbf{x}_t \mathbf{x}_t^\top$
5:     Update the estimator $\overrightarrow{W}_{t+1}$ for the next iteration by (6)
6:     Update $H_{t+1} = H_t + \nabla \boldsymbol{\sigma}(W_{t+1} \mathbf{x}_t) \otimes \mathbf{x}_t \mathbf{x}_t^\top$ and
7:     Construct the optimistic reward by $\widetilde{r}_{t+1}(\mathbf{x}) = \boldsymbol{\rho}^\top \boldsymbol{\sigma}(W_{t+1} \mathbf{x}) + \epsilon_{t+1}^{\mathtt{fst}}(\mathbf{x}) + \epsilon_{t+1}^{\mathtt{snd}}(\mathbf{x})$ as (8).
8: **end for**

---

Here, we discuss about the computation cost of the Algorithm 2. For each iteration, our algorithm requires to maintain the inverse of matrix $H_{t+1} = H_t + \nabla \boldsymbol{\sigma}(W_{t+1} \mathbf{x}_t) \otimes \mathbf{x}_t \mathbf{x}_t^\top$ and $\widetilde{H}_t = H_t + \eta \nabla \boldsymbol{\sigma}(W_t \mathbf{x}_t) \otimes \mathbf{x}_t \mathbf{x}_t^\top$. Since $\nabla \boldsymbol{\sigma}(W_{t+1} \mathbf{x}_t) \otimes \mathbf{x}_t \mathbf{x}_t^\top$ and $\nabla \boldsymbol{\sigma}(W_t \mathbf{x}_t) \otimes \mathbf{x}_t \mathbf{x}_t^\top$ are both rank-$K$ matrix, then one can main $H_t$ and $\widetilde{H}_t$ with $\mathcal{O}(K^3 d^2)$ computation cost per iteration by the Sherman-Morrison-Woodbury formula. Given the $H_t^{-1}$ and $\widetilde{H}_t^{-1}$, we discuss the computation cost of the construction of the optimistic reward 8 and the update rule for the estimator (6).

**Computation Cost of the Estimator** (6). As shown by the discussion in Section 3.2, the update rule (6) is identical to

$$\overrightarrow{Z}_{t+1} = \overrightarrow{W}_t - \eta \widetilde{H}_t^{-1} \nabla \ell_t(\overrightarrow{W}_t) \quad \text{and} \quad \overrightarrow{W}_{t+1} = \arg\min_{\overrightarrow{W} \in \mathcal{W}} \| \overrightarrow{W} - \overrightarrow{Z}_{t+1} \|_{\widetilde{H}_t}.$$

Given the $\widetilde{H}_t^{-1}$, we can perform the gradient step with $\mathcal{O}(K^2 d^2)$ time complexity. As for the projection step, the optimization problem can be solved in $\mathcal{O}(K^3 d^3)$ time [18, Section 4]. As a consequence, the overall time complexity for obtaining the estimator (6) is $\mathcal{O}(K^3 d^3)$.

**Computation Cost of Building Optimistic Reward.** As shown by (8), we construct the optimistic reward by $\widetilde{r}_t^{\mathtt{OL}}(\mathbf{x}) = \boldsymbol{\rho}^\top \boldsymbol{\sigma}(W_t^{\mathtt{OL}} \mathbf{x}) + \epsilon_t^{\mathtt{fst}}(\mathbf{x}) + \epsilon_t^{\mathtt{snd}}(\mathbf{x})$. We can compute $\epsilon_t^{\mathtt{fst}}(\mathbf{x})$ with the following equivalent formulation

$$\epsilon_t^{\mathtt{fst}}(\mathbf{x}) = \beta_t^{\mathtt{OL}}(\delta) \cdot \| H_t^{-\frac{1}{2}} (I_K \otimes \mathbf{x}) \nabla \boldsymbol{\sigma}(W_t) \boldsymbol{\rho} \|_2$$

$$= \beta_t^{\mathtt{OL}}(\delta) \sqrt{\boldsymbol{\rho}^\top (\nabla \boldsymbol{\sigma}(W_t \mathbf{x}) \otimes \mathbf{x}^\top) H_t^{-1} \boldsymbol{\rho}^\top (\nabla \boldsymbol{\sigma}(W_t \mathbf{x}) \otimes \mathbf{x}) \boldsymbol{\rho}}.$$

Given $H_t^{-1}$, it will take $\mathcal{O}(K^2 d^2)$ to calculate $\epsilon_t^{\mathtt{fst}}(\mathbf{x})$. As for the term $\epsilon_t^{\mathtt{fst}}(\mathbf{x})$, it can also be rewritten as

$$\epsilon_t^{\mathtt{snd}}(\mathbf{x}) = 3R \left( \beta_t^{\mathtt{OL}} \right)^2 \cdot \| (I_K \otimes \mathbf{x}^\top) H_t^{-1/2} \|_2^2$$

$$= 3R \left( \beta_t^{\texttt{OL}} \right)^2 \sqrt{\lambda_{\max} \left( (I_K \otimes \mathbf{x}\mathbf{x}^\top) H_t^{-1} \right)},$$

It would take $\mathcal{O}(K^3 d^3)$ time to perform the eigenvalue decomposition. Given there are $|\mathcal{X}|$ arm, the time complexity for identify the arm $\mathbf{x}_t = \arg\max_{\mathbf{x} \in \mathcal{X}} \widetilde{r}_t(\mathbf{x})$ is in total $\mathcal{O}(|\mathcal{X}|K^3 d^3)$.

Overall, Algorithm 2 can be implemented in $\mathcal{O}(|\mathcal{X}|K^3 d^3)$ time, which is independent of $T$.

## C.4 Proof of Corollary 1

### C.4.1 Main Proof

In the binary case, we select the arm by $(\mathbf{x}_t, \widetilde{\mathbf{w}}_t) = \arg\max_{\mathbf{x} \in \mathcal{X}, W \in \mathcal{C}_t(\delta)} \mathbf{w}^\top \mathbf{x}$. In such a case, we show Algorithm 2 achieves an $\widetilde{O}(T/\kappa_*)$ bound with a constant computation cost. The proof is almost the same as that of [10, Theorem 2]. The main difference is that we complete the proof with the confidence set of the efficient online estimator (Theorem 3). We present the proof here for self-containedness.

*Proof of Corollary 1.* When $K = 1$ and $\rho_1 = 1$, the MLogB problem recovers the binary logistic bandit problem with the feedback $y_t = \{0, 1\}$ and the reward model $\Pr[y = 1 | \mathbf{x}] = \sigma(\mathbf{w}_*^\top \mathbf{x})$. In such a case, we can decompose the regret by

$$
\begin{aligned}
\mathrm{Reg}_T &= \sum_{t=1}^T \sigma(\mathbf{w}_*^\top \mathbf{x}_*) - \sum_{t=1}^T \sigma(\mathbf{w}_*^\top \mathbf{x}_t) \\
&\leq \sum_{t=1}^T \sigma(\widetilde{\mathbf{w}}_t^\top \mathbf{x}_t) - \sum_{t=1}^T \sigma(\mathbf{w}_*^\top \mathbf{x}_t) \\
&= \sum_{t=1}^T \sigma'(\mathbf{w}_*^\top \mathbf{x}_t)(\widetilde{\mathbf{w}}_t^\top - \mathbf{w}_*)^\top \mathbf{x}_t + \sum_{t=1}^T \sigma''(\boldsymbol{\xi}_t \mathbf{x}_t)((\widetilde{\mathbf{w}}_t - \mathbf{w}_*)^\top \mathbf{x}_t)^2 \\
&= \underbrace{\sum_{t \in I_1} \sigma'(\mathbf{w}_*^\top \mathbf{x}_t)(\widetilde{\mathbf{w}}_t^\top - \mathbf{w}_*)^\top \mathbf{x}_t}_{\texttt{term (a)}} + \underbrace{\sum_{t \in I_2} \sigma'(\mathbf{w}_*^\top \mathbf{x}_t)(\widetilde{\mathbf{w}}_t^\top - \mathbf{w}_*)^\top \mathbf{x}_t}_{\texttt{term (b)}} + \underbrace{\sum_{t=1}^T \sigma''(\boldsymbol{\xi}_t \mathbf{x}_t)((\widetilde{\mathbf{w}}_t - \mathbf{w}_*)^\top \mathbf{x}_t)^2}_{\texttt{term (c)}},
\end{aligned}
$$

where the first inequality is due to the arm selection rule. The first equality is due to the Taylor series and $\boldsymbol{\xi}_t \in \mathbb{R}^d$ is a certain point on the line connecting $\widetilde{\mathbf{w}}_t$ and $\mathbf{w}_*$. For the last equality, we divide the time horizon into two parts $I_1 = \{t \in [T] \mid \sigma'(\mathbf{w}_*^\top \mathbf{x}_t) \geq \sigma'(\mathbf{w}_{t+1}^\top \mathbf{x}_t)\}$ and $I_2 = [T]/I_1$.

For term (a), we have

$$
\begin{aligned}
&\texttt{term (a)} \\
&= \sum_{t \in I_1} \sigma'(\mathbf{w}_*^\top \mathbf{x}_t)(\widetilde{\mathbf{w}}_t^\top - \mathbf{w}_*)^\top \mathbf{x}_t \\
&\leq \sum_{t \in I_1} \sigma'(\mathbf{w}_{t+1}^\top \mathbf{x}_t)(\widetilde{\mathbf{w}}_t - \mathbf{w}_*)^\top \mathbf{x}_t + \sum_{t \in I_1} |(\mathbf{w}_{t+1} - \mathbf{w}_*)^\top \mathbf{x}_t|(\widetilde{\mathbf{w}}_t - \mathbf{w}_*)^\top \mathbf{x}_t \\
&\leq \underbrace{\sum_{t \in I_1} \sqrt{\sigma'(\mathbf{w}_{t+1}^\top \mathbf{x}_t)} \cdot \|\widetilde{\mathbf{w}}_t - \mathbf{w}_*\|_{H_t} \cdot \|\sqrt{\sigma'(\mathbf{w}_{t+1}^\top \mathbf{x}_t)} \mathbf{x}_t\|_{H_t^{-1}}}_{\texttt{term (a-1)}} \\
&\quad + \underbrace{\sum_{t \in I_1} \|\mathbf{x}_t\|_{H_t^{-1}}^2 \|\widetilde{\mathbf{w}}_t - \mathbf{w}_*\|_{H_t} \cdot \|\mathbf{w}_{t+1} - \mathbf{w}_*\|_{H_{t+1}}}_{\texttt{term (a-2)}},
\end{aligned}
$$

where the first inequality is due to the mean value theorem and the condition that $|\sigma''(z)| \leq 1$ for any $z \in \mathbb{R}$. The second inequality is due to the Cauchy-Schwarz inequality and $H_t \preccurlyeq H_{t+1}$. We can further bound term (a-1) by

$$\texttt{term (a-1)} = \sum_{t \in I_1} \sqrt{\sigma'(\mathbf{w}_{t+1}^\top \mathbf{x}_t)} \cdot \|\widetilde{\mathbf{w}}_t - \mathbf{w}_*\|_{H_t} \cdot \|\sqrt{\sigma'(\mathbf{w}_{t+1}^\top \mathbf{x}_t)} \mathbf{x}_t\|_{H_t^{-1}}$$

$$\leq 2\beta_T(\delta)\sqrt{\sum_{t\in I_1}\sigma'(\mathbf{w}_{t+1}^\top \mathbf{x}_t)}\sqrt{\sum_{t\in I_1}\|\sqrt{\sigma'(\mathbf{w}_{t+1}^\top \mathbf{x}_t)}\mathbf{x}_t\|^2_{H_t^{-1}}}$$

$$\leq 2\beta_T(\delta)\sqrt{\sum_{t\in I_1}\sigma'(\mathbf{w}_*^\top \mathbf{x}_t)}\sqrt{\sum_{t\in I_1}\|\sqrt{\sigma'(\mathbf{w}_{t+1}^\top \mathbf{x}_t)}\mathbf{x}_t\|^2_{H_t^{-1}}}$$

$$\leq 4\beta_T(\delta)\cdot\sqrt{\mathrm{Reg}_T + T\sigma'(\mathbf{w}_*^\top \mathbf{x}_*)}\cdot\sqrt{d\log(1+\frac{T}{\lambda})}, \tag{62}$$

where the first inequality is due to $\widetilde{\mathbf{w}}_t$ and $\mathbf{w}_*$ are both contained in $\mathcal{C}_t(\delta)$ and $\beta_t(\delta) = \mathcal{O}(\log t)$ is the radius of $\mathcal{C}_t(\delta)$ as shown in Theorem 3. The last second inequality is by the condition such that $\sigma'(\mathbf{w}_*^\top \mathbf{x}_t) \geq \sigma'(\mathbf{w}_{t+1}^\top \mathbf{x}_t)$ for all $t \in I_1$. Then, following the same argument in the proof of [10, Theorem 2], one can show

$$\sqrt{\sum_{t\in I_1}\sigma'(\mathbf{w}_*^\top \mathbf{x}_t)} \leq \sqrt{\sum_{t\in T}\sigma'(\mathbf{w}_*^\top \mathbf{x}_t)} \leq \sqrt{\mathrm{Reg}_T + T\sigma'(\mathbf{w}_*^\top \mathbf{x}_*)}, \tag{63}$$

which leads to the last inequality. We also use the elliptical potential lemma [10, Lemma 9] in the last inequality.

As for term (a-2), we have

$$\mathtt{term\ (a\text{-}2)} \leq (\beta_T(\delta))^2 \sum_{t\in[T]}\|\mathbf{x}_t\|^2_{H_t^{-1}} \leq \kappa d(\beta_T(\delta))^2 \ln(1+\frac{T}{\lambda\kappa}), \tag{64}$$

where the first inequality holds because $\mathbf{w}_*$ and $\widetilde{\mathbf{w}}_t$ are contained in $\mathcal{C}_t(\delta)$. The second inequality can be obtained following the similar argument in obtaining (60). Combining the upper bound for term (a-1) and term (a-2), we have

$$\mathtt{term\ (a)} \leq 4\beta_T(\delta)\sqrt{d\log(1+\frac{T}{\lambda})}\cdot\sqrt{\mathrm{Reg}_T + T\sigma'(\mathbf{w}_*^\top \mathbf{x}_*)} + \kappa d(\beta_T(\delta))^2 \ln(1+\frac{T}{\lambda\kappa}).$$

As for term (b), we have $\sigma'(\mathbf{w}_*^\top \mathbf{x}_t) < \sigma'(\mathbf{w}_{t+1}^\top \mathbf{x}_t)$ for all $t \in I_2$. Then, we have

$$\mathtt{term\ (b)} \leq \sum_{t\in I_2}\sqrt{\sigma'(\mathbf{w}_*^\top \mathbf{x}_t)}\cdot\|\widetilde{\mathbf{w}}_t - \mathbf{w}_*\|_{H_t}\cdot\|\sqrt{\sigma'(\mathbf{w}_*^\top \mathbf{x}_t)}\mathbf{x}_t\|_{H_t^{-1}}$$

$$\leq \sum_{t\in I_2}\sqrt{\sigma'(\mathbf{w}_*^\top \mathbf{x}_t)}\cdot\|\widetilde{\mathbf{w}}_t - \mathbf{w}_*\|_{H_t}\cdot\|\sqrt{\sigma'(\mathbf{w}_{t+1}^\top \mathbf{x}_t)}\mathbf{x}_t\|_{H_t^{-1}}$$

$$\leq 4\beta_T(\delta)\sqrt{d\log(1+\frac{T}{\lambda})}\cdot\sqrt{\mathrm{Reg}_T + T\sigma'(\mathbf{w}_*^\top \mathbf{x}_*)},$$

where the second inequality is due to the condition $\sigma'(\mathbf{w}_*^\top \mathbf{x}_t) < \sigma'(\mathbf{w}_{t+1}^\top \mathbf{x}_t)$ and the last inequality can be obtained using similar arguments as those used to derive (62).

Regarding term (c), the application of similar arguments used to derive equation (62) demonstrates.

$$\mathtt{term\ (c)} \leq \kappa d(\beta_T(\delta))^2 \ln(1+\frac{T}{\lambda\kappa}).$$

Then, combining the upper bounds for term (a), term (b) and term (c), we have

$$\mathrm{Reg}_T \leq 8\beta_T(\delta)\sqrt{d\log(1+\frac{T}{\lambda})}\cdot\sqrt{\mathrm{Reg}_T + T\sigma'(\mathbf{w}_*^\top \mathbf{x}_*)} + 2\kappa d(\beta_T(\delta))^2 \ln(1+\frac{T}{\lambda\kappa}).$$

Resolving the above displayed inequality leads to

$$\mathrm{Reg}_T \leq 32\beta_T(\delta)\sqrt{d\log(1+\frac{T}{\lambda})}\sqrt{T\sigma'(\mathbf{w}_t^\top \mathbf{x}_*)} + (8\kappa + 64)d(\beta_T(\delta))^2 \ln(1+\frac{T}{\lambda}).$$

In the binary case, $\beta_T(\delta) = \mathcal{O}(\sqrt{d}\log T)$. Then we have $\mathrm{Reg}_T = \mathcal{O}(d\log^{\frac{3}{2}} T\sqrt{T/\kappa_*} + \kappa d^2 \log^3 T)$, which completes the proof. $\qquad\square$

### C.4.2 Computation Cost for Binary Case

Since the binary logistic bandit is a special case of the MLogB problem, the time complexity analysis in Appendix C.3.4 is also applicable in the binary case. The only difference is that we select the arm as $(\mathbf{x}_t, \widetilde{\mathbf{w}}t) = \arg\max \mathbf{x} \in \mathcal{X}, W \in \mathcal{C}_t(\delta) \mathbf{w}^\top \mathbf{x}$ in the binary case. As $\mathcal{C}_t(\delta) \triangleq \{\mathbf{w} \in \mathcal{W} \mid \|\mathbf{w} - \mathbf{w}_*\|_{H_t} \leq \beta_t(\delta)\}$ is an ellipsoid, the optimization can be rewritten as:

$$\mathbf{x}_t = \arg\max_{\mathbf{x}\in\mathcal{X}} \left( \max_{\|\mathbf{w}-\mathbf{w}_t\|_{H_t}\leq\beta_t(\delta)} \mathbf{w}^\top\mathbf{x} \right)$$

$$= \arg\max_{\mathbf{x}\in\mathcal{X}} \left( \max_{\|\mathbf{u}\|_{H_t}\leq\beta_t(\delta)} \mathbf{w}_t^\top\mathbf{x} + \mathbf{u}^\top\mathbf{x} \right)$$

$$= \arg\max_{\mathbf{x}\in\mathcal{X}} \mathbf{w}_t^\top\mathbf{x} + \beta_t(\delta)\|\mathbf{x}\|_{H_t^{-1}}.$$

Given $H_t^{-1}$, the above optimization problem can be solved in $\mathcal{O}(d^2)$ time.

### C.5 On the Minimax Optimal Bound for the MLogB

In the binary case, we have achieved the minimax dynamic regret bound $\widetilde{\mathcal{O}}(\sqrt{T/\kappa_*})$ up to a logarithmic factor in terms of $T$. In the MLogB problem, the best-known result is $\widetilde{\mathcal{O}}(K\sqrt{T})$. A natural question arises: can we obtain a similar minimax optimal result in the MLogB problem? However, we find that it remains challenging both in terms of regret analysis and the design of efficient algorithms.

**Challenge in Regret Analysis.** To establish the minimax optimal bound for the binary case, as demonstrated in [9, 10] and we restate in (63), a critical step involves showing

$$\sqrt{\sum_{t\in T}\sigma'(\mathbf{w}_*^\top\mathbf{x}_t)} \leq \sqrt{\text{Reg}_T + T\sigma'(\mathbf{w}_*^\top\mathbf{x}_*)}.$$

In the MLogB problem, in order to obtain a similar result to the minimax optimal bound achieved in the binary case, it is sufficient to demonstrate

$$\sqrt{\sum_{t=1}^{T}\boldsymbol{\rho}^\top\nabla\boldsymbol{\sigma}(W_*\mathbf{x}_t)\boldsymbol{\rho}} \leq \sqrt{2R\cdot\text{Reg}_T + \sum_{t=1}^{T}\boldsymbol{\rho}^\top\Sigma(W_*\mathbf{x}_*)\boldsymbol{\rho}}. \tag{65}$$

However, it is unclear how to prove such a relationship as the binary case. Indeed, denoting by $\mathbf{r}_t = \boldsymbol{\sigma}(W_*\mathbf{x}_*) - \boldsymbol{\sigma}(W_*\mathbf{x}_t)$ the reward vector and $r_{t,k}$ by its $k$-th entry, we can further rewrite the $k$-th entry of the vector $\boldsymbol{\rho}^\top\Sigma(W_*\mathbf{x}_t)$ can be further written as

$$[\boldsymbol{\rho}^\top\Sigma(W_*\mathbf{x}_t)]_k = \sigma_k(W_*\mathbf{x}_t)\boldsymbol{\rho}^\top(\mathbf{e}_k - \boldsymbol{\sigma}(W_*\mathbf{x}_t))$$

$$= (\sigma_k(W_*\mathbf{x}_*) - r_{t,k})\rho_k - (\sigma_k(W_*\mathbf{x}_*) - r_{t,k})\boldsymbol{\rho}^\top(\boldsymbol{\sigma}(W_*\mathbf{x}_*) - \mathbf{r}_t)$$

$$= \sigma_k(W_*\mathbf{x}_*)\rho_k - \sigma_k(W_*\mathbf{x}_*)\boldsymbol{\rho}^\top\boldsymbol{\sigma}(W_*\mathbf{x}_*) + (\sigma_k(W_*\mathbf{x}_*))\boldsymbol{\rho}^\top\mathbf{r}_t + r_{t,k}\boldsymbol{\rho}^\top(\boldsymbol{\sigma}(W_*\mathbf{x}_t) - \mathbf{e}_k)$$

$$= [\boldsymbol{\rho}^\top\Sigma(W_*\mathbf{x}_*)]_k + \sigma_k(W_*\mathbf{x}_*)\boldsymbol{\rho}^\top\mathbf{r}_t + r_{t,k}\boldsymbol{\rho}^\top(\boldsymbol{\sigma}(W_*\mathbf{x}_t) - \mathbf{e}_k).$$

In such a case, we can bound the left hand side of (65) by

$$\sqrt{\sum_{t=1}^{T}\boldsymbol{\rho}^\top\Sigma(W_*\mathbf{x}_t)\boldsymbol{\rho}} = \sqrt{\left(\boldsymbol{\rho}^\top\Sigma(W_*\mathbf{x}_*)\boldsymbol{\rho} + \boldsymbol{\rho}^\top\mathbf{r}_t\cdot\boldsymbol{\rho}^\top\left(\boldsymbol{\sigma}(W_*\mathbf{x}_t) + \boldsymbol{\sigma}(W_*\mathbf{x}_*)\right) - \sum_{k=1}^{K}\rho_k^2 r_{t,k}\right)}$$

$$\leq \sqrt{\sum_{t=1}^{T}\boldsymbol{\rho}^\top\Sigma(W_*\mathbf{x}_*)\boldsymbol{\rho} + 2R\cdot\text{Reg}_T - \sum_{t=1}^{T}\sum_{k=1}^{K}\rho_k^2 r_{t,k}}. \tag{66}$$

In the binary case, In the binary case, the additional term becomes $-\sum_{t=1}^{T} r_t = -\text{Reg}_T < 0$, thereby resulting in the derivation of (63). Nevertheless, in the context of the MLogB problem, this particular term has the potential to assume positive values, posing challenges in determining an upper bound for it.

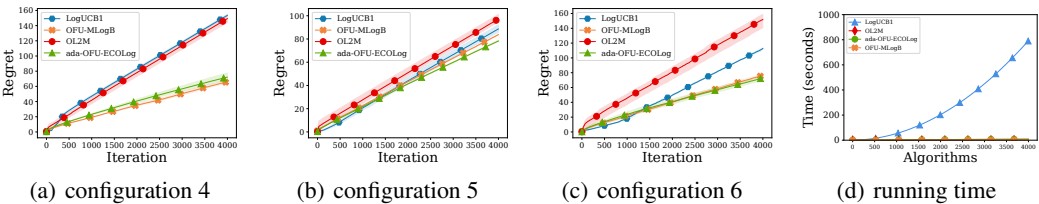

(a) configuration 4  (b) configuration 5  (c) configuration 6  (d) running time

Figure 3: More results for binary logistic bandit.

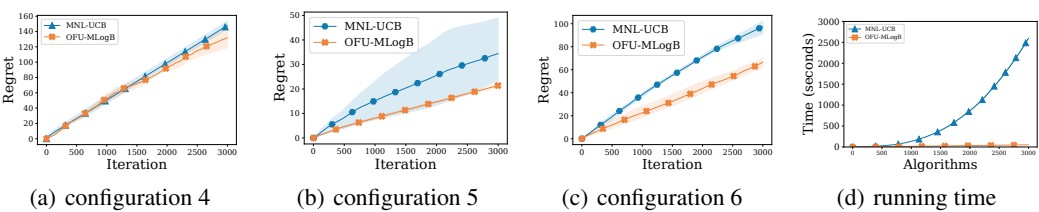

(a) configuration 4  (b) configuration 5  (c) configuration 6  (d) running time

Figure 4: More results for multinomial logistic bandit.

**Challenge in Efficient Algorithm Design.** When extending algorithms that achieve minimax optimal bounds from the binary case to the multinomial case, there are computational concerns to address. In the binary case, the minimax optimal regret bound relies on constructing an optimistic reward through $\widetilde{r}_t(\mathbf{x}) = \arg\max_{\mathbf{w} \in \mathcal{C}_t(\delta)} \sigma(\mathbf{w}^\top \mathbf{x})$. Since $\sigma(\mathbf{z})$ is an increasing function, the optimization problem can be simplified to

$$\widetilde{r}_t(\mathbf{x}) = \arg\max_{\mathbf{w} \in \mathcal{C}_t(\delta)} \mathbf{w}^\top \mathbf{x},$$

which becomes a convex optimization problem when $\mathcal{C}_t(\delta)$ is a convex set. However, when applying this analysis to the multinomial case, solving the problem requires optimizing

$$\widetilde{r}_t(\mathbf{x}) = \arg\max_{W \in \mathcal{C}_t(\delta)} \boldsymbol{\rho}^\top \boldsymbol{\sigma}(W\mathbf{x}).$$

Here, the loss function is non-concave, rendering the maximization problem challenging to efficiently solve. Consequently, new approaches for constructing the optimistic reward function may be necessary.

# D  Omitted Details for Experiments

In this section, we presents more empirical results for other configurations and introduced the parameter configurations for the contenders.

**Experimental Details.** For each configuration, both the arm set $\mathcal{X}$ and the underlying unknown parameter $W_*$ are randomly selected. In the binary case, the norm of the unknown parameter is set as $\|\mathbf{w}_*\|_2 = S$ with $S = 5$. As for the multinomial case, each row of $\mathcal{W}_*$ are randomly sampled with $\|\mathbf{w}_*^{(k)}\|_2 = \widetilde{S}$ for all $k \in [K]$ with $\widetilde{S} = 1$. The parameter of all contenders are set according to their order as suggested in the corresponding paper. We use $\lambda = d\log(T)$ for Log-UCB1, $\lambda = d$ for OL2M, $\lambda = 1$ for ada-OFU-ECOLog and $\lambda = \sqrt{K}\alpha S$ for OFU-MLogB. The step size for OFU-MLogB is set as $\eta = S/2 + \ln(K+1)/4$. The experiments are run on Xeon E-2288G processors (8 cores, 3.7GHz base, 5.0GHz boost).

**More Results.** Figure 3 provides additional empirical results for the binary logistic bandit problem. These results align with the trends observed in the main paper, wherein our `OFU-MLogB` algorithm shows performance comparable to `ada-OFU-ECOLog` but with reduced computational overhead. Meanwhile, more results for the multinomial case are provided by Figure 4. The cumulative running time for `MNL-UCB` increase at a rate of $\mathcal{O}(t^2)$. In contrast, the running time for our `OFU-MLogB` exhibits a linear dependence with $t$, attributable to the constant computation cost per round.

