# OpenReview forum: "Online (Multinomial) Logistic Bandit: Improved Regret and Constant Computation Cost"
_NeurIPS.cc/2023/Conference — NeurIPS 2023 spotlight_

### Official Review · Reviewer_oJs3 · 2023-07-03

**Soundness:** 3 good
**Presentation:** 4 excellent
**Contribution:** 3 good
**Rating:** 7
**Confidence:** 3

**Summary:**

This paper proposes a computationally efficient algorithm for multinomial logistic bandit with improved regret and computation cost. The algorithm uses online mirror descent and new approximation to efficiently compute consistent estimator and construct optimistic reward. Experimental results show improved regrets and computational costs of the proposed algorithm.

**Strengths:**

The contributions are clear, concrete, and well-presented. Improving the regret and computation cost and extending binary cases to multinomial is significant. Applying online mirror decent, novel approximation, and novel construction of optimistic reward to reduce the computation cost are novel and significant in contextual bandits under the parametric model.

**Weaknesses:**

1. Discussion and experimental results on whether there is a trade-off to gaining efficiency would be helpful.
2. In experimental results, while the reduction of the computation cost is significant, the reduction of regret seems minor. More experiments are needed to validate the improvement of the proposed algorithm in terms of regret.

**Questions:**

1. Is there any trade-offs between the computational efficiency and the accuracy of the estimator?
2. Could you explain why the slope of the regret of MLogB increases in Figure 1 (c)?
3. If the number of iterations $T$ increases, can the proposed algorithm find a better policy to reduce the increment of the regret?

**Limitations:**

Yes. This work is mostly theoretical and potential negative societal impact is unseen.

---

> ### Author Rebuttal · Authors · 2023-08-10
>
> Thank you for the positive evaluation of this paper and the insightful common! In the following, we will address your questions and will improve the paper according to your suggestions.
>
> ---
>
> **Q1:** Discussion and experimental results on whether there is a trade-off to gaining efficiency would be helpful.
>
> **A1:** Thank you for the insightful question. There is indeed a trade-off between computational efficiency and the quality of the estimator. This trade-off can be observed by comparing our update rule $W_{t+1} = \arg\min\_{W\in\mathcal{W}}\langle \nabla\ell_t(W_t),W\rangle + \frac{1}{2}\Vert W-W_t\Vert_{\tilde{H}\_t}^2$  with the one $W_{t+1} = \arg\min\_{W\in\mathcal{W}}\ell_t(W) + \frac{1}{2}\Vert W-W_t\Vert_{H_t}^2$ used in [10]. The former is known as a standard online mirror descent (OMD), while the latter is referred to as implicit OMD, which updates on the original loss function instead of the linearized approximation.
>
> In the online learning literature, both algorithms attain the $O(\sqrt{T})$ minimax regret bound. However, implicit OMD has been observed to enjoy superior empirical performance [1] and has been proven to perform better than OMD when $\{\ell_t\}_{t=1}^T$ is stable [2]. The price for the better performance of implicit OMD is increased computational complexity, as it has to solve a convex optimization and requires $O(\log T)$ time complexity.
>
> Our experiments (especially the updated ones during the rebuttal phase) further demonstrate this trade-off, as the performance of our algorithm is slightly inferior to ada-OFU-ECOLog proposed by [12], but with improved computational efficiency. In the next version, we will add more discussion on the trade-off between computational efficiency and statistical efficiency.
>
> **Ref:**
>
> [1] Kulis and Bartlett. Implicit online learning. ICML 2010.
>
> [2] Campolongo and Orabona. Temporal variability in implicit online learning. NeurIPS 2020.
>
> ---
>
> **Q2:** Could you explain why the slope of the regret of MLogB increases in Figure 1 (c)?
>
> **A2:** We think there are two reasons: the first is the time horizon $T$ could be somewhat short to reflect sublinear regret. Even in the binary case, the contenders (OFU-MLogB, LogUCB1 and OL2M) appear to exhibit a linear regret within $T=1200$. The second reason might be that the results are averaged over 20 trails with randomly selected arm set. As shown in the additional experiments, the performance of all contenders are highly affected by the shape of the arm set. The reported averaged result could have a high variance.
>
> To better illustrate the performance of our algorithm, we have conducted additional experiments with longer time horizon $T=6000$ and fixed the arm set. Due to the limited time,the experiments are  conducted on the binary case. We are happy to provide more results for the multinomial case latter. Please refer to the global response for more details. Thanks!
>
> ---
>
> **Q3:** If the number of iterations $T$ increases, can the proposed algorithm find a better policy to reduce the increment of the regret?
>
> **A3:** Thanks for the suggestions. We have conducted additional experiments with a longer $T$ to test our method. Please kindly refer to the global rebuttal for more details. Thanks!

---

> > ### Comment · Reviewer_oJs3 · 2023-08-18
> >
> > I thank the authors for the detailed and helpful responses and additional experiment results. I will change my score from 6 to 7 accordingly.

---

> > > ### Author Response · Authors · 2023-08-18
> > >
> > > Thank you for the insightful comments and for updating the score! In the next version, we will update the paper based on your suggestions and the discussion in the rebuttal phase. Specifically, we will add a discussion on the trade-off between statistical efficiency and computational efficiency in Section 3. We will also reorganize the experimental section to present the comparison using fixed arm sets with different random seeds and a longer time horizon.

---

### Official Review · Reviewer_6tgu · 2023-07-06

**Soundness:** 2 fair
**Presentation:** 3 good
**Contribution:** 3 good
**Rating:** 6
**Confidence:** 4

**Summary:**

This paper considers Multinomial Logistic bandits (the extension of the usual logistic bandit model to a setting where there are more than two possible outcomes - e.g. in advertising one picks an action in response to a context and may observe 'no click'/'click'/'save for later'/etc instead of just 'no click'/'click').

This problem has been previously considered in work such as Amani and Thrampoulidis (2021) where an optimistic algorithm with near optimal dependence on T (up to logarithmic factors) has been derived, but the focus of this paper is on algorithms with improved computational efficiency, and optimal dependence on the parameter $\kappa$. Dependence on this parameter $\kappa$ has been the focus of a series of recent papers on bandit problems involving the logistic function, as many algorithms with an optimal dependence on $T$ can nonetheless have a poor dependence on $\kappa$ which in turn brings an exponential dependence (in terms of worst-case performance) on other problem parameters.

The present paper first improves the MNL-UCB algorithm of Amani and Thrampoulidis by providing a sharper tail-inequality for the MLE under the multinomial model, before providing the main contribution: the OFU-MLogB algorithm, and analysis of its regret and efficiency. The proposed algorithm avoids costly inference by replacing maximum likelihood estimation with online mirror descent, and avoids costly optimisation for the confidence set via an alternative construction of reward. The result is an algorithm with no leading order dependence on $\kappa$ and $O(1)$ computational cost per round.

**Strengths:**

This is a welcome contribution for the literature on contextual bandits. Improvements regards to dependence on $\kappa$ have been important steps forward in this area for various logistic bandit problems in recent years and this development is also likely to be impactful and useful.

I find the paper to be well researched and written. There are a few minor points as discussed below, but I think the paper is generally thoughtfully structured. For instance I liked how the OFU-MLogB algorithm was motivated in terms of certain concerns about efficiency and these same headings were used to frame the discussion of the algorithm's structure.

As far as I can tell the theoretical results are derived correctly, and the analytical work is sound. I have not had as much time as one would like to be 100% confident in this, so this does impact my confidence score.

**Weaknesses:**

I don't have too much to criticise in terms of the originality and significance of the paper: its contribution is an improvement upon flawed but still somewhat practically useful algorithms, so it is not a revolutionary one, but I think it is nonetheless meaningful and appropriate for NeurIPS.

I have a few points for clarification regards to the writing, experimental work, and points that I think would benefit from extra detail which I describe below, and if these are adequately addressed I should be able to increase my score following rebuttal.

**Questions:**

- What do you see as the utility of the algorithm presented in Section 2.3 over OFUMlogB? If there is not one particularly, and it is just a way to introduce important concepts and provide a correction to [12] and/or state an improved concentration result, potentially of independent interest, can you make that clearer from the outset?

- I think Theorem 2 is slightly informal: I guess that you mean the sequence of arms selected by the algorithm using decision rule (5) achieves this regret, but how is this algorithm initialised? What is its first selection, or selections until such a point as the parameter estimates are well defined?

- Which MNL-UCB are you testing in the experiments? The original from [12] or the improved one analysed in Theorem 2?

- The confidence bands on the regret presented in the Appendix are very wide. They suggest less of a statistically significant difference than the plots in the main body. I think you need to move a discussion of this to the main text, and offer some discussion as to whether the ordering of the algorithms remains stable across trials, or whether that is random also. I feel that this experiment probably could have been better designed, and would be yet more satisfied if it was possible to provide a more extensive experiment from which more meaningful conclusions could be drawn.

- Minor: at line 167, I don't think you ever really define 'feedback number' as a piece of terminology. I suggest that you should.
- Minor: at line 255, should say 'condition as Theorem 3'
- Minor: at line 67, it may be worth making clear that MNL-bandits (which people could confuse this model with) specifically will be addressed in Appendix A, it was something I was concerned was missing until I got to the end.
- Minor: at line 312: radio -> radius


**Limitations:**

Appropriate for the nature of the paper.

---

> ### Author Rebuttal · Authors · 2023-08-10
>
> Thank you for the positive evaluation and constructive comments for this paper! In the following, we will address your questions. We will further improve the paper according to your suggestions.
>
> ---
>
> **Q1**: What do you see as the utility of the algorithm presented in Section 2.3 over OFUMlogB?
>
> **A1**: Thank you for the helpful comments! As you have mentioned in the review, we regard the jointly efficient algorithm OFU-MLogB as the main contribution of this paper. The purpose of Section 2.3 is first to introduce the important concept of logistic bandit and then state the improved concentration results that can be potentially useful. In the next version, we will highlight the utility of section 2.3 at the beginning of the section. Thanks!
>
> ---
>
> **Q2:** I think Theorem 2 is slightly informal: I guess that you mean the sequence of arms selected by the algorithm using decision rule (5) achieves this regret, but how is this algorithm initialised? What is its first selection, or selections until such a point as the parameter estimates are well defined?
>
> **A2:** Thank you for pointing this issue out. For the first round, the learner can randomly select any arm for the decision set $\mathcal{X}$. After that, the decision rule (5) is well-defined. We will provide a more formal description of the algorithm in the revision.
>
> ---
>
> **Q3:** Which MNL-UCB are you testing in the experiments? The original from [12] or the improved one analysed in Theorem 2?
>
> **A3:** We implemented the original algorithm from [12]. As discussed in Q1, the purpose of the algorithm in Section 2.3 is to first introduce important concepts for the MLE-based logistic bandit method and then to bring the issue of the dependence on $K$ to the community's attention. Therefore, we initially did not consider the algorithm in Section 2.3 as a contender to be tested in the experiments. However, we admit that an empirical comparison between MNL-UCB and the improved version could further help us understand how significant the improvement on $K$ will be. We will take this as future work.
>
> ---
>
>
> **Q4:** The confidence bands on the regret presented in the Appendix are very wide....a more extensive experiment from which more meaningful conclusions could be drawn.
>
> **A4：**Thank you for the thoughtful comment! During the rebuttal phase, we realized that there could be a better way to organize our experimental results. In the original experiments, we randomly selected the arm set for each set of 20 trials, leading to a high variance in the results. We have provided a new version of our experiments for the binary case, in which we still run the algorithm for 20 trials, but the arm set is fixed. Please kindly refer to the PDF file attached to the global response for more details.
>
> ---
>
> **Q5:** it may be worth making clear that MNL-bandits (which people could confuse this model with) specifically will be addressed in Appendix A
>
> **A5:** Thank you for the suggestions. In the next version, we will include part of the related work discussion with the MNL setup in the main text.
>
> ---
>
> **Q6:** other comments on the minor issues:
>
> **A6:** We sincerely thank the reviewers for the detailed check. We will revise the paper accordingly.

---

> > ### Comment · Reviewer_6tgu · 2023-08-14
> > **Response to Rebuttal (increasing score)**
> >
> > Thanks for the systematic response to my questions and suggestions, and your willingness to undertake edits. It was particularly helpful that you were clear about where and how you will make changes - thank you. I will increase my score and confidence score on the basis of this response.

---

> > > ### Author Response · Authors · 2023-08-18
> > >
> > > Thank you for the insightful comments and for updating the score! In the next version, we will update the paper based on your suggestions and the discussion during the rebuttal phase. Specifically, we will edit Section 2 to ensure that its main purposes are properly highlighted, and add a more detailed algorithmic description for Theorem 2. In the experimental part, we will present the results with a fixed arm set for different random seeds instead of averaging all of them. We will also move some parts of the related work discussion, particularly on MNL, to the main text.

---

### Official Review · Reviewer_Rks7 · 2023-07-12

**Soundness:** 4 excellent
**Presentation:** 3 good
**Contribution:** 3 good
**Rating:** 7
**Confidence:** 2

**Summary:**

This paper considers a generalization of the (binary) logistic bandit problem to the multinomial setting, significantly improving the state of the art for that problem and even for the special binary.  For the binary logistic bandit problem, many (earlier) algorithms proposed were not optimal; their regret bounds depended on a potentially large problem dependent parameter $\kappa$ capturing the nonlinearity of the reward function.   Recent works were able to remove regret dependence on that parameter $\kappa$, though had high per-round computation (depending on the horizon $T$).

The authors propose an algorithm that (up to $\log$ terms) achieves optimal regret bounds even when specialized to the well-studied binary logistic bandit setting with per round constant computation.  Furthermore, compared to prior work for the more general multinomial setting, they not only both obtain better regret (independent of the parameter $\kappa$) but do so with faster per-round computation (constant instead of $O(T)$).


**Strengths:**

- For the multinomial setting, the proposed algorithm significantly improves on the state-of-the-art both in terms of regret bounds (esp. removing the dependence on $\kappa$) and computation (from $O(T)$ per round computation to $O(1)$).  In simple experiments, this is backed up with a (slight) reduction in empirical cumulative regret but much faster run-time.
- Impressively, the proposed algorithm even improves on the state of the art for the (binary) logistic bandit setting.  Among computationally efficient ($O(1)$ per-round complexity) methods it has better regret bounds (also in simple experiments significantly lower empirical cumulative regret with competitive run-time).  Compared to ada-OFU-ECOLog the results are more nuanced (the proposed method has better regret for small horizons but worse regret for large horizons; takes half as long for a horizon of $T=1200$).
- The algorithm and regret analyses do build on recent advances in (binary) logistic bandits, but non-trivially so.
- Overall I found the writing clear and well-organized.   Table 1 and the remarks throughout comparing and contrasting to closely related works were helpful.


**Weaknesses:**


#### Major
I do not have any major concerns.

#### Minor
- Appendix A includes discussion on differences between the problem set up of multinomial logistic bandits considered in this paper and the problem of multinomial Logit (MNL) bandits which has been considered in several recent works.  This is only a minor concern, but given close connections of both to binary logistic bandit problems, a slightly more detailed discussion of similarities (if any) with OFU based methods and analyses (like with [33] Agrawal et al.) could be helpful.


- There is an experimental design based approach for the binary setting (Mason et al. “An Experimental Design Approach for Regret Minimization in Logistic Bandits” in AAAI 2022) that I don’t think is cited.  I did not look carefully into it regarding time complexity so do not know whether it would be competitive with (ada)-OFU-ECOLog or not.

- For the experiments, the $\sqrt{T}$ dependence does not kick in for the horizon $T=1200$ (i.e. the regret appears linear).  For Fig 1(a) ada-OFU-ECOLog is initially worse but its regret is visibly concave and catches up around $t=800$.  I would suggest additional experiments with a longer horizon (maybe just with  ada-OFU-ECOLog and OFU-MLogB) to examine how long OFU-MLogB exhibits near-linear regret (and consequently how big that regret gap can grow).   Just to be clear, I think it is impressive that OFU-MLogB is better for smaller $t$ and $t$ on the scale of $10^4$ or $10^5$ might be unrealistic for applications, but the linear regret is a bit concerning.


**Questions:**


#### Minor editing suggestions
- For Table 1, I’d suggest including a comment in the caption about $\kappa$ and $\kappa_*$ (in addition to the later discussion in the main text).
- In the problem formulation explicitly mention whether or not there are there any assumptions about $\mathcal{X}$ (finite, convex, etc.).
- I’d suggest re-ordering lemmas in C.1.2 (proof of lemma 6 depends on lemma 7, proof of lemma 7 depends on lemma 8)
- For experiments, does MNL-UCB reduce to LogUCB1? If not, I’d (mildly) suggest including it in Figs 1 (a)-(b) for further reference of how well state-of-the-art multinomial methods perform against methods specifically designed for binary setting.
#### Spelling, grammar
- Line 52 argmax subscript ‘$x \in W$’
- line 276 ‘linearlized’
- line 289 ‘logistc’
- line 312 ‘radio’
- Line 953 ‘trails’
- line 537 ‘on the an’
 - Line 571 ‘anlaysis’
- there were more I did not bother to list


**Limitations:**

It is fine

---

> ### Author Rebuttal · Authors · 2023-08-10
>
> Many thanks for your great appreciation and brining the related work to us! In the following, we will address your questions. We will further improve the paper according to your suggestions.
>
> ---
>
> **Q1**: a slightly more detailed discussion of similarities (if any) with OFU based methods and analyses (like with [33] Agrawal et al.) could be helpful.
>
> **A1**: Thank you for the constructive commons. We provide more detailed comparison between MNL and MLogB problem as follows.
>
> - From the problem setup view, both settings utilize the multinomial logistic regression model to capture the probability of feedback, and they can be seen as generalizations of the binary logistic bandit problem. However, the main difference is in the mechanism to generate the feedback. The MLogB problem considers the case where there could be multiple feedbacks for the selected arm. While the MNL problem focuses on situations where the algorithm can submit multiple arms, but the feedback exhibits a binary value.
>
> - For the algorithm design view, the optimistic in the face of uncertainty (OFU) principle can be applied to both settings. However, the different problem setup leads to different algorithm design details and challenges. For instance, when analyzing the concentration property of the parameter estimator, one of the main challenges of MLogB is to handle the multinomial random noise $\epsilon_t\in\mathbb{R}^{K}$ while MNL has to tackle the potentially related arm set for each iteration.
>
> - The previous work [33] proposed a UCB-type algorithm with an improved $\tilde{O}(\sqrt{T})$ bound for the MNL problem. Apart from the differences in problem setup, the main distinction between our work and [33] is that the previous work still employs an MLE-based algorithm, requiring $O(T)$ time complexity to solve the optimization problem. We believe extending our algorithm to the MNL setting to achieve the $\tilde{O}(\sqrt{T})$ bound with constant computational cost would be an interesting direction.
>
> ---
>
> **Q2**: There is an experimental design based approach for the binary setting.
>
> **A2:** Many thanks for sharing the paper! We were not aware of it and will add the result to Table 1 for a more comprehensive comparison. In the paper, the experimental design approach has to solve a min-max optimization problem to select the arm (line 12 of Algorithm 2). It still seems unclear how to solve this min-max optimization problem efficiently.
>
> ---
>
> **Q3**:  I would suggest additional experiments with a longer horizon (maybe just with ada-OFU-ECOLog and OFU-MLogB) to examine how long OFU-MLogB exhibits near-linear regret.
>
> **A3**: Thanks for the suggestions. We agree that the time step is somewhat short in the current experiments. We have conducted additional experiments with a longer $T$ to test our method. Please kindly refer to the global rebuttal for more details. Thanks!
>
> ---
>
>
> **Q4**: For experiments, does MNL-UCB reduce to LogUCB1?
>
> **A4**: Yes, MNL-UCB can be seen as a counterpart of LogUCB1 for MLogB problem. We will make this clear in the revision.
>
> ---
>
> **Q5:** about other editing suggestions and grammar:
>
> **A5:** We sincerely thank the reviewer for the detailed check of our paper. We will revise the paper according to your suggestions. Many thanks!

---

### Official Review · Reviewer_Ru1r · 2023-07-14

**Soundness:** 2 fair
**Presentation:** 3 good
**Contribution:** 3 good
**Rating:** 6
**Confidence:** 4

**Summary:**

This paper has addressed multinomial logistic bandits whose feedback has multiple choices. This paper improves the regret bound in terms of $K$ and reduces the computation cost into constant complexity with respect to $T$. In experiments, the results support that the proposed method is much faster than the prior work.

**Strengths:**

**Computational Complexity**

This paper reduces the time complexity into $O(K^{3}d^{3})$, which is independent on $T$.
The experimental results also support that computational time is dramatically reduced in practice.

**Regret Bound**

The regret bound of the proposed method is $O(K\sqrt{T})$. This result is a substantial improvement over the best-known regret bound of $O(K^{5/4}\sqrt{\kappa T})$, where $K$ is the number of feedback values and $\kappa$ is a constant that increases exponentially in terms of the diameter of the parameter domain.
The proof seems to be correct.

**Weaknesses:**

**Lower Bounds**

Improving the regret bound with respect to $K$ seems like a minor contribution. Especially, since there is no comparison with the lower bound on regret, it is unclear if the improvement in the regret bound with respect to K is theoretically significant.

**Novelty of Analysis Techniques**

Most of the techniques used in this paper are quite similar to the generalized linear bandits (GLB).
Especially, the proof scheme of the Theorem 1 is almost the same as logistic bandits or GLB.
Furthermore, in Theorem 1, Lemma $6$ plays a crucial role, however, as mentioned in the Appendix, the difference from previous research lies solely in the bound on the norm of the error.
The remaining parts of the proof of Theorem 1 heavily depend on Abbasi-Yadkorietal. (2011).
It cannot be considered as a significant theoretical contribution.

**Questions:**

**Questions**

1. What is the memory complexity of each algorithm?
While this paper reduces the algorithm's time complexity, it may lead to requirements for more memory.
Since there is a general trade-off between time complexity and memory complexity, it would be better to discuss the memory complexity of this algorithm (and others).

2. Does this algorithm match the lower bound?
Is there any analysis for the lower bound of this setting?

3. What causes the improvement of the regret bound?
From the new analysis scheme? or finding better parameters?
If I correctly understand the paper, the improvement of the regret bound is just caused by changing the condition on $\epsilon$ from $\| \epsilon \| _{2} \leq \sqrt{K}$ to $\|\epsilon \| _{1} \leq 2$.

**Minor comments**

Table 1: the column "Constant Cost" is unnecessary part since we can check it in the column "Cost Per Round"

line 112: $\textnormal{diag}(z)$ -> $\textnormal{diag}(\sigma(z))$

**Limitations:**

Increasing the memory complexity might be a limitation of this work.

---

> ### Author Rebuttal · Authors · 2023-08-10
>
> Thank you for the careful review. In the following, we will first highlight the technical contribution of the paper then address your questions. If your concerns have been properly addressed, please consider updating your score to this paper. Thanks!
>
> ---
>
> **Q1:** novelty of analysis techniques
>
> **A1:** We would like to note that there are two parts of the paper:
>
> - an inefficient algorithm with improved $K$ (Section 2.3)
> - efficient algorithm with further improved dependence on $\kappa$. (Section 3)
>
> For the first part of the paper, we agree with your comments. While we believe the improvement on $K$ is worthy of the community's attention, we recognize that the result is somewhat built upon standard analysis from existing logistic bandit literature, with refined conditions on $\epsilon_t$. The result is not considered a main contribution of this work as we have chosen not to include it in the Introduction section.
>
> The main contribution of this paper lies in **the second part**, where we developed novel efficient algorithms for both binary and multinomial logistic bandits problems with improved bounds. Developing a computationally efficient algorithm that also maintains statistical efficiency is a non-trivial task. Most previous methods [8,9,12]  with improved dependence on $\kappa$ crucially rely on the inefficient MLE estimator. The only joint efficient algorithm is developed in the binary case and can hardly be extended to the multinomial setting. Additionally, even in the binary case, our method improves the computational efficiency of [10] from $O(\log T)$ to $O(1)$. We have highlighted the technical challenge and our contribution in Remark 2 with more details explained in Section 3.3.
>
> In summary, we recognize the reviewer's concern about the novelty of the first part but feel the second part, our main contribution, has been somewhat overlooked. This might be because we have spent slightly more pages than expected on the first part. In the next version, we will consider your comments and further emphasize the main contribution of the paper.
>
> ---
>
> **Q2:** What is the memory complexity of each algorithm?
>
> **A2:**  At every iteration, our method requires to maintain the matrices $H_t,\tilde{H}\_t,\nabla\sigma(W)$ , the single round data point $(x,y)\in \mathbb{R}^d\times\mathcal{Y}$ and the model $W\_{t+1}\in\mathbb{R}^{K\times d}$. Therefore, the storage complexity of our algorithm is $O(K^2d^2)$,  a constant in terms of $T$. In the binary case, the storage complexity becomes $O(d^2)$ which is the same as the joint efficient algorithm [10].
>
> However, the situation is different for the MLE-based algorithm [8,9,12]. This method requires storing all historical data to solve the optimization problem, leading to a storage complexity $O(dt)$ for round $t$. Additionally, the MLE-based algorithm must store the matrix $H_t^{-1}(W)\in\mathbb{R}^{Kd\times Kd}$ to perform the optimistic rule, as dictated by equation (4). Therefore, the total storage complexity of the MLE-based method amounts to $O(dt + K^2d^2)$.
>
> In summary, the storage complexity of our algorithm is the same as [10] and substantially improves the MLE-based method [8,9,12]. Thank you for the comments. We will add more discussion on the storage comparison in the revision.
>
> ---
>
> **Q3:** Does this algorithm match the lower bound? Is there any analysis for the lower bound of this setting?
>
> **A3:** Since the binary logistic bandit problem is a special case of the multinomial logistic bandit, the lower bound established in the binary case also holds for the multinomial case. First, as outlined in Remark 3, the minimax optimal rate for the binary case is $O(\sqrt{T/\kappa_*})$  as established by [9]. Our method can achieve this rate up to logarithmic factors. Besides, in Remark 3, we also provide a discussion on the tightness of the bound in terms of $\kappa$ for the multinomial case. While we admittedly found it challenging to achieve the $O(\sqrt{T/\kappa_*})$ bound, our algorithm has already achieved **the best known result** for the problem. It is an intriguing open question whether the $O(\sqrt{T/\kappa_*})$ regret bound is achievable in the MLogB problem. Furthermore, as we mentioned in Remark 1, the optimality of the $O(K)$ dependence has been discussed in [12] (the paragraph under Theorem 3), which demonstrates the tightness of our bound.
>
> ---
>
> **Q4:** What causes the improvement of the regret bound? From the new analysis scheme? or finding better parameters? If I correctly understand the paper, the improvement of the regret bound is just caused by changing the condition...
>
> **A4**: As we have discussed in response to Q1, the main contribution of this paper is to provide an **efficient algorithm** with an improved bound. The improvement of our algorithm crucially relies on novel analyses, which we highlight in Remark 2. First, to achieve the improved dependence on $\kappa$ in the MLogB problem, we introduce a novel intermediary decision that helps us prove a tight confidence set for the efficient online estimator. Besides, to reduce the computational complexity of the algorithm from $O(\log T)$ to $O(1)$, we carefully exploit a negative term in the analysis, which helps to eliminate the requirement of learning with the original loss, thus speeding up the algorithm. Thank you for the comment. We will further highlight the related part to make the technical contributions of this paper more accessible.

---

> > ### Comment · Reviewer_Ru1r · 2023-08-18
> > **Response to the rebuttal**
> >
> > I am sorry for the late response and appreciate the authors for their thoughtful responses.
> >
> > After reading their rebuttal, I agree that the improvements in both computational complexity and memory complexity compared to existing methods are noteworthy. I think using online mirror descent updates plays a crucial role. Recognizing sufficient contributions, I have raised my score.
> >
> > One last concern is that it might be necessary to mention in the main paper that the regret improvement is due to a special parameter choice rather than a new analysis technique. This clarification would help future researchers easily understand this aspect.

---

> > > ### Author Response · Authors · 2023-08-18
> > >
> > > Thank you for the helpful comments and for updating the score! In the next version, we will revise the paper based on your suggestions and the discussion in the rebuttal phase. During the rebuttal phase, we realized that the current writing of Section 2 does not accurately reflect its purpose: to first introduce the important concept of logistic bandit and then bring the issue of $K$ to the community's attention. In the next version, we will further clarify the utility of Section 2 at the beginning of this section. We will also revise Remark 1 in Section 2 to clearly convey that the improved $K$ bound for the MLE estimator mainly comes from a better upper bound for the noise, while the analyses are largely based on existing techniques in logistic bandit literature.

---

### Official Review · Reviewer_XVWd · 2023-07-16

**Soundness:** 2 fair
**Presentation:** 3 good
**Contribution:** 3 good
**Rating:** 6
**Confidence:** 3

**Summary:**


This paper examines multinomial logistic bandits (MLogB), a problem where the learner's action $x_t$ produces feedback $y_t$ with $K+1$ possible outcomes. The probabilities of these outcomes are modeled using a logistic model. In real-world scenarios like online advertising, customers may provide various types of feedback, such as 'buy now', 'add to cart', 'view related item', or simply leave without any click.

 The MLogB problem assumes that at each time $t$, the learner selects an action $x_t$ from $\mathcal{X}$. Each outcome $k\in[K]$ is associated with a latent parameter $w_*^{(k)}\in \mathbb{R}^d$ and the probability of the outcome $P[y_t=k|x_t]$ follows the logistic model,
$P[y_t|x_t]=exp((w_*^{(k)})^\top x_t)/(1+\sum_j exp((w_*^{(j)})^\top x_t))=\sigma(z)_k$.

Then the expected reward of the learner's action is defined$ \sum_t  \rho^\top \sigma (W_* x_t)$ where $\rho$ is a known rewards vector.

The authors first introduce an improved version of MNL UCB algorithm [12] with the optimal dependency on K. With an improved concentration set for OFU in Theorem1, MLE based algorithm achieves $O(Kd \log(T)\sqrt{\kappa ST})$ in theorem 2.

In the next section, they present an efficient and improved algorithm (OFU-MlogB). For the efficiency, they address the computation cost of MLE and OFU methods. Instead of using MLE, they suggest using online mirror descent algorithm to estimate $W$, incurring $O(1)$ cost per round and achieving $\kappa$=independent confidence set in Theorem 3 (similar to theorem 1).  For the optimistic reward construction, they suggested a novel optimistic reward that can be solved in a constant time per round by introducing bonus terms for exploration in proposition 1.  This method archieves an improved regret bound of $\tilde{O}(Kd\sqrt{T})$ in Theorem 4 and constant computation cost compared to $\tilde{O}(K^{5/6}\sqrt{\kappa T})$ regret bound and $O(T)$ computation cost in [12]. In the binary setting where $K=1$ the proposed algorithm archives a minimax regret bound of $O(\sqrt{T/\kappa_*})$ with an optimistic rule similar to [9,10] while maintaining lower computation cost.

In their analysis for theorem 3, they leverage negative terms from the efficient update of mirror descent and a novel intermediary prediction construction designed for multiclass logistic loss.

Lastly, they provide experimental results for both binary and multi-class settings. In the binary case, the proposed algorithm demonstrates lower computational overhead while delivering comparable performance to the state-of-the-art algorithm, ada-OFU-ECOLog[10]. For the multinomial setting, the proposed algorithm exhibits improved computational efficiency compared to MNL-UCB [12], while achieving similar performance in terms of regret.



**Strengths:**

1. In the binary case, the proposed algorithm attains the minimax optimal guarantee while reducing the computation cost per round from $O(\log T)$ to $O(1)$.

2. In the multinomial case, the proposed algorithm achieves a regret bound of $\tilde{O}(K\sqrt{T})$ with constant computation cost, surpassing the performance of the best-known algorithm which achieves a regret bound of $\tilde{O}(K^{5/4}\sqrt{\kappa T})$ with $O(T)$ computation cost per round.

3. The authors present empirical evidence demonstrating the improved computation cost of the proposed algorithm.


**Weaknesses:**

1. Algorithm 1 requires the computation of the matrix inverse and projection, resulting in computational costs of $O(d^2K^3)$ and $O(K^3d^3)$ per round.

2. In the binary case, it appears that achieving optimal regret in Algorithm 1 necessitates the use of a linear optimistic rule (as stated in Corollary 1), rather than relying on an efficient optimistic reward with a bonus term. It is not clear why a linear model works (details are in Question 1).

3. In the experimental evaluation, concerning the binary case, the proposed algorithm exhibits regret that increases linearly, while the previously suggested algorithm, ada-OFU-ECOLog, demonstrates sublinear regret. Regarding the multinomial case, although the proposed algorithm achieves a theoretically superior regret bound considering $K$ and $\kappa$, it shows similar performance to MNL-UCB. This observation does not provide a clear validation of the theoretical analysis.

4. In the multinomial case, there might exist a difference between the attained regret bound of $\sqrt{T}$ and a regret lower bound.

**Questions:**

1. In corollary 1, what is the reason the optimistic rule follows a linear model, $\arg\max_w w^\top x$, rather than $\arg\max_w \rho^\top \sigma(w^\top x)$?

2. Is the computation cost for the matrix inverse and projection of the proposed algorithm, which is O(K^3d^3), commonly observed in previous literature like [10]? Otherwise, it appears that the improvement in computation cost mainly applies to scenarios with small values of K and d.

3. In the conducted experiments, why was the time horizon set to $T=1200$, which is relatively smaller compared to the larger time steps typically used in conventional bandit literature?

4. Furthermore, in the experimental results (a), what could be the possible explanation for the observation that the proposed algorithm exhibits linearly increasing regret, while ada-OFU-ECOLog demonstrates sublinear regret?





**Limitations:**

1. The details regarding the method mentioned in Corollary 1 may not be clear (question 1).

2. The experimental results do not seem to provide clear evidence of regret for the proposed algorithm due to linearly increasing patterns and small $T$. (questions 3,4)

I am open to revising my evaluation if these concerns are addressed.

Minor comments:

1.line 99,112: should it be changed from $diag(z)$ to $diag(\sigma(z))$?

2. line 333 ,336: $O(log1)$-> $O(1)$; $O(T/\kappa^*)$-> $O(\sqrt{T/\kappa^*})$

---

> ### Author Rebuttal · Authors · 2023-08-10
>
> Thank you for your insightful comments. We will address your questions below and make enhancements to the paper based on your valuable suggestions.
>
> ---
>
> **Q1:** what is the reason the optimistic rule follows a linear model.
>
> **A1:** We are grateful to the reviewer for highlighting the ambiguity in our presentation. We will now clarify the equivalence between $\arg\max w^\top x$ and $\arg\max \rho^\top\sigma(w^\top x)$ in the binary case. We will now clarify this relationship. Specifically, as we discussed in line 96-97, the binary logistic bandit is a special case of MLogB with $K=1$ and $\rho =1$. In such a case, the reward function is exactly $r(x) = \sigma(w^\top x)$, where $\sigma:\mathbb{R}\rightarrow\mathbb{R}$ is a 1-dim function.  Given $\sigma$ is a monotonically increasing, we can solve $\arg\max \rho^\top\sigma(w^\top x)$ by  $\arg\max w^\top x$.  We provide more detailed will make the equivalence of the two equations clear in the revision.
>
> ---
>
> **Q2:** Is the computation cost for the matrix inverse and projection of the proposed algorithm, which is $O(K^3d^3)$, commonly observed in previous literature like [10]?
>
> **A2:** Thank you for the insightful comments. As demonstrated in the proof of [10, Proposition 8], previous literature also necessitates maintaining the **matrix inverse** and performing $O(\log t)$ **projected gradient steps** (PGD) at each iteration to solve their optimization problem. In contrast, the main computational advantage of our algorithm lies in provably reducing the required PGD steps from $O(\log t)$ to $O(1)$.
>
> Specifically, both [10] and ours share the same matrix inverse cost of $O(d^2)$ in the binary case. For each of the $O(\log t)$ PGD times, [10] employ a distinct method to project the unconstrained solution onto an ellipsoid [10, Lemma 13], resulting in a computational cost of $O(d^2\log t)$. While this projection operation saves a $d$ factor, it incurs an additional cost of $O(\log t)$ when compared to our $O(d^3)$ cost. In scenarios with large dimensions, where $d = \Omega(\log T)$, we can also utilize the same projection method found in [10] to have improved the dependence on dimension. We will provide a more detailed discussion of the computational cost of projection in the revision. Thanks!
>
> ---
>
> **Q3:** The experimental results do not seem to provide clear evidence of regret for the proposed algorithm due to linearly increasing patterns and small $T$ (question 3 and 4).
>
> **A3:** Thank you for the suggestion. During the rebuttal period, we reorganized the experiments for the binary case with a longer running time $T$ and presented the results in a more suitable way. Please kindly refer to the global response for more details. Thanks!

---

> > ### Comment · Reviewer_XVWd · 2023-08-10
> > **Thank you for your detailed response**
> >
> > Thank you for your detailed response. Most of my concerns are resolved by the responses so I raised my score from 5 to 6. Based on your reply, it seems to be better to clarify the tradeoff between $d$ and $\log t$ in computation cost and provide further clarification regarding the comparison for dependency on $K$ in the final version.

---

> > > ### Author Response · Authors · 2023-08-18
> > >
> > > Thank you for the thoughtful questions and for updating the score! In the next version, we will revise the paper according to your suggestions and the discussion in the rebuttal phase. Specifically, we will add a discussion on the time complexity tradeoff between $d$ and $\log t$ when performing the projection step. We will also clarify the comparison regarding the dependency on $K$ and reorganize the experiment section to present the comparison on fixed arm sets with different random seeds and longer time horizons.

---

### Author Rebuttal · Authors · 2023-08-10

We would express our heartfelt thanks to all reviewers for their careful review and constructive feedback. After carefully considering the comments from reviewers XVWd, Rks7, 6tgu, and ojs3, we conduct additional experiments to support the effectiveness of our algorithm by a more suitable way of organizing the results. The results are presented in the attached PDF.

**Setup for original experimental**. In Section 4, we report the average performance of all compared algorithms over 20 trials. As detailed in Appendix E (sorry for that, we should mention this in the main text), the arms are randomly sampled in each trial of our experiments, which finally leads to a result with an inherently large variance.

**Setup for additional Experiments**: During the rebuttal, to more accurately present the empirical performance of the compared algorithms, we reorganized the experiments by **fixing the arm set** for the 20 trials. In the attached PDF, we report the mean and variance of the compared algorithm across these 20 trials, under 6 different configurations of the randomly generated arm set. Additionally, we extended the experiment for a longer time horizon with **$T=6000$**. The results show that OFU-MLogB is comparable (slightly worse) with the state-of-the-art joint efficient algorithm for the binary case (ada-OFU-ECOLog), but achieves much better performance than O2LM.

Due to time constraints, we were only able to conduct the experiments for the binary case. We are happy to provide additional results for the multinomial case at a later date. Thanks!

---

### Decision · Program_Chairs · 2023-09-21

**Decision:**

Accept (spotlight)

**Comment:**

On the fundamental problem of online multi-class logistic bandits, this paper makes a solid contribution of improving the regret bound significantly and improving the computational complexity.